# GLGENN: A Novel Parameter-Light Equivariant Neural Networks Architecture Based on Clifford Geometric Algebras

**Ekaterina Filimoshina** [1][2]   **Dmitry Shirokov** [1][3]

## Abstract

We propose, implement, and compare with competitors a new architecture of equivariant neural networks based on geometric (Clifford) algebras: Generalized Lipschitz Group Equivariant Neural Networks (GLGENN). These networks are equivariant to all pseudo-orthogonal transformations, including rotations and reflections, of a vector space with any non-degenerate or degenerate symmetric bilinear form. We propose a weight-sharing parametrization technique that takes into account the fundamental structures and operations of geometric algebras. Due to this technique, GLGENN architecture is parameter-light and has less tendency to overfitting than baseline equivariant models. GLGENN outperforms or matches competitors on several benchmarking equivariant tasks, including estimation of an equivariant function and a convex hull experiment, while using significantly fewer optimizable parameters.

## 1. Introduction

Equivariant neural networks are a class of neural networks that explicitly incorporate symmetries into their architecture, making them well-suited for tasks that inherently require equivariance to transformations with respect to some group's action (e.g., rotations, permutations, translations, etc.). Equivariant neural networks were introduced in Cohen & Welling, 2016 and have since been extensively developed and applied in a wide range of tasks in computer and natural sciences. These applications include modeling dynamical systems (Finzi et al., 2021), particle physics (Ruhe et al., 2023; Finzi et al., 2021), analyzing molecular properties and

protein structures (Townshend et al., 2021; Liu et al., 2024; Satorras et al., 2021; Pepe et al., 2024; Fuchs et al., 2020), estimating arterial wall-shear stress (Brehmer et al., 2023), processing tasks involving point clouds (Thomas et al., 2018; Fuchs et al., 2020), motion capture (Liu et al., 2024), robotic planning (Brehmer et al., 2023), etc. Early works addressing equivariance with respect to pseudo-orthogonal transformations (pseudo-orthogonal groups) include Cohen & Welling, 2016; 2017; Weiler & Cesa, 2019; Thomas et al., 2018; Weiler et al., 2018; Anderson et al., 2019; Finzi et al., 2021.

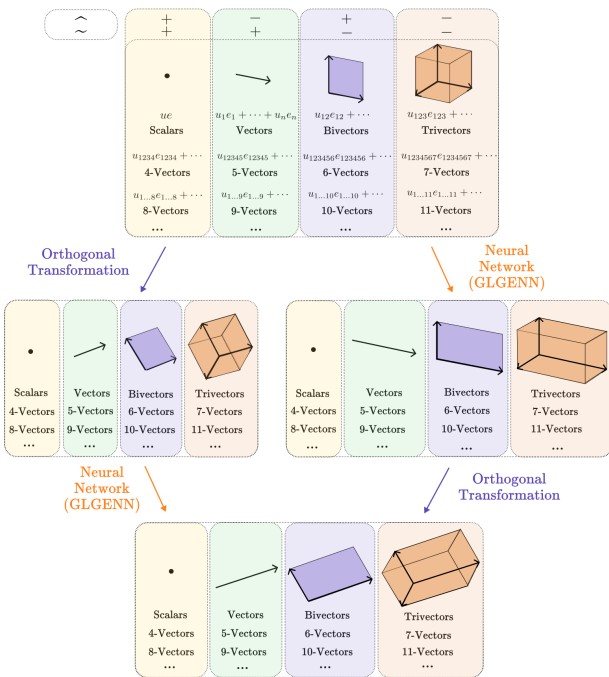

*Figure 1.* GLGENN is an architecture of neural networks equivariant with respect to any pseudo-orthogonal transformation. Inputs and outputs are represented as multivectors (elements of geometric algebras), which encode various geometric quantities such as scalars, vectors, oriented areas (bivectors) and volumes (trivectors), and higher-dimensional objects (4-vectors, etc.). GLGENN are parameter-light, since they operate in a unified manner across 4 fundamental subspaces of geometric algebras defined by the grade involution ($\widehat{\phantom{a}}$) and reversion ($\widetilde{\phantom{a}}$); they processes geometric objects in groups with a step size of 4.

[1]HSE University, Moscow, Russia [2] Skolkovo Institute of Science and Technology, Moscow, Russia [3] Institute for Information Transmission Problems of the Russian Academy of Sciences, Moscow, Russia. Correspondence to: Ekaterina Filimoshina <filimoshinaek@gmail.com>, Dmitry Shirokov <dm.shirokov@gmail.com>.

*Proceedings of the $42^{nd}$ International Conference on Machine Learning*, Vancouver, Canada. PMLR 267, 2025. Copyright 2025 by the author(s).

This work introduces a novel neural network architecture that is equivariant with respect to any pseudo-orthogonal transformation, including rotations and reflections. Our approach is based on the well-known mathematical framework of (Clifford) geometric algebras (GAs), which have found applications in various scientific fields, including physics (Doran & Lasenby, 2003; Hestenes & Sobczyk, 1984), computer science (Bayro-Corrochano, 2019; Brandstetter et al., 2023; Dorst et al., 2002), engineering (Dorst et al., 2002), and other scientific fields. Works on applications of GAs in neural networks include Pearson & Bisset, 1994; Bayro-Corrochano & Buchholz, 1997; Buchholz & Sommer, 2008; Kuroe, 2011; Brandstetter et al., 2023.

In particular, GAs provide an efficient and elegant representation of pseudo-orthogonal transformations via spin groups, Lipschitz groups (sometimes also called Clifford groups in the literature), and twisted adjoint representations. Specifically, for any pseudo-orthogonal matrix, there exists a corresponding element of the Lipschitz group in GAs that induces the same transformation of vectors through the twisted adjoint action. The transition from matrix formalism to GA formalism is highly beneficial and inspiring for applications, as it introduces a rich set of operations that are naturally defined within the GA framework.

The idea of leveraging GAs to construct pseudo-orthogonal group equivariant neural networks was first proposed by the Amsterdam Machine Learning Lab (University of Amsterdam). Their pioneering work Ruhe et al., 2023 introduced Clifford Group Equivariant Neural Networks (CGENN) and served as the foundation for many subsequent research works. Brehmer et al., 2023 propose Geometric Algebra Transformer (GATr), which incorporates GA into ordinary transformer architecture and outperforms traditional non-geometric baselines in $N$-body modeling and robotic planning. Zhdanov et al., 2024 introduce Clifford-Steerable Convolutional Neural Networks based on CGENN approach, demonstrating superior performance in tasks related to fluid dynamics and relativistic electrodynamics forecasting. Liu et al., 2024 introduce Clifford Group Equivariant Simplicial Message Passing Networks, a method for steerable $\mathrm{E}(n)$-equivariant message passing on simplicial complexes. Pepe et al., 2024 use CGENN to predict protein coordinates to estimate the 3D structure of a protein.

The above mentioned papers demonstrate the importance of Lipschitz groups in GA as a fundamental tool for building expressive state-of-the-art equivariant neural networks. However, one major challenge in the design of GA-based equivariant neural networks is overparameterization. This issue often leads to a tendency to overfit, particularly in cases where the training dataset is small – a common scenario in natural science applications where such networks are typically employed. Additionally, the excessive number of parameters results in inefficient training times. The goal of this work is to present a novel, parameter-light equivariant neural network architecture, which balances between expressiveness of CGENN and parameter efficiency. We call this architecture *Generalized Lipschitz Group Equivariant Neural Networks (GLGENN)*, see Fig. 1. Our approach introduces a new parameter-light parametrization technique, which enables our model to either outperform or match existing equivariant models while significantly reducing the number of trainable parameters. The key idea is to design a novel weight-sharing approach (Lecun et al., 1998) for GA-based neural networks that respects the fundamental algebraic structures of GAs, thereby improving efficiency without sacrificing expressive power.[1]

The key contributions are as follows:

- **Introduction of generalized Lipschitz groups.** We introduce and study a new class of Lie groups in arbitrary geometric algebra, which are related to pseudo-orthogonal groups and useful for construction of equivariant neural networks.

- **Design and implementation of GLGENN.** We construct a novel, parameter-light architecture of pseudo-orthogonal groups equivariant neural networks based on geometric algebras. Code is available at https://github.com/katyafilimoshina/glgenn

- **Superior performance.** GLGENN achieve state-of-the-art performance on benchmark equivariant tasks with significantly fewer trainable parameters.

The paper is organized as follows. Section 2 provides all necessary definitions related to GAs and equivariant neural networks. In Section 3, we present our main theoretical results and introduce and study generalized Lipschitz groups. Section 4 applies these results for GLGENN construction. In Section 5, we evaluate GLGENN through experiments. All the mathematical details and proofs can be found in the appendix.

---

[1]Application of the generalized Lipschitz groups instead of ordinary Lipschitz groups allows to achieve parameter efficiency. The generalized Lipschitz groups are important because they preserve the four fundamental subspaces of GAs under the significant operation of the twisted adjoint representation. We prove that equivariance of a mapping w.r.t. these groups, as well as the ordinary Lipschitz groups, implies its orthogonal groups equivariance. The key distinction is that the generalized Lipschitz groups contain ordinary Lipschitz groups as subgroups. As a result, the set of operations equivariant w.r.t. the generalized Lipschitz groups is a subset of the set of operations equivariant w.r.t. ordinary Lipschitz groups. This reduction in the number of 'degrees of freedom' encourages us to parametrize operations in layers in a more 'economic' way (there is a smaller number of parameters that we can place). Specifically, in all GLGENN layers, we employ such equivariant operations as projections of inputs-multivectors onto the four fundamental subspaces of GAs mentioned above, whereas CGENN relies on projections onto the subspaces of fixed grades. GLGENN and CGENN layers parameterize linear combinations, products, and normalizations of the corresponding projections.

## 2. Theoretical Background

### 2.1. Geometric (Clifford) Algebras

Let us consider *(Clifford) geometric algebra (GA)* (Hestenes & Sobczyk, 1984; Lounesto, 1997; Porteous, 1995) $C\ell(V) = C\ell_{p,q,r}$, $p + q + r = n \geq 1$, over a vector space $V$ with a symmetric bilinear form $\flat$ and the corresponding quadratic form $\mathsf{q}$, where $V$ can be real $\mathbb{R}^{p,q,r}$ or complex $\mathbb{C}^{p+q,0,r}$. We use $\mathbb{F}$ to denote the field of real numbers $\mathbb{R}$ in the first case and the field of complex numbers $\mathbb{C}$ in the second case. In this work, we consider both the case of the non-degenerate GAs $C\ell_{p,q} := C\ell_{p,q,0}$, $r = 0$, and the case of the degenerate GAs $C\ell_{p,q,r}$, $r \neq 0$. We use $\Lambda_r$ to denote the subalgebra $C\ell_{0,0,r}$, which is the Grassmann (exterior) algebra (Doran & Lasenby, 2003; Lounesto, 1997). A well-known example of geometric algebra is the spacetime algebra $C\ell_{1,3} = C\ell(\mathbb{R}^{1,3})$, associated with the Minkowski space $\mathbb{R}^{1,3}$, which plays a central role in relativistic physics. For the readers' convenience, in Appendix B, we illustrate all key theoretical concepts introduced in this section with examples in the setting of $C\ell_{1,3}$.

The identity element of $C\ell_{p,q,r}$ is denoted by $e \equiv 1$, the generators are denoted by $e_a$, $a = 1, \ldots, n$. The generators satisfy the following conditions: $e_a e_b + e_b e_a = 2\eta_{ab} e$ for any $a, b = 1, \ldots, n$, where $\eta = (\eta_{ab})$ is the diagonal matrix with $p$ times $+1$, $q$ times $-1$, and $r$ times $0$ on the diagonal in the real case $C\ell(\mathbb{R}^{p,q,r})$ and $p + q$ times $+1$ and $r$ times $0$ on the diagonal in the complex case $C\ell(\mathbb{C}^{p+q,0,r})$.

Let us consider the *subspaces $C\ell_{p,q,r}^k$ of fixed grades* $k = 0, \ldots, n$. Their elements are linear combinations of the basis elements $e_{a_1 \ldots a_k} := e_{a_1} \cdots e_{a_k}$, $a_1 < \cdots < a_k$. The grade-0 subspace can be denoted by $C\ell^0$ without the lower indices $p, q, r$, since it does not depend on the GA's signature. We have $C\ell_{p,q,r}^k = \{0\}$ for $k < 0$ and $k > n$. We can represent any element *(multivector)* $U \in C\ell_{p,q,r}$ as a sum of $n + 1$ elements $U = \langle U \rangle_0 + \cdots + \langle U \rangle_n$, $\langle U \rangle_k \in C\ell_{p,q,r}^k$, $k = 0, \ldots, n$. We call any operation of the form

$$U \mapsto \sum_{k=0}^{n} \lambda_k \langle U \rangle_k, \quad \lambda_k = \pm 1, \tag{1}$$

a *conjugation operation* in $C\ell_{p,q,r}$. Consider conjugation operations called *grade involution* and *reversion*. The grade involute of an element $U \in C\ell_{p,q,r}$ is denoted by $\widehat{U}$ and the reversion is denoted by $\widetilde{U}$, and they are defined for an arbitrary $U \in C\ell_{p,q,r}$ as

$$\widehat{U} := \sum_{k=0}^{n} (-1)^k \langle U \rangle_k, \quad \widetilde{U} := \sum_{k=0}^{n} (-1)^{\frac{k(k-1)}{2}} \langle U \rangle_k. \tag{2}$$

The composition of grade involution and reversion is called *Clifford conjugation*. The Clifford conjugate of $U \in C\ell_{p,q,r}$ is denoted by $\widehat{\widetilde{U}}$. The grade involution defines the *even* $C\ell_{p,q,r}^{(0)}$ and *odd* $C\ell_{p,q,r}^{(1)}$ *subspaces*:

$$C\ell_{p,q,r}^{(l)} := \{U \in C\ell_{p,q,r} : \ \widehat{U} = (-1)^l U\}, \quad l = 0, 1. \tag{3}$$

*Table 1.* Signs of the projections of $U = \langle U \rangle_{\overline{0}} + \langle U \rangle_{\overline{1}} + \langle U \rangle_{\overline{2}} + \langle U \rangle_{\overline{3}} \in C\ell_{p,q,r}$ for the grade involution ($\widehat{\phantom{x}}$), reversion ($\widetilde{\phantom{x}}$), and Clifford conjugation ($\widehat{\widetilde{\phantom{x}}}$) acting on it.

| $C\ell_{p,q,r}^{\overline{k}}$ | $k = 0$ | $k = 1$ | $k = 2$ | $k = 3$ |
|:---:|:---:|:---:|:---:|:---:|
| $\widehat{\phantom{x}}$ | $+$ | $-$ | $+$ | $-$ |
| $\widetilde{\phantom{x}}$ | $+$ | $+$ | $-$ | $-$ |
| $\widehat{\widetilde{\phantom{x}}}$ | $+$ | $-$ | $-$ | $+$ |

We can represent any element $U \in C\ell_{p,q,r}$ as a sum $U = \langle U \rangle_{(0)} + \langle U \rangle_{(1)}$, $\langle U \rangle_{(l)} \in C\ell_{p,q,r}^{(l)}$, $l = 0, 1$. We use angle brackets $\langle \cdot \rangle_{(l)}$ to denote the operation of *projection* of multivectors and sets onto $C\ell_{p,q,r}^{(l)}$. The grade involution and reversion define four subspaces $C\ell_{p,q,r}^{\overline{0}}$, $C\ell_{p,q,r}^{\overline{1}}$, $C\ell_{p,q,r}^{\overline{2}}$, and $C\ell_{p,q,r}^{\overline{3}}$ (they are called the *subspaces of quaternion types* $0, 1, 2$, and $3$ respectively in Shirokov, 2012a;b):

$$C\ell_{p,q,r}^{\overline{k}} := \{U \in C\ell_{p,q,r} : \ \widehat{U} = (-1)^k U,$$
$$\widetilde{U} = (-1)^{\frac{k(k-1)}{2}} U\}, \quad k = 0, 1, 2, 3. \tag{4}$$

In other words, $C\ell_{p,q,r}^{\overline{k}} := C\ell_{p,q,r}^k \oplus C\ell_{p,q,r}^{k+4} \oplus C\ell_{p,q,r}^{k+8} \oplus \cdots$ for $k = 0, 1, 2, 3$. A discussion on the significance of the grade involution, reversion, and these subspaces is provided in Appendix C. Note that the GA $C\ell_{p,q,r}$ can be represented as a direct sum of the subspaces $C\ell_{p,q,r}^{\overline{k}}$, $k = 0, 1, 2, 3$, and viewed as $\mathbb{Z}_2 \times \mathbb{Z}_2$-graded algebra with respect to the commutator and anticommutator (Shirokov, 2018). We can represent any element $U \in C\ell_{p,q,r}$ as a sum of $4$ elements: $U = \langle U \rangle_{\overline{0}} + \langle U \rangle_{\overline{1}} + \langle U \rangle_{\overline{2}} + \langle U \rangle_{\overline{3}}$, where $\langle U \rangle_{\overline{m}} \in C\ell_{p,q,r}^{\overline{m}}$, $m = 0, 1, 2, 3$. The action of the grade involution, reversion, and Clifford conjugation on a multivector $U$ of the form above is summarized in Table 1. Note that $C\ell_{p,q,r}^{\overline{k}} = C\ell_{p,q,r}^k$, $k = 0, 1, 2, 3$, in the cases $n \leq 3$.

### 2.2. Lipschitz Groups and Twisted Adjoint Representations

We use the upper index $\times$ to denote the subset $\mathrm{H}^\times$ of all invertible elements of any set $\mathrm{H}$. Let us consider the *adjoint representation* $\mathrm{ad}$ acting on the group of all invertible elements $\mathrm{ad} : C\ell_{p,q,r}^\times \to \mathrm{Aut}(C\ell_{p,q,r})$ as $T \mapsto \mathrm{ad}_T$, where $\mathrm{ad}_T : C\ell_{p,q,r} \to C\ell_{p,q,r}$:

$$\mathrm{ad}_T(U) := TUT^{-1}, \quad U \in C\ell_{p,q,r}, \ T \in C\ell_{p,q,r}^\times. \tag{5}$$

Also let us consider the *twisted adjoint representation*. This notion is introduced by Atiyah, Bott, and Shapiro (Atiyah et al., 1964) in a particular case, and there are two ways how to generalize it (see motivation in Appendix E). The first approach (Choi et al., 2002; Harvey, 1990; Lundholm & Svensson, 2009) is to define it as the operation $\breve{\mathrm{ad}}$ acting on the group of all invertible elements $\breve{\mathrm{ad}} : C\ell_{p,q,r}^\times \to \mathrm{Aut}(C\ell_{p,q,r})$ as $T \mapsto \breve{\mathrm{ad}}_T$ with $\breve{\mathrm{ad}}_T : C\ell_{p,q,r} \to C\ell_{p,q,r}$:

$$\breve{\mathrm{ad}}_T(U) := \widehat{T} U T^{-1}, \quad U \in C\ell_{p,q,r}, \ T \in C\ell_{p,q,r}^\times. \tag{6}$$

The second approach (Helmstetter & Micali, 2008; Knus, 1991; Walpuski, 2022) is to define the twisted adjoint representation as the operation $\tilde{\mathrm{ad}} : Cl_{p,q,r}^{\times} \to \mathrm{Aut}(Cl_{p,q,r})$ acting as $T \mapsto \tilde{\mathrm{ad}}_T$ with $\tilde{\mathrm{ad}}_T : Cl_{p,q,r} \to Cl_{p,q,r}$:

$$\tilde{\mathrm{ad}}_T(U) := T\langle U \rangle_{(0)} T^{-1} + \widehat{T} \langle U \rangle_{(1)} T^{-1} \quad (7)$$

for any $U \in Cl_{p,q,r}$ and $T \in Cl_{p,q,r}^{\times}$. The representation $\tilde{\mathrm{ad}}$ has the following properties, which we prove in Lemma E.2 and apply in construction of equivariant layers in Section 3.3. The properties of $\mathrm{ad}$ and $\check{\mathrm{ad}}$, are considered in Lemma E.2 as well.

**Lemma 2.1.** *Let* $W \in Cl_{p,q,r}^{\times}$, $U, V \in Cl_{p,q,r}$, *and* $T \in (Cl_{p,q,r}^{(0)\times} \cup Cl_{p,q,r}^{(1)\times})\Lambda_r^{\times}$, *where the notation* $(Cl_{p,q,r}^{(0)\times} \cup Cl_{p,q,r}^{(1)\times})\Lambda_r^{\times} := \{ab \mid a \in Cl_{p,q,r}^{(0)\times} \cup Cl_{p,q,r}^{(1)\times}, \ b \in \Lambda_r^{\times}\}$ *stands for the product of two groups. Then* $\mathrm{ad}_W$ *satisfies:*

$$\tilde{\mathrm{ad}}_W(U + V) = \tilde{\mathrm{ad}}_W(U) + \tilde{\mathrm{ad}}_W(V), \quad \tilde{\mathrm{ad}}_W(c) = c \quad (8)$$

*for any* $c \in Cl^0$. *Moreover,* $\tilde{\mathrm{ad}}_T$ *satisfies multiplicativity:*

$$\tilde{\mathrm{ad}}_T(UV) = \tilde{\mathrm{ad}}_T(U)\tilde{\mathrm{ad}}_T(V). \quad (9)$$

Consider the *Lipschitz groups* $\tilde{\Gamma}_{p,q,r}^1$ (Benn & Tucker, 1987; Lounesto, 1997; Ablamowicz, 1986; Brooke, 1978; 1980):

$$\tilde{\Gamma}_{p,q,r}^1 := \{T \in Cl_{p,q,r}^{\times} : \ \tilde{\mathrm{ad}}_T(Cl_{p,q,r}^1) \subseteq Cl_{p,q,r}^1\}, \quad (10)$$

where $\tilde{\mathrm{ad}}_T(Cl_{p,q,r}^1) = \widehat{T} Cl_{p,q,r}^1 T^{-1}$. These groups are often considered in the literature in the case of the non-degenerate geometric algebra $Cl_{p,q}$. The definition (10) straightforwardly generalizes the common definition to the case of arbitrary $Cl_{p,q,r}$. The well-known spin groups (see details in Appendix E) in the non-degenerate (Benn & Tucker, 1987; Lounesto, 1997; Porteous, 1995) and degenerate (Ablamowicz, 1986; Crumeyrolle, 1990; Brooke, 1978; 1980; Dereli et al., 2010) cases, which have various applications in physics, are normalized subgroups of $\tilde{\Gamma}_{p,q,r}^1$ defined using the following *norm functions* $\psi, \chi : Cl_{p,q,r} \to Cl_{p,q,r}$ of the GA elements:

$$\psi(U) := \widetilde{U}U, \qquad \chi(U) := \widehat{\widetilde{U}}U. \quad (11)$$

### 2.3. Equivariant Mappings and Neural Networks

Suppose $G$ is a group and $\circ_X$ and $\circ_Y$ are its actions on the sets $X$ and $Y$ respectively. A *function* (network) $L : X \to Y$ is called *G-equivariant* (see, e.g., Bredon, 1972; Pitts, 2013; Yarotsky, 2022) iff it commutes with these actions:

$$L(g \circ_X x) = g \circ_Y L(x), \quad \forall g \in G, \quad \forall x \in X. \quad (12)$$

The definition means that we get the same output if we transform the input to the neural network or transform the output. We prove several general statements about equivariance with respect to an arbitrary group in Appendix G.

## 3. Theoretical Results
### 3.1. Generalized Lipschitz Groups

In this section, we introduce and study the generalized Lipschitz groups $\tilde{\Gamma}_{p,q,r}^{\overline{k}}$, $k = 0, 1, 2, 3$, in the case of arbitrary degenerate or non-degenerate geometric algebra $Cl_{p,q,r}$. For GLGENN, we are mainly interested in $\tilde{\Gamma}_{p,q,r}^{\overline{1}}$, however other groups $\tilde{\Gamma}_{p,q,r}^{\overline{k}}$, $k = 0, 2, 3$, are necessary to prove the main statements about $\tilde{\Gamma}_{p,q,r}^{\overline{1}}$ and serve as auxiliary tools. The (ordinary) Lipschitz groups $\tilde{\Gamma}_{p,q,r}^1$ (10) preserve the subspace $Cl_{p,q,r}^1$ of the first grade under the twisted adjoint representation $\tilde{\mathrm{ad}}$ (6). The *generalized Lipschitz groups* preserve the subspaces $Cl_{p,q,r}^{\overline{k}}$, $k = 0, 1, 2, 3$, determined by the grade involution and reversion (4), under the same representation $\tilde{\mathrm{ad}}$:

$$\tilde{\Gamma}_{p,q,r}^{\overline{k}} := \{T \in Cl_{p,q,r}^{\times} : \ \tilde{\mathrm{ad}}_T(Cl_{p,q,r}^{\overline{k}}) \subseteq Cl_{p,q,r}^{\overline{k}}\}. \quad (13)$$

The groups $\tilde{\Gamma}_{p,q,r}^{\overline{k}}$, $k = 0, 1, 2, 3$, can be considered as generalizations of Lipschitz groups because of the following theorem.

**Theorem 3.1.** *The (ordinary) Lipschitz groups are subgroups of the generalized Lipschitz groups and coincide with some of them in the low-dimensional case:*

$$\tilde{\Gamma}_{p,q,r}^1 \subseteq \tilde{\Gamma}_{p,q,r}^{\overline{1}} \subseteq \tilde{\Gamma}_{p,q,r}^{\overline{k}}, \quad k = 0, 1, 2, 3, \quad \forall n; \quad (14)$$

$$\tilde{\Gamma}_{p,q,r}^1 = \tilde{\Gamma}_{p,q,r}^{\overline{1}}, \qquad n \leq 4. \quad (15)$$

*Proof.* The proof is based on the equivalent definitions of $\tilde{\Gamma}_{p,q,r}^{\overline{k}}$, $k = 0, 1, 2, 3$, and is provided in Appendix F (see Corollary F.4, Remark F.6, and Theorem F.7). $\square$

In this work, we construct neural networks that are equivariant with respect to the action $\tilde{\mathrm{ad}}$ of the generalized Lipschitz groups $\tilde{\Gamma}_{p,q,r}^{\overline{1}}$ (13). To prove the main statements for the design of the layers, we need Theorem 3.2 about these groups.

Consider the following groups preserving the subspaces $Cl_{p,q,r}^{\overline{k}}$, $k = 0, 1, 2, 3$, under the adjoint representation $\mathrm{ad}$ (5) and twisted adjoint representation $\check{\mathrm{ad}}$ (6) respectively:

$$\Gamma_{p,q,r}^{\overline{k}} := \{T \in Cl_{p,q,r}^{\times} : \ \mathrm{ad}_T(Cl_{p,q,r}^{\overline{k}}) \subseteq Cl_{p,q,r}^{\overline{k}}\}, \quad (16)$$

$$\check{\Gamma}_{p,q,r}^{\overline{k}} := \{T \in Cl_{p,q,r}^{\times} : \ \check{\mathrm{ad}}_T(Cl_{p,q,r}^{\overline{k}}) \subseteq Cl_{p,q,r}^{\overline{k}}\}, \quad (17)$$

where $\mathrm{ad}_T(Cl_{p,q,r}^{\overline{k}}) = T Cl_{p,q,r}^{\overline{k}} T^{-1}$ and $\check{\mathrm{ad}}_T(Cl_{p,q,r}^{\overline{k}}) = \widehat{T} Cl_{p,q,r}^{\overline{k}} T^{-1}$. Further, we show that these groups can be defined in an equivalent way, using only the norm functions $\psi$ and $\chi$ (11) applied in the theory of spin groups.

Let us introduce the families of Lie groups $Q_{p,q,r}^{\overline{1}}$, $Q_{p,q,r}^{\overline{2}}$, $Q_{p,q,r}^{\overline{3}}$, and $Q_{p,q,r}^{\overline{0}}$:

$$Q_{p,q,r}^{\overline{k}} := \{T \in Cl_{p,q,r}^{\times} : \ \widetilde{T}T, \widehat{\widetilde{T}}T \in Z_{p,q,r}^{k\times}\}, \quad (18)$$

$$Q_{p,q,r}^{\overline{0}} := \{T \in Cl_{p,q,r}^{\times} : \ \widetilde{T}T, \widehat{\widetilde{T}}T \in Z_{p,q,r}^{4\times}\}, \quad (19)$$

and $\check{Q}_{p,q,r}^{\overline{1}}$, $\check{Q}_{p,q,r}^{\overline{2}}$, $\check{Q}_{p,q,r}^{\overline{3}}$, $\check{Q}_{p,q,r}^{\overline{0}}$:

$$\check{Q}_{p,q,r}^{\overline{k}} := \{T \in Cl_{p,q,r}^{\times} : \ \widetilde{T}T, \widehat{\widetilde{T}}T \in \check{Z}_{p,q,r}^{k\times}\}, \quad (20)$$

$$\check{Q}_{p,q,r}^{\overline{0}} := \{T \in Cl_{p,q,r}^{\times} : \ \widetilde{T}T, \widehat{\widetilde{T}}T \in \langle Z_{p,q,r}^4 \rangle_{(0)}^{\times}\}, \quad (21)$$

with $k = 1, 2, 3$, where the sets

$$Z_{p,q,r}^m := \{X \in C\ell_{p,q,r} : \quad XV = VX, \quad \forall V \in C\ell_{p,q,r}^m\}$$

are the *centralizers* of the subspaces $C\ell_{p,q,r}^m$ in $C\ell_{p,q,r}$, and

$$\check{Z}_{p,q,r}^m := \{X \in C\ell_{p,q,r} : \quad \widehat{X}V = VX, \quad \forall V \in C\ell_{p,q,r}^m\}$$

are the *twisted centralizers* of $C\ell_{p,q,r}^m$ in $C\ell_{p,q,r}$. These sets are studied in detail in Filimoshina & Shirokov, 2024b. We provide explicit forms of $Z_{p,q,r}^m$ and $\check{Z}_{p,q,r}^m$, $m \leq 4$, in Remark D.1.

**Theorem 3.2.** *In degenerate and non-degenerate geometric algebras $C\ell_{p,q,r}$, we have for $k = 1, 2, 3$,*

$$\Gamma_{p,q,r}^{\overline{k}} = Q_{p,q,r}^{\overline{k}}, \quad \Gamma_{p,q,r}^{\overline{0}} = Q_{p,q,r}^{\overline{0}}, \qquad (22)$$

$$\check{\Gamma}_{p,q,r}^{\overline{1}} = \check{Q}_{p,q,r}^{\overline{1}} \subseteq \tilde{\Gamma}_{p,q,r}^{\overline{3}} = \check{Q}_{p,q,r}^{\overline{3}}, \qquad (23)$$

$$\check{\Gamma}_{p,q,r}^{\overline{2}} = \check{Q}_{p,q,r}^{\overline{2}}, \quad \check{\Gamma}_{p,q,r}^{\overline{0}} = \check{Q}_{p,q,r}^{\overline{0}}. \qquad (24)$$

*Moreover, for $m = 0, 1, 2, 3$,*

$$\Gamma_{p,q,r}^{\overline{1}} \subseteq \Gamma_{p,q,r}^{\overline{m}} \subseteq \Gamma_{p,q,r}^{\overline{0}}, \qquad (25)$$

$$\check{\Gamma}_{p,q,r}^{\overline{m}} \subseteq \Gamma_{p,q,r}^{\overline{0}}, \quad \check{\Gamma}_{p,q,r}^{\overline{1}} \subseteq \Gamma_{p,q,r}^{\overline{2}}. \qquad (26)$$

*Proof.* The proof relies on the relation between the preservation of the subspaces $C\ell_{p,q,r}^{\overline{k}}$, $k = 0, 1, 2, 3$, under $\tilde{\mathrm{ad}}$ and the values of the norm functions $\psi$ and $\chi$ (11). It is provided in Appendix F (see Theorems F.1, F.2 and Remark F.3). $\square$

The generalized Lipschitz groups $\tilde{\Gamma}_{p,q,r}^{\overline{k}}$ (13), $k = 0, 1, 2, 3$, are related to the groups $\Gamma_{p,q,r}^{\overline{k}}$ (16) and $\check{\Gamma}_{p,q,r}^{\overline{k}}$ (17):

$$\tilde{\Gamma}_{p,q,r}^{\overline{k}} = \begin{cases} \check{\Gamma}_{p,q,r}^{\overline{k}}, & k = 1, 3, \\ \Gamma_{p,q,r}^{\overline{k}}, & k = 0, 2, \end{cases} \qquad (27)$$

since $\tilde{\mathrm{ad}}_T(C\ell_{p,q,r}^{\overline{k}}) = \mathrm{ad}_T(C\ell_{p,q,r}^{\overline{k}})$ in the cases $k = 0, 2$ and $\tilde{\mathrm{ad}}_T(C\ell_{p,q,r}^{\overline{k}}) = \check{\mathrm{ad}}_T(C\ell_{p,q,r}^{\overline{k}})$ in the cases $k = 1, 3$.

As a corollary of Theorem 3.2 and (27), the generalized Lipschitz groups $\tilde{\Gamma}_{p,q,r}^{\overline{k}}$ (13) satisfy

$$\tilde{\Gamma}_{p,q,r}^{\overline{1}} = \check{Q}_{p,q,r}^{\overline{1}} \subseteq \tilde{\Gamma}_{p,q,r}^{\overline{2}} = Q_{p,q,r}^{\overline{2}} \subseteq \tilde{\Gamma}_{p,q,r}^{\overline{0}} = Q_{p,q,r}^{\overline{0}}, \quad (28)$$

$$\tilde{\Gamma}_{p,q,r}^{\overline{1}} \subseteq \tilde{\Gamma}_{p,q,r}^{\overline{3}} = \check{Q}_{p,q,r}^{\overline{3}}. \qquad (29)$$

Note that the group $\tilde{\Gamma}_{p,q,r}^{\overline{1}}$ can be regarded as the most significant among the generalized Lipschitz groups $\tilde{\Gamma}_{p,q,r}^{\overline{k}}$, $k = 0, 1, 2, 3$, because it contains the (ordinary) Lipschitz group as a subgroup and is itself a subgroup of all other generalized Lipschitz groups (see Theorem 3.1 and (28)-(29)). We prove that its elements are of a special form:

**Theorem 3.3.** *We have $\tilde{\Gamma}_{p,q,r}^{\overline{1}} \subseteq (C\ell_{p,q,r}^{(0)\times} \cup C\ell_{p,q,r}^{(1)\times})\Lambda_r^{\times}$.*
*Proof.* The proof is based on the equivalent definition of the group $(C\ell_{p,q,r}^{(0)\times} \cup C\ell_{p,q,r}^{(1)\times})\Lambda_r^{\times}$ and is provided in Theorem F.5. $\square$

The **key takeaways** from this section for constructing the generalized Lipschitz groups $\tilde{\Gamma}_{p,q,r}^{\overline{1}}$-equivariant layers are as follows: (1) the ordinary Lipschitz groups are subgroups of the generalized Lipschitz groups $\tilde{\Gamma}_{p,q,r}^{\overline{k}}$ (Theorem 3.1), (2) elements of the generalized Lipschitz groups $\tilde{\Gamma}_{p,q,r}^{\overline{1}}$ preserve not only the subspace $C\ell_{p,q,r}^{\overline{1}}$, but also all other subspaces $C\ell_{p,q,r}^{\overline{m}}$, $m = 0, 2, 3$, under the twisted adjoint representation $\tilde{\mathrm{ad}}$ (see (28)-(29)), and (3) elements of $\tilde{\Gamma}_{p,q,r}^{\overline{1}}$ have a special form (Theorem 3.3).

### 3.2. Pseudo-orthogonal Groups, Complex Orthogonal Groups, and Generalized Lipschitz Groups Equivariance

Let us consider the degenerate and non-degenerate pseudo-orthogonal group (in the real case $V = \mathbb{R}^{p,q,r}$) or complex orthogonal group (in the complex case $V = \mathbb{C}^{p+q,0,r}$) denoted by $\mathrm{O}(V, \mathfrak{q})$, which is the Lie group of all linear transformations of an $n$-dimensional vector space $V$ that leave invariant a quadratic form $\mathfrak{q}$ of signature $(p, q, r)$ if $V$ is real and $(p + q, 0, r)$ if $V$ is complex (see Appendix H):

$$\mathrm{O}(V, \mathfrak{q}) := \{\Phi : V \to V : \quad \Phi \text{ is linear, invertible,}$$
$$\mathfrak{q}(\Phi(v)) = \mathfrak{q}(v), \quad \forall v \in V\}. \qquad (30)$$

When considering both the real and complex cases, we refer to the group $\mathrm{O}(V, \mathfrak{q})$ as the orthogonal group. Consider the following subgroup of the orthogonal group, which leaves invariant the *radical subspace* $\Lambda_r^1 := C\ell_{0,0,r}^1$:

$$\mathrm{O}_{\Lambda_r^1}(V, \mathfrak{q}) := \{\Phi \in \mathrm{O}(V, q) : \quad \Phi|_{\Lambda_r^1} = \mathrm{id}_{\Lambda_r^1}\}. \qquad (31)$$

We have the following relation between the Lipschitz group $\tilde{\Gamma}_{p,q,r}^1$ and the corresponding restricted orthogonal group $\mathrm{O}_{\Lambda_r^1}(V, \mathfrak{q})$:

$$\tilde{\mathrm{ad}}^1 : \quad \tilde{\Gamma}_{p,q,r}^1 \Big/ \Lambda_r^{\times} \cong \mathrm{O}_{\Lambda_r^1}(V, \mathfrak{q}), \qquad (32)$$

where $\tilde{\mathrm{ad}}^1 : C\ell_{p,q,r}^{\times} \to \mathrm{Aut}(C\ell_{p,q,r})$ is $\tilde{\mathrm{ad}}$ restricted to $V$ (a real $\mathbb{R}^{p,q,r}$ or complex $\mathbb{C}^{p+q,0,r}$ vector space, which can be identified with the space of vectors $C\ell_{p,q,r}^1$, see page 3), i.e. it acts as $T \mapsto \tilde{\mathrm{ad}}_T^1 : C\ell_{p,q,r}^1 \to C\ell_{p,q,r}^1$ defined as $\tilde{\mathrm{ad}}_T^1(v) := \widehat{T}vT^{-1}$ for any vector $v \in C\ell_{p,q,r}^1$. From (32), we have that for any $T \in \tilde{\Gamma}_{p,q,r}^1$ there exists $\Phi \in \mathrm{O}_{\Lambda_r^1}(V, \mathfrak{q})$ such that $\tilde{\mathrm{ad}}_T^1 = \Phi$ and vice versa. We prove (32) in Theorem H.10. In the particular case $C\ell_{p,q}$, the statement (32) is well-known (see, for example, Benn & Tucker, 1987), and the similar statements are proved in Crumeyrolle, 1990; Ruhe et al., 2023.

The relation (32) implies that $\mathrm{O}_{\Lambda_r^1}(V, \mathfrak{q})$ acts on the whole $C\ell_{p,q,r}$ in a well-defined way (Ruhe et al., 2023). Namely, for an arbitrary element $x = \sum_i u_i v_{i,1} \cdots v_{i,k_i} \in C\ell_{p,q,r}$ with $v_{i,j} \in V$, $u_i \in \mathbb{F}$, for any $T \in \tilde{\Gamma}_{p,q,r}^1$ with the corresponding $\Phi = \tilde{\mathrm{ad}}_T^1$, we have

$$\tilde{\mathrm{ad}}_T(x) = \sum_i u_i \tilde{\mathrm{ad}}_T^1(v_{i,1}) \cdots \tilde{\mathrm{ad}}_T^1(v_{i,k_i}) \qquad (33)$$

$$= \sum_i u_i \Phi(v_{i,1}) \cdots \Phi(v_{i,k_i}) =: \Phi(x). \qquad (34)$$

The following theorem discusses the relation between $O_{\Lambda_r^1}(V, \mathfrak{q})$- and $\tilde{\Gamma}_{p,q,r}^{\overline{1}}$-equivariance.

**Theorem 3.4.** *If a mapping $f : Cl_{p,q,r} \to Cl_{p,q,r}$ is equivariant with respect to any group $H$ that contains the Lipschitz group $\tilde{\Gamma}_{p,q,r}^1$ as a subgroup, then $f$ is equivariant with respect to the corresponding orthogonal group. That is, if $\tilde{\mathrm{ad}}_T(f(x)) = f(\tilde{\mathrm{ad}}_T(x))$ for any $T \in H$ and $x \in Cl_{p,q,r}$, then $f(\Phi(x)) = \Phi(f(x))$ for any $\Phi \in O_{\Lambda_r^1}(V, \mathfrak{q})$ and $x \in Cl_{p,q,r}$, where $\Phi$ acts on $x$ in a sense* (34).

*Proof.* The proof relies on the direct relation between $\tilde{\Gamma}_{p,q,r}^1$- and $O_{\Lambda_r^1}(V, \mathfrak{q})$-equivariance (see Theorem H.11). $\qquad\square$

Since $\tilde{\Gamma}_{p,q,r}^1 \subseteq \tilde{\Gamma}_{p,q,r}^{\overline{1}}$ by Theorem 3.1, **all generalized Lipschitz group $\tilde{\Gamma}_{p,q,r}^{\overline{1}}$-equivariant mappings are restricted orthogonal group $O_{\Lambda_r^1}(V, \mathfrak{q})$-equivariant as well**.

### 3.3. Generalized and Ordinary Lipschitz Groups Equivariant Mappings

This section proves that several mappings are generalized Lipschitz groups $\tilde{\Gamma}_{p,q,r}^{\overline{1}}$-equivariant.

**Theorem 3.5.** *Let $T \in (Cl_{p,q,r}^{(0)\times} \cup Cl_{p,q,r}^{(1)\times})\Lambda_r^\times$ and $F \in \mathbb{F}[T_1, \ldots, T_l]$ be a polynomial in $l$ variables with coefficients in $\mathbb{F}$. Consider $l$ multivectors $x_1, \ldots, x_l \in Cl_{p,q,r}$. We have the following equivariance property with respect to the group $(Cl_{p,q,r}^{(0)\times} \cup Cl_{p,q,r}^{(1)\times})\Lambda_r^\times$:*

$$\tilde{\mathrm{ad}}_T(F(x_1, \ldots, x_l)) = F(\tilde{\mathrm{ad}}_T(x_1), \ldots, \tilde{\mathrm{ad}}_T(x_l)). \quad (35)$$

*Proof.* This fact follows directly from additivity and multiplicativity of $\tilde{\mathrm{ad}}_T$ for $T \in (Cl_{p,q,r}^{(0)\times} \cup Cl_{p,q,r}^{(1)\times})\Lambda_r^\times$, which are proved in Lemma 2.1. $\qquad\square$

Note that the statement that all polynomials are $\tilde{\Gamma}_{p,q}^1$-equivariant, which follows as a corollary of Theorem 3.5, is well-known (Ruhe et al., 2023).

**Theorem 3.6.** *For any $x \in Cl_{p,q,r}$ and $m = 0, 1, 2, 3$, we have the following equivariant property with respect to the generalized Lipschitz groups:*

$$\tilde{\mathrm{ad}}_T(\langle x \rangle_{\overline{m}}) = \langle \tilde{\mathrm{ad}}_T(x) \rangle_{\overline{m}}, \qquad \forall T \in \tilde{\Gamma}_{p,q,r}^{\overline{1}}. \quad (36)$$

*Proof.* Suppose $T \in \tilde{\Gamma}_{p,q,r}^{\overline{1}}$ and $x = x_{(0)} + x_{(1)} \in Cl_{p,q,r}$, where $x_{(0)} := \langle x \rangle_{(0)}$ and $x_{(1)} := \langle x \rangle_{(1)}$. For any $m = 0, 1, 2, 3$, we get

$$\langle \tilde{\mathrm{ad}}_T(x) \rangle_{\overline{m}} = \langle T x_{(0)} T^{-1} + \widehat{T} x_{(1)} T^{-1} \rangle_{\overline{m}} \quad (37)$$

$$= \langle \sum_{i=0,2} T \langle x \rangle_{\overline{i}} T^{-1} + \sum_{i=1,3} \widehat{T} \langle x \rangle_{\overline{i}} T^{-1} \rangle_{\overline{m}} \quad (38)$$

$$= \sum_{i=0,2} \langle T \langle x \rangle_{\overline{i}} T^{-1} \rangle_{\overline{m}} + \sum_{i=1,3} \langle \widehat{T} \langle x \rangle_{\overline{i}} T^{-1} \rangle_{\overline{m}} \quad (39)$$

$$= \begin{cases} \langle T \langle x \rangle_{\overline{m}} T^{-1} \rangle_{\overline{m}} = T \langle x \rangle_{\overline{m}} T^{-1}, & m \text{ is even,} \\ \langle \widehat{T} \langle x \rangle_{\overline{m}} T^{-1} \rangle_{\overline{m}} = \widehat{T} \langle x \rangle_{\overline{m}} T^{-1}, & m \text{ is odd,} \end{cases} \quad (40)$$

where in (39) we apply linearity of projection $\langle \rangle_{\overline{m}}$, and in (40), we use Theorem 3.2 and the notes below it. $\qquad\square$

In Theorem 3.7, we prove that several conjugation operations (1) in $Cl_{p,q,r}$ are generalized Lipschitz group $\tilde{\Gamma}_{p,q,r}^{\overline{1}}$-equivariant. Note that it does not hold for any conjugation operation. However, we prove that any conjugation operation is (ordinary) Lipschitz group $\tilde{\Gamma}_{p,q,r}^1$-equivariant in Theorem 3.8.

**Theorem 3.7.** *The grade involution, reversion, and Clifford conjugation are $\tilde{\Gamma}_{p,q,r}^{\overline{1}}$-equivariant:* $\tilde{\mathrm{ad}}_T(\widehat{x}) = \widehat{\tilde{\mathrm{ad}}_T(x)}$, $\tilde{\mathrm{ad}}_T(\widetilde{x}) = \widetilde{\tilde{\mathrm{ad}}_T(x)}$, $\tilde{\mathrm{ad}}_T(\widehat{\widetilde{x}}) = \widehat{\widetilde{\tilde{\mathrm{ad}}_T(x)}}$ *for any $T \in \tilde{\Gamma}_{p,q,r}^{\overline{1}}$ and $x \in Cl_{p,q,r}$.*

*Proof.* Since the grade involute, reversion, and Clifford conjugate of an element $x \in Cl_{p,q,r}$ are linear combinations of the projections onto the subspaces $Cl_{p,q,r}^{\overline{m}}$, $m = 0, 1, 2, 3$, (see Table 1) which are $\tilde{\Gamma}_{p,q,r}^{\overline{1}}$-equivariant, we get the statement by Lemma G.3 and Theorem 3.6. $\qquad\square$

**Theorem 3.8.** *Any conjugation operation in $Cl_{p,q,r}$ is Lipschitz group $\tilde{\Gamma}_{p,q,r}^1$-equivariant:*

$$\tilde{\mathrm{ad}}_T\Big(\sum_{k=0}^n \lambda_k \langle x \rangle_k\Big) = \sum_{k=0}^n \lambda_k \langle \tilde{\mathrm{ad}}_T(x) \rangle_k, \quad \lambda_k = \pm 1,$$

*for any $T \in \tilde{\Gamma}_{p,q,r}^1$ and $x \in Cl_{p,q,r}$.*

*Proof.* By definition (1), any conjugation operation is a linear combination of projections onto the subspaces $Cl_{p,q,r}^k$, $k = 0, \ldots, n$, which are $\tilde{\Gamma}_{p,q,r}^1$-equivariant (see Corollary 3.3 Ruhe et al., 2023): $\tilde{\mathrm{ad}}_T(\langle x \rangle_k) = \langle \tilde{\mathrm{ad}}_T(x) \rangle_k$, for any $T \in \tilde{\Gamma}_{p,q,r}^1$ and $x \in Cl_{p,q,r}$. Then, we get the statement by Lemma G.3. $\qquad\square$

**Theorem 3.9.** *The norm functions $\psi$ and $\chi$ (11) are $\tilde{\Gamma}_{p,q,r}^{\overline{1}}$-equivariant:*

$$\tilde{\mathrm{ad}}_T\big(\psi(x)\big) = \psi(\tilde{\mathrm{ad}}_T(x)), \quad \tilde{\mathrm{ad}}_T\big(\chi(x)\big) = \chi(\tilde{\mathrm{ad}}_T(x)) \quad (41)$$

*for any $T \in \tilde{\Gamma}_{p,q,r}^{\overline{1}}$ and $x \in Cl_{p,q,r}$.*

*Proof.* Since $\tilde{\Gamma}_{p,q,r}^{\overline{1}} \subseteq (Cl_{p,q,r}^{(0)\times} \cup Cl_{p,q,r}^{(1)\times})\Lambda_r^\times$ (Theorem 3.3), $\tilde{\mathrm{ad}}_T$ is multiplicative for any $T \in \tilde{\Gamma}_{p,q,r}^{\overline{1}}$ by Lemma 2.1. The mappings $\psi(x)$ and $\chi(x)$ are $\tilde{\Gamma}_{p,q,r}^{\overline{1}}$-equivariant by Lemma G.4, since they are products of the $\tilde{\Gamma}_{p,q,r}^{\overline{1}}$-equivariant mappings: $x \mapsto \widetilde{x}$, $x \mapsto \widehat{\widetilde{x}}$, and $x \mapsto x$ (Theorem 3.7). $\qquad\square$

## 4. Methodology

In this section, we describe the architecture of Generalized Lipschitz Groups Neural Networks layers. In this and the following sections, we consider the real geometric algebra $Cl(\mathbb{R}^{p,q,r})$. Code is available at https://github.com/katyafilimoshina/glgenn. We plan to continue developing this repository.

### 4.1. Geometric Algebra Data Embeddings

GLGENN, along with other GA-based equivariant neural networks such as CGENN (Ruhe et al., 2023), can be applied to any data representable as multivectors. For example, consider the following task in $\mathbb{R}^{0,3}$. We have data objects $x_1, \ldots, x_l$, where each $x_i$ contains a scalar (e.g. $a \in \mathbb{R}$, representing mass, temperature, or a one-hot-encoded feature,

etc.), a vector (e.g. $(b, c, d) \in \mathbb{R}^{0,3}$, where $b, c, d \in \mathbb{R}$, representing position, velocity, etc.), and a pseudoscalar (e.g., a signed volume value $f \in \mathbb{R}$). These components can be embedded in GA as $ae + be_1 + ce_2 + de_3 + fe_{123} \in Cl_{0,3}$ and processed by our layers. Additional grades beyond 0, 1, and 3 can be utilized as well.[2] The output can be obtained by projecting the resulting multivectors onto the appropriate subspace based on the task. E.g., projecting onto $Cl^0$ allows for scalar predictions (which will be both equivariant and invariant), or onto $Cl_{p,q,r}^1$ if vector predictions are required.

### 4.2. Conjugation Operations Layers

These layers are based on the concept of conjugation operations in $Cl_{p,q,r}$ (1). Suppose $x_1, \ldots, x_l \in Cl_{p,q,r}$ are input data multivectors, where $l$ is a number of input channels. For an input data multivector $x_{c_{in}} \in Cl_{p,q,r}$, $c_{in} = 1, \ldots, l$, a conjugation operations layer is constructed using

$$x_{c_{in}} \mapsto \sum_{k=0}^{n} \phi_{c_{in}k} \langle x_{c_{in}} \rangle_k, \quad \phi_{c_{in}k} = \pm 1, \quad (42)$$

where $\phi_{c_{in}k} \in \{-1, 1\}$ are optimizable coefficients. The conjugation operation layer is $\tilde{\Gamma}_{p,q,r}^1$-equivariant by Theorem 3.8 and contains $l(n + 1)$ optimizable parameters. We suggest the following method to make the optimization of conjugation layers with discrete parameters possible. We first apply a linear transformation of the projections of the input multivector as in (42) but with any parameters. Then we round them to $-1$ or 1 either by directly applying $\phi_{c_{in}k} \mapsto \text{sgn}(\phi_{c_{in}k})$ or by firstly applying the sigmoid function, then scaling the value to $[-1, 1]$, and applying sign function to the result. Conjugation operations, including $\widehat{\phantom{x}}$, $\widetilde{\phantom{x}}$, and $\widehat{\widetilde{\phantom{x}}}$, play a significant role in GAs and their applications (see Appendix C for a detailed discussion), making these layers highly promising for experiments.

### 4.3. $Cl_{p,q,r}^{\overline{k}}$-Linear Layers

A $Cl_{p,q,r}^{\overline{k}}$-linear layer is constructed using

$$\langle y_{c_{out}} \rangle_{\overline{k}} := \sum_{c_{in}=0}^{l} \phi_{c_{out}c_{in}\overline{k}} \langle x_{c_{in}} \rangle_{\overline{k}}, \quad (43)$$

where $y_{c_{out}} := \sum_{k=0}^{3} \langle y_{c_{out}} \rangle_{\overline{k}}$ is an output channel, $\phi_{c_{in}c_{out}\overline{k}} \in \mathbb{R}$ are optimizable coefficients, and $c_{in}$ and

$c_{out}$ are used to denote the number of the input and the output channels respectively. In other words, for any fixed $k = 0, 1, 2, 3$, the value $\langle y_{c_{out}} \rangle_{\overline{k}}$ is a linear combination of $\langle x_{c_{in}} \rangle_{\overline{k}}$, where $c_{in} = 1, \ldots, l$. One $Cl_{p,q,r}^{\overline{k}}$-linear layer is parametrized with $4lm$ optimizable coefficients, where $l$ and $m$ are the numbers of input and output channels respectively. Such layers are $\tilde{\Gamma}_{p,q,r}^1$-equivariant by Theorems 3.5 and 3.6.

### 4.4. $Cl_{p,q,r}^{\overline{k}}$-Geometric Product Layers

Now let us parameterize product interaction terms. We consider only the second-order interactions, because the higher-order interactions are indirectly modeled via multiple successive layers (Ruhe et al., 2023). A second-order interaction term for the pair of multivectors $x_1$ and $x_2$ has the form $\langle \langle x_1 \rangle_{\overline{i}} \langle x_2 \rangle_{\overline{j}} \rangle_{\overline{k}}$, where $i, j, k = 0, 1, 2, 3$. All the terms from $Cl_{p,q,r}^{\overline{k}}$ resulting from the interaction of $x_1$ and $x_2$ are parameterized as

$$P(x_1, x_2)^{\overline{k}} := \sum_{i=0}^{3} \sum_{j=0}^{3} \phi_{ijk} \langle \langle x_1 \rangle_{\overline{i}} \langle x_2 \rangle_{\overline{j}} \rangle_{\overline{k}}, \quad (44)$$

where $\phi_{ijk} \in \mathbb{R}$ are optimizable coefficients. $P(x_1, x_2)^{\overline{k}}$ is $\tilde{\Gamma}_{p,q,r}^1$-equivariant by Theorem 3.5 and Theorem 3.6. We get $4^3$ parameters $\phi_{ijk}$ for each geometric product of a pair of multivectors. Since we have $l^2$ possible pairs of multivectors for $l$ input channels, parameterizing such number of coefficients can be computationally expensive. Therefore, if a number of channels $l$ is large, let us compute geometric products for less than $l^2$ pairs of multivectors by an approach proposed in Ruhe et al., 2023. Firstly, we apply to $x_1, \ldots, x_l \in Cl_{p,q,r}$ a linear map to get $y_1, \ldots, y_l \in Cl_{p,q,r}$ and then compute the products of the pairs $x_i, y_i$ for $i = 1, \ldots, l$, which we denote by $z_1, \ldots, z_l$ respectively. Using such a trick, the terms that will be multiplied get learned. Now, we need to calculate only $l$ geometric products. We get $\langle z_{c_{out}} \rangle_{\overline{k}} := P(x_{c_{in}}, y_{c_{in}})^{\overline{k}}$, where $z_{c_{out}} := \sum_{k=0}^{3} \langle z_{c_{out}} \rangle_{\overline{k}}$ and $c_{in} = c_{out} = 1, \ldots, l$. In total, for a $Cl_{p,q,r}^{\overline{k}}$-geometric product layer, we get $4l^2 + 4^3 l$ parameters for $l$ input channels.

### 4.5. $Cl_{p,q,r}^{\overline{k}}$-Normalization Layers

For numerical stability of $Cl_{p,q,r}^{\overline{k}}$-geometric product layers, we apply normalization to the four projections $\langle x_{c_{in}} \rangle_{\overline{0}}$, $\langle x_{c_{in}} \rangle_{\overline{1}}$, $\langle x_{c_{in}} \rangle_{\overline{2}}$, and $\langle x_{c_{in}} \rangle_{\overline{3}}$ for each multivector $x_{c_{in}}$, $c_{in} = 1, \ldots, l$ before geometric product:

$$\langle x_{c_{in}} \rangle_{\overline{k}} \mapsto \frac{\langle x_{c_{in}} \rangle_{\overline{k}}}{\sigma(\phi_{c_{in}\overline{k}})(\langle \widetilde{\langle x_{c_{in}} \rangle_{\overline{k}}} \langle x_{c_{in}} \rangle_{\overline{k}} \rangle_0 - 1) + 1}, \quad (45)$$

where $\sigma(x) := \frac{1}{1+e^{-x}} \in (0, 1)$ is the logistic sigmoid function and $\phi_{c_{in}\overline{m}} \in \mathbb{R}$ are optimizable parameters. The similar approach is applied in Ruhe et al., 2023. Note that $\langle \widetilde{\langle x_{c_{in}} \rangle_{\overline{k}}} \langle x_{c_{in}} \rangle_{\overline{k}} \rangle_0$ is $\tilde{\Gamma}_{p,q,r}^1$-equivariant by Theorems 3.6 and 3.9. Each normalization layer contains $4l$ optimizable parameters, where $l$ is the number of input channels.

---

[2]Note that in the case of $Cl_{0,3}$, the equivariant model proposed in this paper has some connection with equivariant models based on irreducible representations (irreps) of O(3) (Han et al., 2022). Generally speaking, these are two different approaches: irreps of O(3) follow the tensor approach, whereas the GAs operate with multivectors. However, there are connections between the two approaches: scalars, vectors, bivectors, and trivectors in GA are equivalent to $l = 0$ irreps with even parity, $l = 1$ irreps with odd parity, $l = 1$ irreps with even parity, and $l = 0$ irreps with odd parity, respectively. The dimensions are the same (1, 3, 3, and 1, respectively), the behavior under rotations/reflections of the coordinate system matches, and the coupling operations (such as scalar multiplication, vector scalar and cross products in various cases) are also the same. Further study of this connection deserves a separate study and is not the topic of the current work, but seems important for building bridges between the two communities.

# 5. Experiments

We demonstrate that GLGENN either outperforms or matches the performance of other equivariant models, while using significantly fewer optimizable parameters. For CGENN (Ruhe et al., 2023), the training setups are aligned with its public code release. For other models, we use the loss values from the corresponding code repository (Finzi et al., 2021). All experimental details and discussions can be found in Appendix I and our code repository. For simplicity, the current examples focus on the case of the non-degenerate GA $C\ell_{p,q}$. Experiments involving $C\ell_{p,q,r}$ will be explored in future research. We construct our models to closely resemble the CGENN architecture, replacing the $C\ell_{p,q}^k$-linear, $C\ell_{p,q}^k$-geometric product, and $C\ell_{p,q}^k$-normalization layers from Ruhe et al., 2023 with the same number of GLGENN's $C\ell_{p,q}^{\overline{k}}$ counterparts (see Section 4).

### O$(5,0)$-Regression Task

We consider an O$(5,0)$-invariant regression task (Finzi et al., 2021) to estimate the function $\sin(\|x_1\|) - \|x_2\|^3/2 + \frac{x_1^T x_2}{\|x_1\|\|x_2\|}$, where $x_1, x_2 \in \mathbb{R}^{5,0}$ are vectors sampled from a standard Gaussian distribution. The loss function used is Mean squared error (MSE). We evaluate the performance using four different training dataset sizes and compare against the ordinary MLP, MLP with augmentations, the O$(5,0)$- and SO$(5,0)$-equivariant MLP architectures proposed in (Finzi et al., 2021), and with CGENN (Ruhe et al., 2023). The results, presented in Tables 2 and 6 and Figure 2 (left), demonstrate that GLGENN achieves performance on a par with CGENN, while significantly outperforming the other models. Moreover, GLGENN attains these results with fewer parameters and reduced training time compared to CGENN. In this experiment, CGENN has approximately 1.8K parameters associated with GA layers, while GLGENN has around 0.6K parameters. Other models have approximately the same number of parameters as CGENN. In Tables 2 and 6, MSE for CGENN and GLGENN are averaged over 5 runs. MSE for other models are averaged over 3 runs (from Finzi et al., 2021). Number of iterations is the same for all algorithms. Note that when CGENN has the same size as GLGENN (with $\approx 0.6$K GA-associated parameters), we get the similar results (Table 10).

### O$(5,0)$- and O$(7,0)$-Convex Hull Volume Estimation

In these experiments, the task is to estimate the volume of a convex hull generated by $K$ points in $\mathbb{R}^{5,0}$ and $\mathbb{R}^{7,0}$ respectively. We first consider the setup with $K = 16$ points, as in

*Table 2.* MSE ($\downarrow$) on the O$(5,0)$-Regression Experiment

| MODEL | # OF TRAINING SAMPLES | | | |
|---|---|---|---|---|
| | $3 \cdot 10^1$ | $3 \cdot 10^2$ | $3 \cdot 10^3$ | $3 \cdot 10^4$ |
| **GLGENN** | 0.1055 | 0.0020 | 0.0031 | 0.0011 |
| MLP | 28.1011 | 0.2482 | 0.0623 | 0.0622 |
| MLP+AUG | 0.4758 | 0.0936 | 0.0889 | 0.0672 |
| EMLP-O(5) | 0.152 | 0.0344 | 0.0310 | 0.0273 |
| EMLP-SO(5) | 0.1102 | 0.0384 | 0.032 | 0.0279 |
| CGENN | 0.0791 | 0.0089 | 0.0012 | 0.0003 |

Ruhe et al., 2023; Liu et al., 2024. The average convex hull volume in a training dataset is $\approx 11.4$. The results, which are presented in Figure 2 (middle) and Table 3, demonstrate that GLGENN either outperform or match CGENN, which itself outperforms ordinary MLP, Geometric Vector Perceptrons (Jing et al., 2021), and Vector Neurons (Deng et al., 2021). In Table 3, MSE for $K = 16$ are averaged over 5 runs, number of iterations is the same for both algorithms. Notably, as highlighted Ruhe et al., 2023, CGENN tends to overfit when trained on small datasets. In contrast, GLGENN demonstrate a reduced tendency to overfitting. The reason might be that GLGENN achieve these superior results with more than twice the parameter efficiency, using significantly fewer parameters than CGENN. Note that when CGENN is scaled down to approximately the same size as GLGENN (around 25K trainable parameters), its performance deteriorates compared to its original configuration (see Table 9). In the case $n = 7$, we get similar results (see Appendix I).

Following the suggestion of one of the anonymous reviewers, we further increase the difficulty and real-world relevance of the experiment by estimating the convex hull volumes of $K = 256$ and $512$ points in $\mathbb{R}^{5,0}$. In this setup, the average convex hull volumes in the training datasets are $\approx 430.3$ and $694$ respectively. GLGENN consistently outperforms the best-performing CGENN architecture, with the results summarized in Table 3. Due to its significantly smaller number of trainable parameters, GLGENN also require less training time (see Table 7 in Appendix I).

**Combining GLGENN with Typical Activation Functions**

Standard activation functions (e.g. SiLU or ReLU) can be applied to elements of $C\ell^0$ (scalars) without breaking equivariance. For simple tasks, the best performance is achieved by combining GLGENN (applied to all grade subspaces) with a standard neural network, such as MLP, acting on elements of $C\ell^0$. E.g., in the O$(5,0)$-Regression Task: (1) a standalone MLP with three ReLU-activated layers performs poorly; (2) GLGENN alone performs reasonably well but converges slower than (3) the combination of GLGENN (to all grades) + MLP (to scalars). With 300 training samples, this improvement is shown in Figure 3 and Table 11. The main limitation is that applying nonlinearities only to certain subspaces (e.g., scalars) prevents interaction between different grades (e.g., vectors and bivectors), effectively isolating them. In contrast, the nonlinearities introduced via geometric product layers inherently mix all grades, creating strong interactions.

### O$(5,0)$-$N$-Body Experiment

In this benchmarking equivariant experiment (Ruhe et al., 2023; Brehmer et al., 2023), we consider a system of $N = 5$ charged particles (bodies) with given masses, initial positions, and velocities. The objective is to predict the final positions of the bodies after the system evolves under

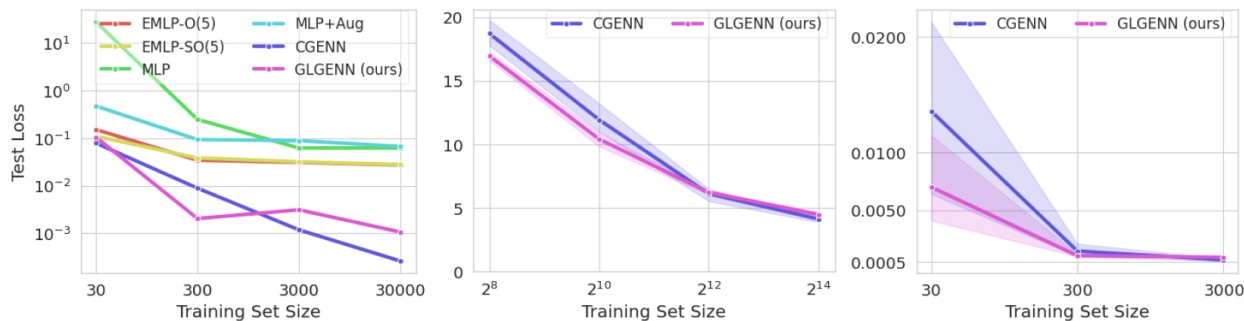

*Figure 2.* **Left:** O(5, 0)-Regression. **Middle:** O(5, 0)-Convex Hull, 16 points. The shaded regions depict 95% confidence intervals taken over 5 runs. **Right:** O(5, 0)-$N$-Body. The shaded regions depict 95% confidence intervals taken over 3 runs.

*Table 3.* MSE ($\downarrow$) on the O(5, 0)-Convex Hull Experiment ($K$ points). The average convex hull volume in the training dataset for $K = 16$, 256, 512 points is $\approx 11.4$, $\approx 430.3$, and $\approx 694$, respectively. The number of trainable parameters is as follows: for $K = 16$, GLGENN 24.1K vs. CGENN 58.8K; for $K = 256$, GLGENN 791K vs. CGENN 1.72M; for $K = 512$, GLGENN 922K vs. CGENN 1.75M.

| K | 16 | | | | | | | 256 | | 512 | |
|---|---|---|---|---|---|---|---|---|---|---|---|
| **MODEL** | **# TRAIN SAMPLES** | | | | | | | **# TRAIN SAMPLES** | | **# TRAIN SAMPLES** | |
| | $2^8$ | $2^{10}$ | $2^{12}$ | $2^{14}$ | $2^{15}$ | $2^{16}$ | $2^{17}$ | $2^{10}$ | $2^{14}$ | $2^{10}$ | $2^{14}$ |
| **GLGENN** | 16.94 | 10.40 | 6.2 | 4.46 | 3.62 | 3.04 | 2.61 | 2908.16 | 2918.09 | 8538.86 | 4872.39 |
| CGENN | 18.71 | 11.93 | 6.1 | 4.11 | 3.23 | 2.52 | 2.08 | 5176.7 | 3384.62 | 14727.6 | 7212.44 |
| GAP | −1.77 | −1.53 | 0.1 | 0.35 | 0.39 | 0.52 | 0.53 | −2268.54 | −466.53 | −6188.74 | −2340.05 |

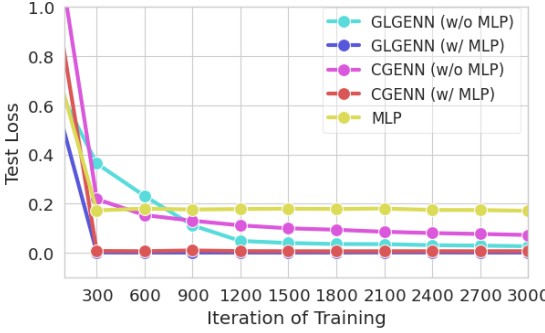

*Figure 3.* Combination of MLP with GLGENN and CGENN in O(5, 0)-Regression.

Newtonian gravity for 1000 Euler integration steps. Unlike standard low-dimensional setups, we simulate the system in $\mathbb{R}^{5,0}$ and embed the data into $C\ell_{5,0}$. We construct a graph neural network (GNN) based on the message-passing paradigm (Gilmer et al., 2017), where bodies are treated as nodes in a graph, and their pairwise interactions are modeled as edges. For each edge, we compute a message and then aggregate at each node to update the particle states. The message and update networks are equivariant GLGENN. We compare against CGENN, which itself outperforms several state-of-the-art method (see Appendix I). To ensure a fair comparison, we use the best-performing CGENN architecture and then replace its layers with analogous GLGENN counterparts to obtain the GLGENN architecture, which automatically has two times fewer trainable parameters. The results are presented in Table 12 and Figure 2 (right).

## 6. Conclusions

We introduce a new equivariant neural networks architecture based on geometric algebras (GAs), called Gen-

eralized Lipschitz Groups Equivariant Neural Networks (GLGENN). These networks are equivariant with respect to pseudo-orthogonal transformations of a vector space with any non-degenerate or degenerate bilinear form. GLGENN are parameter-light, since they operate in a unified manner across four fundamental subpaces of GAs defined by the grade involution and reversion. GLGENN strike a balance between the expressiveness of Clifford Group Equivariant Neural Networks (CGENN) and parameter efficiency by respecting the fundamental algebraic structures of GAs[3].

To develop GLGENN layers, we introduce and study the notion of generalized Lipschitz groups in GAs. We prove that the equivariance of a mapping with respect to these groups implies equivariance with respect to pseudo-orthogonal groups. As a result, GLGENN are equivariant with respect to a larger group than the pseudo-orthogonal group. However, empirically, we find that these networks are sufficiently expressive for tasks that require pseudo-orthogonal group equivariance. Experimental results demonstrate that GLGENN either outperform or match state-of-the-art models, while having significantly fewer trainable parameters and lower training time. Future research will focus on applying GLGENN to more complex real-world data experiments. Given their lower training time and promising results, we are optimistic about their potential.

---

[3]GLGENN goes beyond CGENN in the following: (1) Parameter-sharing approach for GA-based neural networks is presented for the first time. (2) New layers with parameter sharing are proposed, theoretical justification is provided (Section 4). (3) General idea is proposed that if we need orthogonal groups equivariance, then we may search for broader groups equivariance, such as new generalized Lipschitz groups, and get reasonable results (Theorem 3.4 and experiments).

## Acknowledgements

The article was prepared within the framework of the project 'Mirror Laboratories' HSE University ('Quaternions, geometric algebras and applications').

The authors are grateful to the four anonymous reviewers for their careful reading of the paper and helpful comments on how to improve the presentation.

## Impact Statement

The broader impact of this work is in improvement of the efficiency and effectiveness of equivariant neural networks. The significance of these networks is substantial, especially in the field of natural sciences, where the tasks inherently involve equivariance with respect to pseudo-orthogonal transformations. Parameter-light equivariant architectures reduce the risk of model's overfitting in case of small training datasets, which is a common scenario in the natural science field. Efficient equivariant neural networks have great potential to stimulate breakthroughs and innovations in modeling and simulations of physical systems, robotics, material science molecular biology, geoscience, etc.

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

## Brief Summary of Appendix

In Appendix A, we provide an overview of notation that we use throughout the paper.

Appendix B provides an example of the main theoretical constructions considered in Section 2 using the geometric algebra $C\ell_{1,3}$ of the Minkowski space $\mathbb{R}^{1,3}$.

Appendix C discusses the significance of grade involution and reversion in geometric algebras (GAs) $C\ell_{p,q,r}$. We show that these operations are in some sense unique in geometric algebras. This paragraph serves as motivation for considering the four subspaces of geometric algebras determined by the grade involution and the reversion. Our proposed architecture GLGENN actively operates with these subspaces in its structure.

In Appendix D, we consider the notion of centralizers and twisted centralizers of the subspaces of fixed grades and subspaces determined by the grade involution and reversion. We write out explicit forms of these sets. The centralizers and twisted centralizers play a crucial role in our main theoretical results on generalized Lipschitz groups.

Appendix E considers the (ordinary) Lipschitz groups, spin groups, and adjoint and twisted adjoint representations in arbitrary degenerate and non-degenerate geometric algebras. We prove several properties of these representations and find their kernels.

Appendix F presents and proves the key theoretical results necessary to constructing GLGENN. Specifically, we introduce and study generalized Lipschitz groups in arbitrary degenerate or non-degenerate geometric algebra. These groups preserve the subspaces determined by the grade involution and reversion under the twisted adjoint representation. We prove that these groups can be defined equivalently, using only the norm functions, which are applied in the theory of spin groups. This results allows us to prove that the (ordinary) Lipschitz groups are subgroups of the generalized Lipschitz groups.

Appendix G proves several properties of equivariant mappings, which are applied in Section 4.

Appendix H considers the relation between the well-known degenerate and non-degenerate pseudo-orthogonal (or complex orthogonal) groups and generalized Lipschitz groups. The obtained results enable us to prove that equivariance of a mapping with respect to the generalized Lipschitz groups implies its equivariance with respect to the corresponding pseudo-orthogonal group. This result is crucial for the construction of GLGENN, which are equivariant with respect to pseudo-orthogonal transformations.

Appendix I provides experimental details and discussions.

## A. Notation

In Table 4, we provide an overview of notation used throughout the paper. We write out the notation, its meaning, and the place where it is mentioned for the first time.

## B. Example of Geometric Algebra $C\ell_{1,3}$ of Minkowski Space $\mathbb{R}^{1,3}$

In this section, we provide an example of the key theoretical concepts and constructions considered in Section 2. Let us consider the Minkowski space $V = \mathbb{R}^{1,3}$ equipped with the Minkowski metric $\eta = \mathrm{diag}(1, -1, -1, -1)$. In this case, we have $p = 1$, $q = 3$, $r = 0$, and $n = p + q = 4$. The corresponding geometric (Clifford) algebra is $C\ell_{1,3} = C\ell(\mathbb{R}^{1,3})$, which is the real non-degenerate $2^4 = 16$-dimensional algebra with the identity element $e$ and generators $e_1, e_2, e_3, e_4$ that satisfy:

$$e_a e_b + e_b e_a = 2\eta_{ab} e, \quad a, b = 1, \ldots, 4, \tag{46}$$

which implies that the distinct generators anticommute, the first generator squared gives $+e$, and the other three generators squared give $-e$:

$$e_a e_b = -e_b e_a, \quad a, b = 1, \ldots 4, \quad a \neq b; \quad (e_1)^2 = +e, \quad (e_2)^2 = (e_3)^2 = (e_4)^2 = -e. \tag{47}$$

The basis elements of the algebra $C\ell_{1,3}$ are constructed as ordered products of distinct generators, such as $e_{123} = e_1 e_2 e_3$ and, in general, $e_{a_1 \ldots a_k} = e_{a_1} \cdots e_{a_k}$ for $a_1, \ldots, a_k \in \{1, 2, 3, 4\}$ and $a_1 < \cdots < a_k$. In the case of $C\ell_{1,3}$, we have five non-trivial subspaces of fixed grades:

$$C\ell_{1,3}^0 = \mathrm{span}\{e\} \equiv \mathbb{R}, \quad C\ell_{1,3}^1 = \mathrm{span}\{e_1, e_2, e_3, e_4\} \equiv V, \quad C\ell_{1,3}^2 = \mathrm{span}\{e_{12}, e_{13}, e_{14}, e_{23}, e_{24}, e_{34}\}, \tag{48}$$

*Table 4.* Summary of notation.

| Notation | Meaning | First mention |
|---|---|---|
| $C\ell_{p,q,r}$ | (Clifford) Geometric algebra (GA) over the real $\mathbb{R}^{p,q,r}$ or complex $\mathbb{C}^{p+q,0,r}$ vector space | Page 3 |
| $C\ell_{p,q}$ | Non-degenerate geometric algebra $C\ell_{p,q,0}$ | Page 3 |
| $\Lambda_r$ | Grassmann subalgebra $C\ell_{0,0,r}$ of $C\ell_{p,q,r}$ | Page 3 |
| $V$ | Real $\mathbb{R}^{p,q,r}$ or complex $\mathbb{C}^{p+q,0,r}$ vector space | Page 3 |
| $\mathbb{F}$ | Field of real or complex numbers in the cases $\mathbb{R}^{p,q,r}$ and $\mathbb{C}^{p+q,0,r}$ respectively | Page 3 |
| $\eta$ | Matrix of a bilinear form $V \times V \to \mathbb{F}$ | Page 3 |
| $\mathfrak{b}$ | Symmetric bilinear form, $\mathfrak{b} : V \times V \to \mathbb{F}$ | Page 3 |
| $\mathfrak{q}$ | Quadratic form, $\mathfrak{q} : V \to \mathbb{F}$ | Page 3 |
| $e$ | Identity element | Page 3 |
| $e_1, \ldots, e_n$ | Generators of $C\ell_{p,q,r}$ | Page 3 |
| $C\ell^0$ | Subspace of grade 0 | Page 3 |
| $C\ell_{p,q,r}^k$ | Subspace of grade $k$, where $k = 1, \ldots, n$ | Page 3 |
| $\{C\ell_{p,q}^k \Lambda_r^l\}$ | Subspace of $C\ell_{p,q,r}$ spanned by the elements of the form $ab$, where $a \in C\ell_{p,q}^k$ and $b \in \Lambda_r^l$ | Page 17 |
| $\langle \rangle_k$ | Projection onto the subspace of grade $k$ | Page 3 |
| $\widehat{U}$ | Grade involute of $U \in C\ell_{p,q,r}$ | Formula (2) |
| $\widetilde{U}$ | Reversion of $U \in C\ell_{p,q,r}$ | Formula (2) |
| $\widehat{\widetilde{U}}$ | Clifford conjugate of $U \in C\ell_{p,q,r}$ | Page 3 |
| $C\ell_{p,q,r}^{(0)}, C\ell_{p,q,r}^{(1)}$ | Even and odd subspaces of $C\ell_{p,q,r}$ | Formula (3) |
| $(C\ell_{p,q,r}^{(0)\times} \cup C\ell_{p,q,r}^{(1)\times})\Lambda_r^\times$ | Product of two groups: $\{ab \mid a \in C\ell_{p,q,r}^{(0)\times} \cup C\ell_{p,q,r}^{(1)\times},\ b \in \Lambda_r^\times\}$ | Lemma 2.1 |
| $\langle \rangle_{(l)}$ | Projection onto the subspace $C\ell_{p,q,r}^{(l)}$, where $l = 0, 1$ | Page 3 |
| $C\ell_{p,q,r}^{\overline{m}}$ | Subspaces determined by grade involution and reversion | Formula (4) |
| $\langle \rangle_{\overline{m}}$ | Projection onto the subspace $C\ell_{p,q,r}^{\overline{m}}$, where $m = 0, 1, 2, 3$ | Page 3 |
| $H^\times$ | Subset of all invertible elements of a set $H$ | Page 3 |
| $\mathrm{ad}$ | Adjoint representation $\mathrm{ad}_T(U) = TUT^{-1}$ | Formula (5) |
| $\check{\mathrm{ad}}$ | Twisted adjoint representation $\check{\mathrm{ad}}_T(U) = \widehat{T}UT^{-1}$ | Formula (6) |
| $\tilde{\mathrm{ad}}$ | Twisted adjoint representation $\tilde{\mathrm{ad}}_T(U) = T\langle U\rangle_{(0)}T^{-1} + \widehat{T}\langle U\rangle_{(1)}T^{-1}$ | Formula (7) |
| $\tilde{\mathrm{ad}}^1$ | $\tilde{\mathrm{ad}}$ restricted to $C\ell_{p,q,r}^1$ | Page 5 |
| $\tilde{\Gamma}_{p,q,r}^1$ | Ordinary Lipschitz groups | Formula (10) |
| $\psi, \chi$ | Norm functions of $C\ell_{p,q,r}$ elements | Formula (11) |
| $\tilde{\Gamma}_{p,q,r}^{\overline{k}}$ | Generalized Lipschitz groups, $k = 0, 1, 2, 3$ | Formula (13) |
| $\Gamma_{p,q,r}^{\overline{k}} = Q_{p,q,r}^{\overline{k}}$, | Groups preserving the subspaces $C\ell_{p,q,r}^{\overline{k}}$, $k = 0, 1, 2, 3$, under $\mathrm{ad}$ | Formulas (16) and (18)–(19) |
| $\check{\Gamma}_{p,q,r}^{\overline{k}} = \check{Q}_{p,q,r}^{\overline{k}}$ | Groups preserving the subspaces $C\ell_{p,q,r}^{\overline{k}}$, $k = 0, 1, 2, 3$, under $\check{\mathrm{ad}}$ | Formulas (17) and (20)–(21) |
| $Z_{p,q,r}$ | Center of $C\ell_{p,q,r}$ | Formula (70) |
| $Z_{p,q,r}^k, \check{Z}_{p,q,r}^k$ | Centralizers and twisted centralizers respectively of $C\ell_{p,q,r}^k$, $k = 0, \ldots, n$ | Page 4 |
| $Z_{p,q,r}^{\overline{k}}, \check{Z}_{p,q,r}^{\overline{k}}$ | Centralizers and twisted centralizers respectively of $C\ell_{p,q,r}^{\overline{k}}$, $k = 0, 1, 2, 3$ | Formulas (86)–(87) |
| $\mathrm{GL}(n, \mathbb{F})$ | General linear group acting on an $n$-dimensional vector space over $\mathbb{F}$ | Formula (30) |
| $\mathrm{O}(V, \mathfrak{q})$ | Orthogonal group | Formula (30) |
| $\mathrm{O}_{\Lambda_r^1}(V, \mathfrak{q})$ | Subgroup of $\mathrm{O}(V, \mathfrak{q})$ that leaves invariant $\Lambda_r^1$ | Formula (31) |

$$C\ell_{1,3}^3 = \mathrm{span}\{e_{123}, e_{124}, e_{134}, e_{234}\}, \quad C\ell_{1,3}^4 = \mathrm{span}\{e_{1234}\}, \tag{49}$$

which we call the subspaces of scalars, vectors, bivectors, trivectors, and 4-vectors respectively. An arbitrary element

$U \in C\ell_{1,3}$ can be written as

$$U = ue + \sum_{a=1,\ldots,4} u_a e_a + \sum_{\substack{a,b=1,\ldots,4, \\ a<b}} u_{ab} e_{ab} + \sum_{\substack{a,b,c=1,\ldots,4, \\ a<b<c}} u_{abc} e_{abc} + u_{1234} e_{1234}, \qquad u, \ldots, u_{1234} \in \mathbb{R}. \qquad (50)$$

The even $C\ell_{1,3}^{(0)}$ and odd $C\ell_{1,3}^{(1)}$ subspaces are defined as

$$C\ell_{1,3}^{(0)} = C\ell_{1,3}^0 \oplus C\ell_{1,3}^2 \oplus C\ell_{1,3}^4, \qquad C\ell_{1,3}^{(1)} = C\ell_{1,3}^1 \oplus C\ell_{1,3}^3. \qquad (51)$$

In $C\ell_{1,3}$, the four subspaces determined by the grade involution $\widehat{\phantom{x}}$ and the reversion $\widetilde{\phantom{x}}$ (the subspaces of quaternion types) are defined as

$$C\ell_{1,3}^{\bar{0}} = C\ell_{1,3}^0 \oplus C\ell_{1,3}^4, \quad C\ell_{1,3}^{\bar{1}} = C\ell_{1,3}^1, \quad C\ell_{1,3}^{\bar{2}} = C\ell_{1,3}^2, \quad C\ell_{1,3}^{\bar{3}} = C\ell_{1,3}^3 \qquad (52)$$

and are very similar to the subspaces of fixed grades because of small $n = 4$.

Let us consider the following example on how the projections $\langle \rangle_k$ onto the subspaces of fixed grades $C\ell_{1,3}^k$, $k = 0, \ldots, 4$, the projections $\langle \rangle_{(l)}$ onto the even and odd subspaces $C\ell_{1,3}^{(l)}$, $l = 0, 1$, and the projections $\langle \rangle_{\overline{m}}$ onto $C\ell_{1,3}^{\overline{m}}$, $m = 0, 1, 2, 3$ act on the element $W = e + 2e_1 + 3e_2 + 4e_{234} + 5e_{1234}$:

$$\langle W \rangle_0 = e, \quad \langle W \rangle_1 = \langle W \rangle_{\bar{1}} = 2e_1 + 3e_2, \quad \langle W \rangle_2 = \langle W \rangle_{\bar{2}} = 0, \quad \langle W \rangle_3 = \langle W \rangle_{\bar{3}} = 4e_{234}, \quad \langle W \rangle_4 = 5e_{1234}, \quad (53)$$

$$\langle W \rangle_{\bar{0}} = \langle W \rangle_{(0)} = e + 5e_{1234}, \quad \langle W \rangle_{(1)} = 2e_1 + 3e_2 + 4e_{234}. \qquad (54)$$

For an arbitrary $U \in C\ell_{1,3}$ (50), we have

$$\widehat{U} = \langle U \rangle_{(0)} - \langle U \rangle_{(1)}, \qquad \widetilde{U} = \langle U \rangle_{\bar{0}} + \langle U \rangle_{\bar{1}} - \langle U \rangle_{\bar{2}} - \langle U \rangle_{\bar{3}}, \qquad \widehat{\widetilde{U}} = \langle U \rangle_{\bar{0}} - \langle U \rangle_{\bar{1}} - \langle U \rangle_{\bar{2}} + \langle U \rangle_{\bar{3}}. \qquad (55)$$

Now let us consider the example on how the adjoint representation $\mathrm{ad}$ (5) and twisted adjoint representations $\check{\mathrm{ad}}$ (6) and $\widetilde{\mathrm{ad}}$ (7) act. Let us consider the element $T = e_1 + e_{123} \in C\ell_{1,3}^\times$, which is invertible because $\frac{1}{2}(e_1 + e_{123})(e_1 - e_{123}) = e$. Suppose $U = e + e_4 \in C\ell_{1,3}$. We have

$$\mathrm{ad}_T(U) = TUT^{-1} = T(e + e_4)T^{-1} = TT^{-1} + Te_4T^{-1} = e + \frac{1}{2}(e_1 + e_{123})e_4(e_1 - e_{123}) \qquad (56)$$

$$= e + \frac{1}{2}(e_1 + e_{123})(-e_1 + e_{123})e_4 = e - \frac{1}{2}(e_1 + e_{123})(e_1 - e_{123})e_4 = e - e_4, \qquad (57)$$

$$\check{\mathrm{ad}}_T(U) = \widehat{T}UT^{-1} = -TUT^{-1} = -e + e_4, \qquad (58)$$

$$\widetilde{\mathrm{ad}}_T(U) = T\langle U \rangle_{(0)}T^{-1} + \widehat{T}\langle U \rangle_{(1)}T^{-1} = TeT^{-1} - Te_4T^{-1} = e + e_4. \qquad (59)$$

Note that the results are different for different operations $\mathrm{ad}$, $\check{\mathrm{ad}}$, $\widetilde{\mathrm{ad}}$.

The Lipschitz group in the case $C\ell_{1,3}$ has the form

$$\tilde{\Gamma}_{1,3}^1 = \{T \in C\ell_{1,3}^\times : \quad \widehat{T}C\ell_{1,3}^1 T^{-1} \subseteq C\ell_{1,3}^1\}. \qquad (60)$$

A simple example of an element of $\tilde{\Gamma}_{1,3}^1$ is any basis element $e_{a_1 \ldots a_k}$, since $e_{a_1 \ldots a_k}^{-1} = \pm e_{a_1 \ldots a_k}$, where the sign depends on $a_1, \ldots, a_k$ and $k$, and $\widehat{e_{a_1 \ldots a_k}} e_a e_{a_1 \ldots a_k}^{-1} = \pm e_a \in C\ell_{1,3}^1$ for any $a = 1, \ldots, 4$. A slightly more complicated example of the Lipschitz group $\tilde{\Gamma}_{1,3}^1$ element is $T = e_1 + e_{123} \in C\ell_{1,3}^\times$ considered in the paragraph above. We have:

$$\widehat{T}e_1T^{-1} = -\frac{1}{2}(e_1 + e_{123})e_1(e_1 - e_{123}) = -e_1, \quad \widehat{T}e_2T^{-1} = e_3, \quad \widehat{T}e_3T^{-1} = e_2, \quad \widehat{T}e_4T^{-1} = e_4. \qquad (61)$$

Therefore, $\widehat{T}e_aT^{-1} \in C\ell_{1,3}^1$ for $a = 1, \ldots, 4$; thus, $T \in \tilde{\Gamma}_{1,3}^1$.

## C. Grade Involution and Reversion as Fundamental Operations in $C\ell_{p,q,r}$

This work introduces the neural networks that are equivariant with respect to the groups $\tilde{\Gamma}^{\bar{1}}_{p,q,r}$ (27) preserving the subspace $C\ell^{\bar{1}}_{p,q,r}$ (4) determined by the grade involution and reversion under the twisted adjoint representation $\tilde{\mathrm{ad}}$. These operations can be considered as fundamental operations in geometric algebras $C\ell_{p,q,r}$. In this section, we discuss their significance.

In Subsection 2.1, the grade involute and reversion of an arbitrary $U \in C\ell_{p,q,r}$ are defined in the following way respectively:

$$\widehat{U} := \sum_{k=0}^{n}(-1)^k \langle U \rangle_k, \qquad \widetilde{U} := \sum_{k=0}^{n}(-1)^{\frac{k(k-1)}{2}} \langle U \rangle_k. \tag{62}$$

These mappings have the following well-known properties, which we apply for many times in our work:

$$\widehat{(UV)} = \widehat{U}\widehat{V}, \qquad \widetilde{(UV)} = \widetilde{V}\widetilde{U}, \qquad \forall U, V \in C\ell_{p,q,r}. \tag{63}$$

Let us show that these operations are in some sense unique in $C\ell_{p,q,r}$. For this purpose, we need to consider another equivalent definition of geometric algebras $C\ell_{p,q,r}$. More detailed discussion of definitions and statements of this section can be found, for example, in Lounesto, 1997; Helmstetter & Micali, 2008.

The geometric (Clifford) algebra $C\ell(V)$ is an associative algebra with the unity $e$ over a vector space $V$ with a symmetric bilinear form $\mathfrak{b} : V \times V \to \mathbb{F}$ and the corresponding quadratic form $\mathfrak{q} : V \to \mathbb{F}$, with a linear map $j : V \to C\ell(V)$ such that

$$j^2(x) = \mathfrak{q}(x)e, \qquad \forall x \in V, \tag{64}$$

and for any other associative algebra $A$ over $V$ with the unity $e_A$ and any linear map $k : V \to A$, such that $k^2(x) = \mathfrak{q}(x)e_A$ for any $x \in V$, there exists a unique algebra homomorphism $f : C\ell(V) \to A$, such that $f \circ j = k$, i.e. the corresponding diagram is commutative:

$$
\begin{array}{ccc}
V & \xrightarrow{\;j\;} & C\ell(V) \\
 & \searrow^{k} & \downarrow^{f} \\
 & & A
\end{array}
$$

**Theorem C.1.** *In any geometric algebra $C\ell(V)$, there is a unique antiautomorphism $\tau : C\ell(V) \to C\ell(V)$, such that*

$$\tau(xy) = \tau(y)\tau(x), \quad \tau \circ \tau = \mathrm{id}, \quad \tau(j(v)) = j(v), \qquad \forall x, y \in C\ell(V), \quad \forall v \in V. \tag{65}$$

*Proof.* Let us consider the opposite algebra $C\ell(V)^o$, in which the product of $x$ and $y$ is given by $yx$. Then there exists a unique algebra homomorphism $\tau : C\ell(V) \to C\ell(V)^o$ as in the diagram below

$$
\begin{array}{ccc}
V & \xrightarrow{\;j\;} & C\ell(V) \\
 & \searrow^{j} & \downarrow^{\tau} \\
 & & C\ell(V)^o
\end{array}
$$

This completes the proof. $\qquad\square$

Since in our case, $V$ has finite dimension $n$ with a basis $e_1, \ldots, e_n$, the mapping $\tau$ is completely determined by its action on the basis elements. Namely, $\tau$ is defined by

$$\tau(e_i) = e_i, \qquad \tau(e_{i_1}e_{i_2}\cdots e_{i_k}) = e_{i_k}e_{i_{k-1}}\cdots e_{i_1}, \qquad \tau(e) = e, \tag{66}$$

where $1 \le i_1 < i_2 < \cdots < i_k \le n$. The mapping $\tau$ coincides with the reversion $\widetilde{\phantom{x}}$ defined as in (62).

**Theorem C.2.** *In any geometric algebra $C\ell(V)$, there is a unique automorphism $\alpha : C\ell(V) \to C\ell(V)$, such that*

$$\alpha(xy) = \alpha(x)\alpha(y), \qquad \alpha \circ \alpha = \mathrm{id}, \qquad \alpha(j(v)) = -j(v), \qquad \forall x, y \in C\ell(V), \quad \forall v \in V. \tag{67}$$

*Proof.* Let us consider the linear mapping $\alpha_0 : V \to C\ell(V)$ defined as $\alpha_0(v) = -j(v)$ for any $v \in V$. We get a unique homomorphism $\alpha$ as in the diagram below

$$
\begin{array}{ccc}
V & \xrightarrow{\ j\ } & C\ell(V) \\
 & {\alpha_0}\searrow & \downarrow{\alpha} \\
 & & C\ell(V)
\end{array}
$$

Furthermore, any $x \in C\ell(V)$ can be represented as

$$x = \sum x_1 \cdots x_m, \qquad x_1, \ldots, x_m \in j(V), \tag{68}$$

and since $\alpha(x_j) = -x_j$ for any $j = 1, \ldots, m$, we get $\alpha \circ \alpha = \mathrm{id}$. This completes the proof. $\qquad\square$

Again, since $V$ has finite dimension, the mapping $\alpha$ is completely determined by its action on the basis elements. Namely, $\alpha$ is defined by

$$\alpha(e_i) = -e_i, \qquad \alpha(e_{i_1} e_{i_2} \cdots e_{i_k}) = (-1)^k e_{i_1} e_{i_2} \cdots e_{i_k}, \qquad \alpha(e) = e, \tag{69}$$

where $1 \leq i_1 < i_2 < \cdots < i_k \leq n$. The mapping $\alpha$ coincides with the grade involution $\widehat{\phantom{x}}$ defined as in (62).

## D. Centralizers and Twisted Centralizers

We denote the *center* of the algebra $C\ell_{p,q,r}$ by $\mathrm{Z}_{p,q,r}$. It is a well-known fact (Crumeyrolle, 1990) that the center has the following form

$$
\mathrm{Z}_{p,q,r} = \begin{cases} (\Lambda_r^{(0)} \oplus C\ell_{p,q,r}^n)^\times, & n \text{ is odd;} \\ \Lambda_r^{(0)\times}, & n \text{ is even.} \end{cases} \tag{70}
$$

Consider the *centralizers* (see, for example, Garling, 2011; Isaacs, 2009) of the subspaces of fixed grades $C\ell_{p,q,r}^m$, $m = 0, \ldots, n$, in $C\ell_{p,q,r}$:

$$\mathrm{Z}_{p,q,r}^m := \{ X \in C\ell_{p,q,r} : \ XV = VX, \ \forall V \in C\ell_{p,q,r}^m \}. \tag{71}$$

The centralizer $\mathrm{Z}_{p,q,r}^m$ contains all the elements of $C\ell_{p,q,r}$ that commute with any grade-$m$ element. The center of the Clifford algebra $C\ell_{p,q,r}$ is the centralizer of the entire Clifford algebra $C\ell_{p,q,r}$.

Similarly, we consider the *twisted centralizers* of $C\ell_{p,q,r}^m$, $m = 0, \ldots, n$, in $C\ell_{p,q,r}$:

$$\check{\mathrm{Z}}_{p,q,r}^m := \{ X \in C\ell_{p,q,r} : \ \widehat{X}V = VX, \ \forall V \in C\ell_{p,q,r}^m \}. \tag{72}$$

Note that $\mathrm{Z}_{p,q,r}^m = \check{\mathrm{Z}}_{p,q,r}^m = C\ell_{p,q,r}$ for $m < 0$ and $m > n$. The particular case $\check{\mathrm{Z}}_{p,q,r}^1$ is considered in the papers Ruhe et al., 2023; Brooke, 1980; Filimoshina & Shirokov, 2024b; 2023.

Note that the projections $\langle \mathrm{Z}_{p,q,r}^m \rangle_{(0)}$ and $\langle \check{\mathrm{Z}}_{p,q,r}^m \rangle_{(0)}$ of $\mathrm{Z}_{p,q,r}^m$ and $\check{\mathrm{Z}}_{p,q,r}^m$ respectively onto the even subspace $C\ell_{p,q,r}^{(0)}$ coincide by definition:

$$\langle \mathrm{Z}_{p,q,r}^m \rangle_{(0)} = \langle \check{\mathrm{Z}}_{p,q,r}^m \rangle_{(0)}, \qquad \forall m = 0, 1, \ldots, n. \tag{73}$$

The paper Filimoshina & Shirokov, 2024b studies explicit forms of the sets $\mathrm{Z}_{p,q,r}^m$ and $\check{\mathrm{Z}}_{p,q,r}^m$ in the case of arbitrary $m = 0, \ldots, n$. In Remark D.1, for the readers' convenience, we write out several particular cases of these sets, which we further use. A few words about the notation in this remark and below. The spaces $C\ell_{p,q}^k$ and $\Lambda_r^k$, $k = 0, \ldots, n$, are regarded as subspaces of $C\ell_{p,q,r}$. By $\{C\ell_{p,q}^k \Lambda_r^l\}$, we denote the subspace of $C\ell_{p,q,r}$ spanned by the elements of the form $ab$, where $a \in C\ell_{p,q}^k$ and $b \in \Lambda_r^l$.

*Remark* D.1 (Explicit forms of centralizers and twisted centralizers). We have:

$$Z_{p,q,r}^1 = Z_{p,q,r} = \begin{cases} \Lambda_r^{(0)} \oplus C\ell_{p,q,r}^n, & n \text{ is odd,} \\ \Lambda_r^{(0)}, & n \text{ is even;} \end{cases} \tag{74}$$

$$Z_{p,q,r}^2 = \begin{cases} \Lambda_r \oplus C\ell_{p,q,r}^n, & r \neq n, \\ \Lambda_r, & r = n; \end{cases} \tag{75}$$

$$Z_{p,q,r}^3 = \begin{cases} \Lambda_r^{(0)} \oplus \Lambda_r^{n-2} \oplus \{C\ell_{p,q}^1(\Lambda_r^{n-3} \oplus \Lambda_r^{n-2})\} \oplus \{C\ell_{p,q}^2 \Lambda_r^{n-3}\} \oplus C\ell_{p,q,r}^n, & n \text{ is odd,} \\ \Lambda_r^{(0)} \oplus \Lambda_r^{n-1} \oplus \{C\ell_{p,q}^1 \Lambda_r^{\geq n-2}\} \oplus \{C\ell_{p,q}^2 \Lambda_r^{n-2}\}, & n \text{ is even;} \end{cases} \tag{76}$$

$$Z_{p,q,r}^4 = \begin{cases} \Lambda_r \oplus \{C\ell_{p,q}^1(\Lambda_r^{n-3} \oplus \Lambda_r^{n-2})\} \oplus \{C\ell_{p,q}^2(\Lambda_r^{n-4} \oplus \Lambda_r^{n-3})\} \oplus C\ell_{p,q,r}^n, & r \neq n, \\ \Lambda_r, & r = n; \end{cases} \tag{77}$$

and:

$$\check{Z}_{p,q,r}^1 = \Lambda_r; \tag{78}$$

$$\check{Z}_{p,q,r}^2 = \begin{cases} \Lambda_r^{(0)} \oplus \Lambda_r^n \oplus \{C\ell_{p,q}^1 \Lambda_r^{n-1}\}, & n \text{ is odd,} \\ \Lambda_r^{(0)} \oplus \Lambda_r^{n-1} \oplus \{C\ell_{p,q}^1 \Lambda_r^{n-2}\} \oplus C\ell_{p,q,r}^n, & n \text{ is even, } r \neq n; \\ \Lambda_r^{(0)} \oplus \Lambda_r^{n-1}, & n \text{ is even, } r = n; \end{cases} \tag{79}$$

$$\check{Z}_{p,q,r}^3 = \Lambda_r \oplus \{C\ell_{p,q}^1 \Lambda_r^{\geq n-2}\} \oplus \{C\ell_{p,q}^2 \Lambda_r^{\geq n-3}\}. \tag{80}$$

*Remark* D.2. In the particular case of the non-degenerate geometric algebras $C\ell_{p,q}$, we have

$$Z_{p,q}^1 = Z_{p,q} = \begin{cases} C\ell^0 \oplus C\ell_{p,q}^n, & n \text{ is odd,} \\ C\ell^0, & n \text{ is even;} \end{cases} \qquad Z_{p,q}^2 = C\ell^0 \oplus C\ell_{p,q}^n; \tag{81}$$

$$Z_{p,q}^3 = \begin{cases} C\ell_{p,q}, & n = 2,3, \\ C\ell^0, & n \neq 2,3; \end{cases} \qquad Z_{p,q}^4 = \begin{cases} C\ell_{p,q}, & n = 2,3, \\ C\ell_{p,q}^{(0)}, & n = 4, \\ C\ell^0 \oplus C\ell_{p,q}^n, & n \neq 2,3,4; \end{cases} \tag{82}$$

$$\check{Z}_{p,q}^1 = C\ell^0, \qquad \check{Z}_{p,q}^2 = \begin{cases} C\ell_{p,q}, & n = 1,2, \\ C\ell^0, & n \neq 1 \text{ is odd,} \\ C\ell^0 \oplus C\ell_{p,q}^n, & n \neq 2 \text{ is even;} \end{cases} \qquad \check{Z}_{p,q}^3 = \begin{cases} C\ell_{p,q}, & n = 1,2, \\ C\ell_{p,q}^{(0)}, & n = 3, \\ C\ell^0, & n \geq 4. \end{cases} \tag{83}$$

Using the definition of the even subspace $C\ell_{p,q,r}^{(0)}$ and explicit forms of centralizers of fixed grades, presented in Filimoshina & Shirokov, 2024b, the following lemma can be obtained.

**Lemma D.3.** *We have*

$$\{X \in C\ell_{p,q,r}: \quad XV = VX, \quad \forall V \in C\ell_{p,q,r}^{(0)}\} = \begin{cases} \Lambda_r \oplus C\ell_{p,q,r}^n, & r \neq n, \\ \Lambda_n, & r = n, \end{cases} \tag{84}$$

$$\{X \in C\ell_{p,q,r}: \quad \widehat{X}V = VX, \quad \forall V \in C\ell_{p,q,r}^{(0)}\} = \begin{cases} \Lambda_r^{(0)}, & n \text{ is odd; } r = n \text{ is even;} \\ \Lambda_r^{(0)} \oplus C\ell_{p,q,r}^n, & n \text{ is even, } r \neq n. \end{cases} \tag{85}$$

Now let us consider the centralizers and twisted centralizers respectively of the subspaces $C\ell_{p,q,r}^{\overline{k}}$ (4), $k = 0, 1, 2, 3$, defined by the grade involution and reversion, in $C\ell_{p,q,r}$:

$$Z_{p,q,r}^{\overline{k}} := \{X \in C\ell_{p,q,r}: \quad XV = VX, \quad \forall V \in C\ell_{p,q,r}^{\overline{k}}\}, \tag{86}$$

$$\check{Z}_{p,q,r}^{\overline{k}} := \{X \in C\ell_{p,q,r}: \quad \widehat{X}V = VX, \quad \forall V \in C\ell_{p,q,r}^{\overline{k}}\}. \tag{87}$$

**Theorem D.4.** *(Filimoshina & Shirokov, 2024b) We have*

$$Z_{p,q,r}^{\overline{m}} = Z_{p,q,r}^m, \quad \check{Z}_{p,q,r}^{\overline{m}} = \check{Z}_{p,q,r}^m, \quad m = 1,2,3; \qquad Z_{p,q,r}^{\overline{0}} = Z_{p,q,r}^4, \quad \check{Z}_{p,q,r}^{\overline{0}} = \langle Z_{p,q,r}^4 \rangle_{(0)}. \tag{88}$$

The centralizers $Z_{p,q,r}^m$ and twisted centralizers $\check{Z}_{p,q,r}^m$, $m = 1, 2, 3, 4$, are written out explicitly in Remark D.1. In the formula (88), we have

$$\langle Z_{p,q,r}^4 \rangle_{(0)} = \begin{cases} \Lambda_r^{(0)} \oplus \{Cl_{p,q}^1 \Lambda_r^{n-2}\} \oplus \{Cl_{p,q}^2 \Lambda_r^{n-3}\}, & n \text{ is odd or } r = n; \\ \Lambda_r^{(0)} \oplus \{Cl_{p,q}^1 \Lambda_r^{n-3}\} \oplus \{Cl_{p,q}^2 \Lambda_r^{n-4}\} \oplus Cl_{p,q,r}^n, & n \text{ is even}, r \neq n. \end{cases} \tag{89}$$

Let us consider the following Lemma D.5 about $T \in Cl_{p,q,r}$ such that $\psi(T)$ and $\chi(T)$ (11) are in the centralizers $Z_{p,q,r}^{\overline{k}}$ (86) and twisted centralizers $\check{Z}_{p,q,r}^{\overline{k}}$ (87). The proofs of theorems in Appendix F are based on this lemma.

**Lemma D.5.** *(Filimoshina & Shirokov, 2025) For any $T \in Cl_{p,q,r}^\times$, in the cases $(k, l) = (0, 1), (1, 0), (2, 3), (3, 2)$:*

$$TCl_{p,q,r}^{\overline{k}} T^{-1} \subseteq Cl_{p,q,r}^{\overline{kl}} \quad \Leftrightarrow \quad \widetilde{T} T \in Z_{p,q,r}^{\overline{k}\times}, \tag{90}$$

$$\widehat{T} Cl_{p,q,r}^{\overline{k}} T^{-1} \subseteq Cl_{p,q,r}^{\overline{kl}} \quad \Leftrightarrow \quad \widehat{\widetilde{T}} T \in \check{Z}_{p,q,r}^{\overline{k}\times}, \tag{91}$$

*and in the cases $(k, l) = (0, 3), (3, 0), (1, 2), (2, 1)$, we have:*

$$TCl_{p,q,r}^{\overline{k}} T^{-1} \subseteq Cl_{p,q,r}^{\overline{kl}} \quad \Leftrightarrow \quad \widehat{\widetilde{T}} T \in Z_{p,q,r}^{\overline{k}\times}, \tag{92}$$

$$\widehat{T} Cl_{p,q,r}^{\overline{k}} T^{-1} \subseteq Cl_{p,q,r}^{\overline{kl}} \quad \Leftrightarrow \quad \widetilde{T} T \in \check{Z}_{p,q,r}^{\overline{k}\times}. \tag{93}$$

## E. Ordinary Lipschitz Groups, Spin Groups, Adjoint and Twisted Adjoint Representations

In the non-degenerate geometric algebras $Cl_{p,q}$, the Lipschitz group is defined as the group of all invertible elements preserving the grade-1 subspace under the twisted adjoint representation $\check{\mathrm{ad}}$. Generalizing this definition to the case of the degenerate $Cl_{p,q,r}$, we similarly define the *Lipschitz group* $\tilde{\Gamma}_{p,q,r}^1$ as

$$\tilde{\Gamma}_{p,q,r}^1 := \{T \in Cl_{p,q,r}^\times : \quad \widehat{T} Cl_{p,q,r}^1 T^{-1} \subseteq Cl_{p,q,r}^1\}. \tag{94}$$

The definition (94) of the degenerate Lipschitz group is used, for example, in the works Brooke, 1980; 1978; Crumeyrolle, 1990.

The well-known spin groups (see, for example, Lounesto, 1997; Crumeyrolle, 1990) are defined as normalized subgroups of the Lipschitz groups $\tilde{\Gamma}_{p,q,r}^1$, using the norm functions $\psi$ and $\chi$ (11). For example, in the non-degenerate geometric algebras $Cl_{p,q}$, they are defined as

$$\mathrm{Pin}_{p,q} := \{T \in \tilde{\Gamma}_{p,q}^1 : \quad \widetilde{T} T = \pm e\} = \{T \in \tilde{\Gamma}_{p,q}^1 : \quad \widehat{\widetilde{T}} T = \pm e\}, \tag{95}$$

$$\mathrm{Pin}_{p,q}^\psi := \{T \in \tilde{\Gamma}_{p,q}^1 : \quad \widetilde{T} T = +e\}, \tag{96}$$

$$\mathrm{Pin}_{p,q}^\chi := \{T \in \tilde{\Gamma}_{p,q}^1 : \quad \widehat{\widetilde{T}} T = +e\}, \tag{97}$$

$$\mathrm{Spin}_{p,q} := \{T \in \tilde{\Gamma}_{p,q}^1 \cap Cl_{p,q}^{(0)\times} : \quad \widetilde{T} T = \pm e\} = \{T \in \tilde{\Gamma}_{p,q}^1 \cap Cl_{p,q}^{(0)\times} : \quad \widehat{\widetilde{T}} T = \pm e\}, \tag{98}$$

$$\mathrm{Spin}_{p,q}^+ := \{T \in \tilde{\Gamma}_{p,q}^1 \cap Cl_{p,q}^{(0)\times} : \quad \widetilde{T} T = +e\} = \{T \in \tilde{\Gamma}_{p,q}^1 \cap Cl_{p,q}^{(0)\times} : \quad \widehat{\widetilde{T}} T = +e\}. \tag{99}$$

Let us consider the *adjoint representation* $\mathrm{ad}$ acting on the group of all invertible elements $\mathrm{ad} : Cl_{p,q,r}^\times \to \mathrm{Aut}(Cl_{p,q,r})$ as $T \mapsto \mathrm{ad}_T$, where $\mathrm{ad}_T : Cl_{p,q,r} \to Cl_{p,q,r}$:

$$\mathrm{ad}_T(U) = TUT^{-1}, \qquad U \in Cl_{p,q,r}, \qquad T \in Cl_{p,q,r}^\times. \tag{100}$$

Since $\mathrm{ad}_T(UV) = \mathrm{ad}_T(U)\mathrm{ad}_T(V)$ for all $U, V \in Cl_{p,q,r}$, the mapping $\mathrm{ad}_T$ is an algebra homomorphism, which is moreover an algebra automorphism for all $T \in Cl_{p,q,r}^\times$. It is called *inner automorphism* (or conjugation).

Also let us consider the *twisted adjoint representation*, which is introduced by Atiyah, Bott, and Shapiro in Atiyah et al., 1964. In this paper, the twisted adjoint representation is defined in the case of the non-degenerate algebra $Cl_{0,q,0}$. It acts

on the Lipschitz group $\tilde{\Gamma}^1_{0,q,0}$ (94) in the way $\breve{\mathrm{ad}} : \Gamma^{\pm}_{0,q,0} \to \mathrm{Aut}(Cl^1_{0,q,0})$ as $T \to \breve{\mathrm{ad}}_T$, where $\breve{\mathrm{ad}}_T : Cl^1_{0,q,0} \to Cl^1_{0,q,0}$ is defined for elements of the grade-1 subspace (vectors) as

$$\breve{\mathrm{ad}}_T(U) = \widehat{T}UT^{-1}, \qquad U \in Cl^1_{0,q,0}, \qquad T \in \tilde{\Gamma}^1_{0,q,0}. \tag{101}$$

There are two ways how to generalize the definition (101) of the twisted adjoint representation to the case of any degenerate and non-degenerate $Cl_{p,q,r}$ and arbitrary $T \in Cl^{\times}_{p,q,r}$, $U \in Cl_{p,q,r}$. The first approach (Choi et al., 2002; Harvey, 1990; Lundholm & Svensson, 2009) is to define it as the operation $\breve{\mathrm{ad}}$ acting on the group of all invertible elements $\breve{\mathrm{ad}} : Cl^{\times}_{p,q,r} \to \mathrm{Aut}(Cl_{p,q,r})$ as $T \mapsto \breve{\mathrm{ad}}_T$ with $\breve{\mathrm{ad}}_T : Cl_{p,q,r} \to Cl_{p,q,r}$:

$$\breve{\mathrm{ad}}_T(U) = \widehat{T}UT^{-1}, \qquad U \in Cl_{p,q,r}, \qquad T \in Cl^{\times}_{p,q,r}. \tag{102}$$

The operation $\breve{\mathrm{ad}}_T$ (102) is similar to some operations considered in representation theory of Lie groups (Zerouali, 2020). Note that unlike $\mathrm{ad}_T$ (100) the mapping $\breve{\mathrm{ad}}_T$ (102) is not multiplicative in $U$ and, therefore, is not an algebra homomorphism for all $T \in Cl^{\times}_{p,q,r}$.

The second approach (Helmstetter & Micali, 2008; Knus, 1991; Walpuski, 2022) is to define the twisted adjoint representation as the operation acting on the group of all invertible grade-$m$ elements $\tilde{\mathrm{ad}} : Cl^{m\times}_{p,q,r} \to \mathrm{Aut}(Cl_{p,q,r})$ as $T \mapsto \tilde{\mathrm{ad}}_T$ with $\tilde{\mathrm{ad}}_T : Cl_{p,q,r} \to Cl_{p,q,r}$ defined as:

$$\tilde{\mathrm{ad}}_T(U) = (-1)^{km}TUT^{-1}, \qquad U \in Cl^k_{p,q,r}, \qquad T \in Cl^{m\times}_{p,q,r}, \tag{103}$$

where we use another notation $\tilde{\mathrm{ad}}_T$ not to confuse it with $\breve{\mathrm{ad}}_T$ (102). The mapping $\tilde{\mathrm{ad}}_T$ is also called *twisted inner automorphism*, for example, in Helmstetter & Micali, 2008. The operation $\tilde{\mathrm{ad}}_T$ can be extended by linearity $\tilde{\mathrm{ad}}_T(U + V) = \tilde{\mathrm{ad}}_T(U) + \tilde{\mathrm{ad}}_T(V)$, and we finally obtain

$$\tilde{\mathrm{ad}}_T(U) = T\langle U\rangle_{(0)}T^{-1} + \widehat{T}\langle U\rangle_{(1)}T^{-1}, \qquad \forall U \in Cl_{p,q,r}, \qquad T \in Cl^{\times}_{p,q,r}. \tag{104}$$

Note that for any elements of fixed parity $U_{(0)}$ and $U_{(1)}$, we have

$$\tilde{\mathrm{ad}}_T(U_{(0)}) = \mathrm{ad}_T(U_{(0)}), \qquad \forall U_{(0)} \in Cl^{(0)}_{p,q,r}, \qquad T \in Cl^{\times}_{p,q,r}, \tag{105}$$

$$\tilde{\mathrm{ad}}_T(U_{(1)}) = \breve{\mathrm{ad}}_T(U_{(1)}), \qquad \forall U_{(1)} \in Cl^{(1)}_{p,q,r}, \qquad T \in Cl^{\times}_{p,q,r}. \tag{106}$$

Note that both generalizations $\breve{\mathrm{ad}}$ (102) and $\tilde{\mathrm{ad}}$ (104) of the twisted adjoint representation (101) to the case of arbitrary $T \in Cl^{\times}_{p,q,r}$ and $U \in Cl_{p,q,r}$ are used in the literature for various purposes and have some advantages indicated in the works cited above.

Let us consider the *kernels of the adjoint and the twisted adjoint representations*:

$$\ker(\mathrm{ad}) = \{T \in Cl^{\times}_{p,q,r} : \quad TUT^{-1} = U, \quad \forall U \in Cl_{p,q,r}\},$$

$$\ker(\breve{\mathrm{ad}}) = \{T \in Cl^{\times}_{p,q,r} : \quad \widehat{T}UT^{-1} = U, \quad \forall U \in Cl_{p,q,r}\},$$

$$\ker(\tilde{\mathrm{ad}}) = \{T \in Cl^{\times}_{p,q,r} : TU_{(0)}T^{-1} + \widehat{T}U_{(1)}T^{-1} = U, \forall U = U_{(0)} + U_{(1)} \in Cl_{p,q,r}, \ U_{(0)} \in Cl^{(0)}_{p,q,r}, \ U_{(1)} \in Cl^{(1)}_{p,q,r}\}.$$

**Lemma E.1.** *(Filimoshina & Shirokov, 2023) We have*

$$\ker(\mathrm{ad}) = \mathrm{Z}^{\times}_{p,q,r} = \begin{cases} (\Lambda^{(0)}_r \oplus Cl^n_{p,q,r})^{\times} & \text{if } n \text{ is odd,} \\ \Lambda^{(0)\times}_r & \text{if } n \text{ is even;} \end{cases} \tag{107}$$

$$\ker(\breve{\mathrm{ad}}) = \Lambda^{(0)\times}_r; \tag{108}$$

$$\ker(\tilde{\mathrm{ad}}) = \Lambda^{\times}_r. \tag{109}$$

*Proof.* We obtain (107) from (70).

Let us prove $\Lambda^{(0)\times}_r \subseteq \ker(\breve{\mathrm{ad}})$. Suppose $T \in \Lambda^{(0)\times}_r$; then $TUT^{-1} = U$ for any $U \in Cl_{p,q,r}$. Since $T$ is even, we have $\widehat{T} = T$; therefore, $\widehat{T}UT^{-1} = U$ for any $U \in Cl_{p,q,r}$.

Let us prove $\ker(\breve{\mathrm{ad}}) \subseteq \Lambda_r^{(0)\times}$. Suppose $T \in C\ell_{p,q,r}^{\times}$ satisfies $\widehat{T}UT^{-1} = U$ for any $U \in C\ell_{p,q,r}$. Substituting the element $U = e$, we obtain $\widehat{T} = T$; hence, $T \in C\ell_{p,q,r}^{(0)\times}$ and $TUT^{-1} = U$ for any $U \in C\ell_{p,q,r}$. In other words, $T \in C\ell_{p,q,r}^{(0)\times} \cap \ker(\mathrm{ad})$. Using (107), we obtain $T \in C\ell_{p,q,r}^{(0)\times} \cap (\Lambda_r^{(0)} \oplus C\ell_{p,q,r}^n)^{\times} = \Lambda_r^{(0)\times}$ in the case of odd $n$, $T \in \Lambda_r^{(0)\times}$ in the case of even $n$, and the proof is completed.

Let us prove $\ker(\tilde{\mathrm{ad}}) \subseteq \Lambda_r^{\times}$. Suppose $T \in \ker(\tilde{\mathrm{ad}})$; then $\widehat{T}U_1T^{-1} = U_1$ for any $U_1 \in C\ell_{p,q,r}^1$. Thus, $T \in \Lambda_r^{\times}$ by the statement (78) of Remark D.1.

Now we must only prove that $\Lambda_r^{\times} \subseteq \ker(\tilde{\mathrm{ad}})$. Suppose $T \in \Lambda_r^{\times}$; then $TC\ell_{p,q,r}^{(0)} = C\ell_{p,q,r}^{(0)}T$ and $\widehat{T}C\ell_{p,q,r}^1 = C\ell_{p,q,r}^1T$ by Lemma D.3 and the statement (78) of Remark D.1 respectively. Since any odd basis element can be represented as a product of an odd number of generators, we obtain $\widehat{T}C\ell_{p,q,r}^{(1)} = C\ell_{p,q,r}^{(1)}T$. Thus, $TU_{(0)}T^{-1} + \widehat{T}U_{(1)}T^{-1} = U$ for all $U = U_{(0)} + U_{(1)} \in C\ell_{p,q,r}$, where $U_{(0)} \in C\ell_{p,q,r}^{(0)}$ and $U_{(1)} \in C\ell_{p,q,r}^{(1)}$, and the proof is completed. $\qquad\square$

In the next lemma, we study the properties of $\mathrm{ad}$, $\breve{\mathrm{ad}}$, and $\tilde{\mathrm{ad}}$. Note that the multiplicativity of $\tilde{\mathrm{ad}}_T$ (115) is proved in the particular case $T \in C\ell_{p,q,r}^{(0)\times} \cup C\ell_{p,q,r}^{(1)\times}$ in Ruhe et al., 2023, and we generalize this statement to $T \in (C\ell_{p,q,r}^{(0)\times} \cup C\ell_{p,q,r}^{(1)\times})\Lambda_r^{\times}$.

**Lemma E.2** (Lemma 2.1). *Let* $H \in C\ell_{p,q,r}^{(0)\times} \cup C\ell_{p,q,r}^{(1)\times}$, $T \in (C\ell_{p,q,r}^{(0)\times} \cup C\ell_{p,q,r}^{(1)\times})\Lambda_r^{\times}$, $W \in C\ell_{p,q,r}^{\times}$, *and* $U, V \in C\ell_{p,q,r}$. *Then* $\mathrm{ad}_W$, $\breve{\mathrm{ad}}_W$, *and* $\tilde{\mathrm{ad}}_W$ *satisfy additivity:*

$$\mathrm{ad}_W(U + V) = \mathrm{ad}_W(U) + \mathrm{ad}_W(V), \tag{110}$$

$$\breve{\mathrm{ad}}_W(U + V) = \breve{\mathrm{ad}}_W(U) + \breve{\mathrm{ad}}_W(V), \tag{111}$$

$$\tilde{\mathrm{ad}}_W(U + V) = \tilde{\mathrm{ad}}_W(U) + \tilde{\mathrm{ad}}_W(V), \tag{112}$$

$\mathrm{ad}_W$, $\breve{\mathrm{ad}}_H$, *and* $\tilde{\mathrm{ad}}_W$ *satisfy for any* $c \in C\ell^0$,

$$\mathrm{ad}_W(c) = \tilde{\mathrm{ad}}_W(c) = c, \quad \breve{\mathrm{ad}}_H(c) = (-1)^l c, \tag{113}$$

*where* $l = 0$ *if* $H \in C\ell_{p,q,r}^{(0)\times}$ *and* $l = 1$ *if* $H \in C\ell_{p,q,r}^{(1)\times}$. *Moreover,* $\mathrm{ad}_W$ *and* $\tilde{\mathrm{ad}}_T$ *satisfy multiplicativity:*

$$\mathrm{ad}_W(UV) = \mathrm{ad}_W(U)\mathrm{ad}_W(V), \tag{114}$$

$$\tilde{\mathrm{ad}}_T(UV) = \tilde{\mathrm{ad}}_T(U)\tilde{\mathrm{ad}}_T(V). \tag{115}$$

*Proof.* The formulas (110)–(113) can be easily checked using the definitions of $\mathrm{ad}$ (100), $\breve{\mathrm{ad}}$ (102), and $\tilde{\mathrm{ad}}$ (104). Let us prove (114) and (115). We have

$$\mathrm{ad}_W(U)\mathrm{ad}_W(V) = WUW^{-1}WVW^{-1} = \mathrm{ad}_W(UV).$$

Since $U = \langle U \rangle_{(0)} + \langle U \rangle_{(1)}$, we get

$$\tilde{\mathrm{ad}}_T(U)\tilde{\mathrm{ad}}_T(V) = (T\langle U \rangle_{(0)}T^{-1} + \widehat{T}\langle U \rangle_{(1)}T^{-1})(T\langle V \rangle_{(0)}T^{-1} + \widehat{T}\langle V \rangle_{(1)}T^{-1}) \tag{116}$$

$$= T\langle U \rangle_{(0)}\langle V \rangle_{(0)}T^{-1} + \widehat{T}\langle U \rangle_{(1)}\langle V \rangle_{(0)}T^{-1} + T\langle U \rangle_{(0)}T^{-1}\widehat{T}\langle V \rangle_{(1)}T^{-1} + \widehat{T}\langle U \rangle_{(1)}T^{-1}\widehat{T}\langle V \rangle_{(1)}T^{-1}. \tag{117}$$

Now we apply the result of Theorem 4.7 Filimoshina & Shirokov, 2023:

$$(C\ell_{p,q,r}^{(0)\times} \cup C\ell_{p,q,r}^{(1)\times})\Lambda_r^{\times} = \{T \in C\ell_{p,q,r}^{\times} : \quad \widehat{T^{-1}T} \in \Lambda_r^{\times}\}$$

and get $\widehat{T^{-1}T} \in \Lambda_r^{\times}$, which implies $T^{-1}\widehat{T} = \widehat{\widehat{T^{-1}T}} \in \Lambda_r^{\times}$. Since $\Lambda_r^{\times} = \ker(\tilde{\mathrm{ad}})$ by Lemma E.1, we obtain that $(T^{-1}\widehat{T})X = X(T^{-1}\widehat{T})$ for any even $X \in C\ell_{p,q,r}^{(0)}$ and $(T^{-1}\widehat{T})X = X(\widehat{T^{-1}\widehat{T}})$ for any odd $X \in C\ell_{p,q,r}^{(1)}$. Therefore,

$$T\langle U \rangle_{(0)}T^{-1}\widehat{T}\langle V \rangle_{(1)}T^{-1} = TT^{-1}\widehat{T}\langle U \rangle_{(0)}\langle V \rangle_{(1)}T^{-1} = \widehat{T}\langle U \rangle_{(0)}\langle V \rangle_{(1)}T^{-1}, \tag{118}$$

$$\widehat{T}\langle U \rangle_{(1)}T^{-1}\widehat{T}\langle V \rangle_{(1)}T^{-1} = \widehat{T}\widehat{T^{-1}}T\langle U \rangle_{(1)}\langle V \rangle_{(1)}T^{-1} = T\langle U \rangle_{(1)}\langle V \rangle_{(1)}T^{-1}. \tag{119}$$

Finally,

$$\tilde{\mathrm{ad}}_T(U)\tilde{\mathrm{ad}}_T(V) = T\langle U \rangle_{(0)}\langle V \rangle_{(0)}T^{-1} + \widehat{T}\langle U \rangle_{(1)}\langle V \rangle_{(0)}T^{-1} + \widehat{T}\langle U \rangle_{(0)}\langle V \rangle_{(1)}T^{-1} + T\langle U \rangle_{(1)}\langle V \rangle_{(1)}T^{-1} = \tilde{\mathrm{ad}}_T(UV),$$

where we use that the product of two elements of the same parity is even and the product of two elements of different parity is odd. $\qquad\square$

# F. Generalized Lipschitz Groups in $Cl_{p,q,r}$

In this section, we introduce and study degenerate and non-degenerate generalized Lipschitz groups in geometric algebras $Cl_{p,q,r}$ of arbitrary dimension and signature.

The *generalized Lipschitz groups* are defined as the groups preserving the subspaces $Cl_{p,q,r}^{\overline{k}}$ (4), $k = 0, 1, 2, 3$, determined by the grade involution and reversion, under the twisted adjoint representation $\tilde{\mathrm{ad}}$ (104):

$$\tilde{\Gamma}_{p,q,r}^{\overline{k}} := \{T \in Cl_{p,q,r}^{\times} : \ \tilde{\mathrm{ad}}_T(Cl_{p,q,r}^{\overline{k}}) \subseteq Cl_{p,q,r}^{\overline{k}}\}. \tag{120}$$

These groups are setwise stabilizers (see, for example, Isaacs, 2009) of the subspaces $Cl_{p,q,r}^{\overline{k}}$ (4), $k = 0, 1, 2, 3$, in the group $Cl_{p,q,r}^{\times}$ under the group action $\tilde{\mathrm{ad}}$. To study these groups, we introduce and study the following groups, which are setwise stabilizers of $Cl_{p,q,r}^{\overline{k}}$, $k = 0, 1, 2, 3$, in $Cl_{p,q,r}^{\times}$ under $\mathrm{ad}$ (100) and $\check{\mathrm{ad}}$ (102) respectively:

$$\Gamma_{p,q,r}^{\overline{k}} := \{T \in Cl_{p,q,r}^{\times} : \quad \mathrm{ad}_T(Cl_{p,q,r}^{\overline{k}}) := T Cl_{p,q,r}^{\overline{k}} T^{-1} \subseteq Cl_{p,q,r}^{\overline{k}}\}, \tag{121}$$

$$\check{\Gamma}_{p,q,r}^{\overline{k}} := \{T \in Cl_{p,q,r}^{\times} : \quad \check{\mathrm{ad}}_T(Cl_{p,q,r}^{\overline{k}}) := \widehat{T} Cl_{p,q,r}^{\overline{k}} T^{-1} \subseteq Cl_{p,q,r}^{\overline{k}}\}, \tag{122}$$

and are related to $\tilde{\Gamma}_{p,q,r}^{\overline{k}}$ (120) as follows:

$$\tilde{\Gamma}_{p,q,r}^{\overline{k}} = \begin{cases} \check{\Gamma}_{p,q,r}^{\overline{k}}, & k = 1, 3, \\ \Gamma_{p,q,r}^{\overline{k}}, & k = 0, 2, \end{cases} \tag{123}$$

since $\check{\mathrm{ad}}_T(Cl_{p,q,r}^{\overline{k}}) = \mathrm{ad}_T(Cl_{p,q,r}^{\overline{k}})$ in the cases $k = 0, 2$ and $\tilde{\mathrm{ad}}_T(Cl_{p,q,r}^{\overline{k}}) = \check{\mathrm{ad}}_T(Cl_{p,q,r}^{\overline{k}})$ in the cases $k = 1, 3$ by (105)–(106).

In Subsections F.1 and F.2, we find the equivalent definitions of the groups $\Gamma_{p,q,r}^{\overline{k}}$ and $\check{\Gamma}_{p,q,r}^{\overline{k}}$, $k = 0, 1, 2, 3$.

## F.1. The Groups $Q_{p,q,r}^{\overline{0}}, Q_{p,q,r}^{\overline{1}}, Q_{p,q,r}^{\overline{2}}, Q_{p,q,r}^{\overline{3}}, \Gamma_{p,q,r}^{\overline{0}}, \Gamma_{p,q,r}^{\overline{1}}, \Gamma_{p,q,r}^{\overline{2}}$, and $\Gamma_{p,q,r}^{\overline{3}}$

Let us consider the groups $Q_{p,q,r}^{\overline{1}}, Q_{p,q,r}^{\overline{2}}, Q_{p,q,r}^{\overline{3}}$, and $Q_{p,q,r}^{\overline{0}}$:

$$Q_{p,q,r}^{\overline{1}} := \{T \in Cl_{p,q,r}^{\times} : \quad \widetilde{T} T \in Z_{p,q,r}^{1\times} = \ker(\mathrm{ad}), \quad \widehat{\widetilde{T}} T \in Z_{p,q,r}^{1\times} = \ker(\mathrm{ad})\}, \tag{124}$$

$$Q_{p,q,r}^{\overline{2}} := \{T \in Cl_{p,q,r}^{\times} : \quad \widetilde{T} T \in Z_{p,q,r}^{2\times}, \quad \widehat{\widetilde{T}} T \in Z_{p,q,r}^{2\times}\}, \tag{125}$$

$$Q_{p,q,r}^{\overline{3}} := \{T \in Cl_{p,q,r}^{\times} : \quad \widetilde{T} T \in Z_{p,q,r}^{3\times}, \quad \widehat{\widetilde{T}} T \in Z_{p,q,r}^{3\times}\}, \tag{126}$$

$$Q_{p,q,r}^{\overline{0}} := \{T \in Cl_{p,q,r}^{\times} : \quad \widetilde{T} T \in Z_{p,q,r}^{4\times}, \quad \widehat{\widetilde{T}} T \in Z_{p,q,r}^{4\times}\}, \tag{127}$$

where $\ker(\mathrm{ad})$ (107) is the kernel of the adjoint representation $\mathrm{ad}$ (100), $Z_{p,q,r}^1, Z_{p,q,r}^2, Z_{p,q,r}^3$, and $Z_{p,q,r}^4$ (see Remark D.1) are the centralizers of the subspaces $Cl_{p,q,r}^1, Cl_{p,q,r}^2, Cl_{p,q,r}^3$, and $Cl_{p,q,r}^4$ respectively considered in Section D.

The groups $Q_{p,q,r}^{\overline{1}}, Q_{p,q,r}^{\overline{2}}, Q_{p,q,r}^{\overline{3}}$, and $Q_{p,q,r}^{\overline{0}}$ are generalizations of the groups $Q$ and $Q'$ (Shirokov, 2021) in the non-degenerate geometric algebras $Cl_{p,q}$ to the case of the degenerate geometric algebras $Cl_{p,q,r}$ and coincide with them if $r = 0$:

$$Q_{p,q}^{\overline{1}} = Q_{p,q}^{\overline{3}} = Q, \qquad Q_{p,q}^{\overline{0}} = Q_{p,q}^{\overline{2}} = \begin{cases} Q, & n = 1, 2, 3 \bmod 4, \\ Q', & n = 0 \bmod 4, \end{cases} \qquad n \neq 2, 3,$$

and $Q_{p,q}^{\overline{1}} = Q_{p,q}^{\overline{2}} = Q$ if $n = 2, 3$.

**Theorem F.1** (Formulas (22) and (25) of Theorem 3.2)**.** *In the degenerate and non-degenerate geometric algebras $Cl_{p,q,r}$, we have*

$$Q_{p,q,r}^{\overline{1}} = \Gamma_{p,q,r}^{\overline{1}}, \qquad Q_{p,q,r}^{\overline{2}} = \Gamma_{p,q,r}^{\overline{2}}, \qquad Q_{p,q,r}^{\overline{3}} = \Gamma_{p,q,r}^{\overline{3}}, \qquad Q_{p,q,r}^{\overline{0}} = \Gamma_{p,q,r}^{\overline{0}},$$

*where*

$$\Gamma_{p,q,r}^{\overline{1}} \subseteq \Gamma_{p,q,r}^{\overline{m}} \subseteq \Gamma_{p,q,r}^{\overline{0}}, \qquad m = 2, 3. \tag{128}$$

*Proof.* For fixed $k = 0, 1, 2, 3 \mod 4$ and $m = (k-1) \mod 4$, $l = (k+1) \mod 4$, we get

$$
\begin{align}
\Gamma_{p,q,r}^{\overline{k}} &= \{T \in C\ell_{p,q,r}^{\times} : \quad T C\ell_{p,q,r}^{\overline{k}} T^{-1} \subseteq (C\ell_{p,q,r}^{\overline{km}} \cap C\ell_{p,q,r}^{\overline{kl}})\} \tag{129} \\
&= \{T \in C\ell_{p,q,r}^{\times} : \quad T C\ell_{p,q,r}^{\overline{k}} T^{-1} \subseteq C\ell_{p,q,r}^{\overline{km}}, \quad T C\ell_{p,q,r}^{\overline{k}} T^{-1} \subseteq C\ell_{p,q,r}^{\overline{kl}}\} \tag{130} \\
&= \{T \in C\ell_{p,q,r}^{\times} : \quad \widetilde{T}T \in \mathrm{Z}_{p,q,r}^{\overline{k}\times}, \quad \widehat{\widetilde{T}}T \in \mathrm{Z}_{p,q,r}^{\overline{k}\times}\} = \mathrm{Q}_{p,q,r}^{\overline{k}}, \tag{131}
\end{align}
$$

where we use Lemma D.5 in the first equality (131) and $\mathrm{Z}_{p,q,r}^{\overline{k}} = \mathrm{Z}_{p,q,r}^{k}$, $k = 1, 2, 3$, and $\mathrm{Z}_{p,q,r}^{\overline{0}} = \mathrm{Z}_{p,q,r}^{4}$ by Lemma D.4 in the second equality (131). The inclusions in (128) follow from the definitions (124)–(127) and the following facts about explicit forms of centralizers and twisted centralizers implied by Remark D.1:

$$
\ker(\mathrm{ad}) = \mathrm{Z}_{p,q,r}^{1\times} \subseteq \mathrm{Z}_{p,q,r}^{m\times} \subseteq \mathrm{Z}_{p,q,r}^{4\times}, \qquad m = 2, 3. \tag{132}
$$

This completes the proof. $\qquad\square$

## F.2. The Groups $\check{\mathrm{Q}}_{p,q,r}^{\overline{0}}, \check{\mathrm{Q}}_{p,q,r}^{\overline{1}}, \check{\mathrm{Q}}_{p,q,r}^{\overline{2}}, \check{\mathrm{Q}}_{p,q,r}^{\overline{3}}, \check{\Gamma}_{p,q,r}^{\overline{0}}, \check{\Gamma}_{p,q,r}^{\overline{1}}, \check{\Gamma}_{p,q,r}^{\overline{2}}$, and $\check{\Gamma}_{p,q,r}^{\overline{3}}$

Consider the groups $\check{\mathrm{Q}}_{p,q,r}^{\overline{1}}, \check{\mathrm{Q}}_{p,q,r}^{\overline{2}}, \check{\mathrm{Q}}_{p,q,r}^{\overline{3}}$, and $\check{\mathrm{Q}}_{p,q,r}^{\overline{0}}$:

$$
\begin{align}
\check{\mathrm{Q}}_{p,q,r}^{\overline{1}} &:= \{T \in C\ell_{p,q,r}^{\times} : \quad \widetilde{T}T \in \check{\mathrm{Z}}_{p,q,r}^{1\times} = \ker(\tilde{\mathrm{ad}}), \quad \widehat{\widetilde{T}}T \in \check{\mathrm{Z}}_{p,q,r}^{1\times} = \ker(\tilde{\mathrm{ad}})\}, \tag{133} \\
\check{\mathrm{Q}}_{p,q,r}^{\overline{2}} &:= \{T \in C\ell_{p,q,r}^{\times} : \quad \widetilde{T}T \in \check{\mathrm{Z}}_{p,q,r}^{2\times}, \quad \widehat{\widetilde{T}}T \in \check{\mathrm{Z}}_{p,q,r}^{2\times}\}, \tag{134} \\
\check{\mathrm{Q}}_{p,q,r}^{\overline{3}} &:= \{T \in C\ell_{p,q,r}^{\times} : \quad \widetilde{T}T \in \check{\mathrm{Z}}_{p,q,r}^{3\times}, \quad \widehat{\widetilde{T}}T \in \check{\mathrm{Z}}_{p,q,r}^{3\times}\}, \tag{135} \\
\check{\mathrm{Q}}_{p,q,r}^{\overline{0}} &:= \{T \in C\ell_{p,q,r}^{\times} : \quad \widetilde{T}T \in \langle \mathrm{Z}_{p,q,r}^{4}\rangle_{(0)}^{\times}, \quad \widehat{\widetilde{T}}T \in \langle \mathrm{Z}_{p,q,r}^{4}\rangle_{(0)}^{\times}\}, \tag{136}
\end{align}
$$

where $\ker(\tilde{\mathrm{ad}})$ (109) is the kernel of the twisted adjoint representation $\tilde{\mathrm{ad}}$ (104), the sets $\check{\mathrm{Z}}_{p,q,r}^{2}$ (79) and $\check{\mathrm{Z}}_{p,q,r}^{3}$ (80) are the twisted centralizers of the subspaces $C\ell_{p,q,r}^{2}$ and $C\ell_{p,q,r}^{3}$ respectively, and $\mathrm{Z}_{p,q,r}^{4}$ (Remark D.1) is the centralizer of the subspace $C\ell_{p,q,r}^{4}$ (see Appendix D). We use $\check{\ }$ in the notation of the groups (133)–(136) due to Theorem F.2 below. We have (see (109)):

$$
\langle \mathrm{Z}_{p,q,r}^{4}\rangle_{(0)} = \begin{cases} \Lambda_r^{(0)} \oplus \{C\ell_{p,q}^{1}\Lambda_r^{n-2}\} \oplus \{C\ell_{p,q}^{2}\Lambda_r^{n-3}\}, & n \text{ is odd}; \\ \Lambda_r^{(0)} \oplus \{C\ell_{p,q}^{1}\Lambda_r^{n-3}\} \oplus \{C\ell_{p,q}^{2}\Lambda_r^{n-4}\} \oplus C\ell_{p,q,r}^{n}, & n \text{ is even}, \quad r \neq n; \\ \Lambda_r^{(0)}, & n \text{ is even}, \quad r = n. \end{cases} \tag{137}
$$

In the particular case of the non-degenerate geometric algebra $C\ell_{p,q}$, we obtain the groups $\mathrm{Q}^{\pm}$ and $\mathrm{Q}'$ considered in Filimoshina & Shirokov, 2024a:

$$
\check{\mathrm{Q}}_{p,q}^{\overline{1}} = \check{\mathrm{Q}}_{p,q}^{\overline{3}} = \mathrm{Q}^{\pm}, \qquad \check{\mathrm{Q}}_{p,q}^{\overline{0}} = \check{\mathrm{Q}}_{p,q}^{\overline{2}} = \begin{cases} \mathrm{Q}^{\pm}, & n = 1, 2, 3 \mod 4, \\ \mathrm{Q}', & n = 0 \mod 4, \end{cases} \quad n \neq 1, 2,
$$

and $\check{\mathrm{Q}}_{p,q}^{\overline{1}} = \check{\mathrm{Q}}_{p,q}^{\overline{0}} = \mathrm{Q}^{\pm}$ if $n = 1, 2$.

**Theorem F.2** (Formulas (23)–(24) of Theorem 3.2). *In degenerate and non-degenerate geometric algebras $C\ell_{p,q,r}$, we have*

$$
\check{\mathrm{Q}}_{p,q,r}^{\overline{1}} = \check{\Gamma}_{p,q,r}^{\overline{1}} \subseteq \check{\mathrm{Q}}_{p,q,r}^{\overline{3}} = \check{\Gamma}_{p,q,r}^{\overline{3}}, \qquad \check{\mathrm{Q}}_{p,q,r}^{\overline{2}} = \check{\Gamma}_{p,q,r}^{\overline{2}}, \quad \check{\mathrm{Q}}_{p,q,r}^{\overline{0}} = \check{\Gamma}_{p,q,r}^{\overline{0}}.
$$

*Proof.* We get $\check{\mathrm{Q}}_{p,q,r}^{\overline{1}} \subseteq \check{\mathrm{Q}}_{p,q,r}^{\overline{3}}$, using $\check{\mathrm{Z}}_{p,q,r}^{1} \subseteq \check{\mathrm{Z}}_{p,q,r}^{3}$ (see Remark D.1). For fixed $k = 0, 1, 2, 3 \mod 4$ and $m = (k-1) \mod 4$, $l = (k+1) \mod 4$, we obtain

$$
\begin{align}
\check{\Gamma}_{p,q,r}^{\overline{k}} &= \{T \in C\ell_{p,q,r}^{\times} : \quad \widehat{T} C\ell_{p,q,r}^{\overline{k}} T^{-1} \subseteq (C\ell_{p,q,r}^{\overline{km}} \cap C\ell_{p,q,r}^{\overline{kl}})\} \tag{138} \\
&= \{T \in C\ell_{p,q,r}^{\times} : \quad \widehat{T} C\ell_{p,q,r}^{\overline{k}} T^{-1} \subseteq C\ell_{p,q,r}^{\overline{km}}, \quad \widehat{T} C\ell_{p,q,r}^{\overline{k}} T^{-1} \subseteq C\ell_{p,q,r}^{\overline{kl}}\} \tag{139} \\
&= \{T \in C\ell_{p,q,r}^{\times} : \quad \widetilde{T}T \in \check{\mathrm{Z}}_{p,q,r}^{\overline{k}\times}, \quad \widehat{\widetilde{T}}T \in \check{\mathrm{Z}}_{p,q,r}^{\overline{k}\times}\} = \check{\mathrm{Q}}_{p,q,r}^{\overline{k}}, \tag{140}
\end{align}
$$

where we use Lemma D.5 in the first equality (140) and $\check{\mathrm{Z}}_{p,q,r}^{\overline{k}} = \check{\mathrm{Z}}_{p,q,r}^{k}$, $k = 1, 2, 3$, and $\check{\mathrm{Z}}_{p,q,r}^{\overline{0}} = \check{\mathrm{Z}}_{p,q,r}^{4} \cap C\ell_{p,q,r}^{(0)} = \mathrm{Z}_{p,q,r}^{4} \cap C\ell_{p,q,r}^{(0)}$ by Lemma D.4 and (73) in the second equality (140). $\qquad\square$

*Remark* F.3 (Formula (26) of Theorem 3.2). We have the following relations between the groups $\Gamma_{p,q,r}^{\overline{k}}$ (121) and $\check{\Gamma}_{p,q,r}^{\overline{k}}$ (122), $k = 0, 1, 2, 3$:

$$\check{\Gamma}_{p,q,r}^{\overline{m}} \subseteq \Gamma_{p,q,r}^{\overline{0}}, \qquad m = 0, 1, 2, 3; \qquad \check{\Gamma}_{p,q,r}^{\overline{1}} \subseteq \Gamma_{p,q,r}^{\overline{2}}. \tag{141}$$

The statements follow from Theorems F.1 and F.2, the definitions of the groups $\check{Q}_{p,q,r}^{\overline{m}}$ (133)–(136), $m = 0, 1, 2, 3$, $Q_{p,q,r}^{\overline{0}}$ (127), $Q_{p,q,r}^{\overline{2}}$ (125), $\check{Z}_{p,q,r}^{1} \subseteq Z_{p,q,r}^{2}$, and $\check{Z}_{p,q,r}^{k} \subseteq Z_{p,q,r}^{4}$, $k = 1, 2, 3$, by Remark D.1.

## F.3. The Groups $\tilde{\Gamma}_{p,q,r}^{\overline{0}}$, $\tilde{\Gamma}_{p,q,r}^{\overline{1}}$, $\tilde{\Gamma}_{p,q,r}^{\overline{2}}$, and $\tilde{\Gamma}_{p,q,r}^{\overline{3}}$

In Corollary F.4, we summarize the results on the equivalent definitions of the generalized Lipschitz groups $\tilde{\Gamma}_{p,q,r}^{\overline{k}}$ (120), $k = 0, 1, 2, 3$. They follow from Theorems F.1 and F.2 on the equivalent definitions of the groups $\Gamma_{p,q,r}^{\overline{k}}$ (121) and $\check{\Gamma}_{p,q,r}^{\overline{k}}$ (122), $k = 0, 1, 2, 3$, respectively, Remark F.3, and the relation (123) between these groups and the generalized Lipschitz groups $\tilde{\Gamma}_{p,q,r}^{\overline{k}}$.

**Corollary F.4** (Formula (14) of Theorem 3.1). *In the case of arbitrary $C\ell_{p,q,r}$, we have*

$$\tilde{\Gamma}_{p,q,r}^{\overline{1}} = \check{Q}_{p,q,r}^{\overline{1}} \subseteq \tilde{\Gamma}_{p,q,r}^{\overline{2}} = Q_{p,q,r}^{\overline{2}} \subseteq \tilde{\Gamma}_{p,q,r}^{\overline{0}} = Q_{p,q,r}^{\overline{0}}, \tag{142}$$

$$\tilde{\Gamma}_{p,q,r}^{\overline{1}} \subseteq \tilde{\Gamma}_{p,q,r}^{\overline{3}} = \check{Q}_{p,q,r}^{\overline{3}} \tag{143}$$

**Theorem F.5** (Theorem 3.3). *The elements of the generalized Lipschitz groups $\tilde{\Gamma}_{p,q,r}^{\overline{1}}$ have the following special form:*

$$\tilde{\Gamma}_{p,q,r}^{\overline{1}} \subseteq (C\ell_{p,q,r}^{(0)\times} \cup C\ell_{p,q,r}^{(1)\times})\Lambda_r^\times. \tag{144}$$

*Proof.* Suppose $T \in \tilde{\Gamma}_{p,q,r}^{\overline{1}}$; then $\widetilde{T}T = W \in \ker(\check{\mathrm{ad}})$ and $\widehat{\widetilde{T}}T = U \in \ker(\tilde{\mathrm{ad}})$. Therefore, $T = \widetilde{T^{-1}}W$ and $\widehat{T^{-1}} = \widehat{U^{-1}}\widetilde{T}$. Thus, $\widehat{T^{-1}}T = (\widehat{U^{-1}}\widetilde{T})(\widetilde{T^{-1}}W) = \widehat{U^{-1}}W \in \ker(\tilde{\mathrm{ad}}) = \Lambda_r^\times$. Now we apply Theorem 4.7 (Filimoshina & Shirokov, 2023):

$$(C\ell_{p,q,r}^{(0)\times} \cup C\ell_{p,q,r}^{(1)\times})\Lambda_r^\times = \{T \in C\ell_{p,q,r}^\times : \quad \widehat{T^{-1}}T \in \Lambda_r^\times\}.$$

and obtain $T \in (C\ell_{p,q,r}^{(0)\times} \cup C\ell_{p,q,r}^{(1)\times})\Lambda_r^\times$. □

## F.4. Relation Between Ordinary and Generalized Lipschitz Groups

The group $\tilde{\Gamma}_{p,q,r}^{\overline{1}}$ may be considered the most important among the generalized Lipschitz groups $\tilde{\Gamma}_{p,q,r}^{\overline{k}}$ (120), $k = 0, 1, 2, 3$, because its elements preserve under $\tilde{\mathrm{ad}}$ not only the subspace $C\ell_{p,q,r}^{\overline{1}}$, but also all other subspaces $C\ell_{p,q,r}^{\overline{k}}$, $k = 1, 2, 3$, by Corollary F.4. Moreover, the group $\tilde{\Gamma}_{p,q,r}^{\overline{1}}$ coincides with the ordinary Lipschitz group $\tilde{\Gamma}_{p,q,r}^{1}$ (10) in the case of the low-dimensional geometric algebras $C\ell_{p,q,r}$ (Remark F.6 below) and contains it as a subgroup in the case of arbitrary $C\ell_{p,q,r}$ (Theorem F.7).

*Remark* F.6 (Formula (15) of Theorem 3.1). In the case of the small dimensions $n \leq 4$, the (ordinary) Lipschitz groups (10) coincide with the generalized Lipschitz groups $\tilde{\Gamma}_{p,q,r}^{\overline{1}}$ due to $C\ell_{p,q,r}^{\overline{1}} = C\ell_{p,q,r}^{1}$:

$$\tilde{\Gamma}_{p,q,r}^{1} = \tilde{\Gamma}_{p,q,r}^{\overline{1}}, \qquad n \leq 4. \tag{145}$$

Note that in the particular case of the non-degenerate geometric algebra $C\ell_{p,q}$, it is proved (Filimoshina & Shirokov, 2024a) that the groups coincide in the case $n = 5$ as well:

$$\tilde{\Gamma}_{p,q}^{1} = \tilde{\Gamma}_{p,q}^{\overline{1}}, \qquad n \leq 5. \tag{146}$$

**Theorem F.7** (Formula (14) of Theorem 3.1). *In arbitrary $C\ell_{p,q,r}$, the (ordinary) Lipschitz group $\tilde{\Gamma}_{p,q,r}^{1}$ (10) is a subgroup of the generalized Lipschitz group $\tilde{\Gamma}_{p,q,r}^{\overline{1}}$ (120):*

$$\tilde{\Gamma}_{p,q,r}^{1} \subseteq \tilde{\Gamma}_{p,q,r}^{\overline{1}}. \tag{147}$$

*Proof.* Suppose $T \in \tilde{\Gamma}^1_{p,q,r}$, then $T \in C\ell^\times_{p,q,r}$ and $\widehat{T}U_1T^{-1} \in C\ell^1_{p,q,r}$ for any $U_1 \in C\ell^1_{p,q,r}$ by definition. We get

$$\widehat{T}U_1T^{-1} = (\widehat{T}U_1T^{-1})^\sim = \widetilde{T^{-1}}\widetilde{U_1}\widehat{\widetilde{T}} = \widetilde{T^{-1}}U_1\widehat{\widetilde{T}}, \qquad \forall U_1 \in C\ell^1_{p,q,r}, \tag{148}$$

$$\widehat{T}U_1T^{-1} = -(\widehat{T}U_1T^{-1})^{\widehat{\phantom{i}}} = -\widehat{\widetilde{T^{-1}}}\widehat{\widetilde{U_1}}\widetilde{T} = \widehat{\widetilde{T^{-1}}}U_1\widetilde{T}, \qquad \forall U_1 \in C\ell^1_{p,q,r}, \tag{149}$$

where we use $C\ell^1_{p,q,r} \subseteq C\ell^{\bar{1}}_{p,q,r}$, Table 1, and the properties of the reversion $\widetilde{UV} = \widetilde{V}\widetilde{U}$, Clifford conjugation $\widehat{\widetilde{UV}} = \widehat{\widetilde{V}}\widehat{\widetilde{U}}$, and grade involution $\widehat{\widehat{U}} = U$, for any $U, V \in C\ell_{p,q,r}$. We multiply both sides of the equality (148) on the left by $\widetilde{T}$ and on the right by $T$, both sides of the equality (149) on the left by $\widehat{\widetilde{T}}$ and on the right by $T$, and get

$$\widehat{(\widehat{\widetilde{T}}T)}U_1 = U_1(\widehat{\widetilde{T}}T), \qquad \widehat{(\widetilde{T}T)}U_1 = U_1(\widetilde{T}T), \qquad \forall U_1 \in C\ell^1_{p,q,r}. \tag{150}$$

Therefore, by the definition (72) of the twisted centralizers, the elements $\widehat{\widetilde{T}}T$ and $\widetilde{T}T$ belong to the twisted centralizer of the grade-1 subspace $C\ell^1_{p,q,r}$ in $C\ell_{p,q,r}$:

$$\widehat{\widetilde{T}}T \in \check{Z}^1_{p,q,r}, \qquad \widetilde{T}T \in \check{Z}^1_{p,q,r}. \tag{151}$$

We have $\check{Z}^1_{p,q,r} = \Lambda_r$ by Remark D.1. Therefore, $\widehat{\widetilde{T}}T, \widetilde{T}T \in \Lambda^\times_r$, and $T \in \check{Q}^{\bar{1}}_{p,q,r}$ by definition (133). By Corollary F.4, $\check{Q}^{\bar{1}}_{p,q,r} = \tilde{\Gamma}^{\bar{1}}_{p,q,r}$. Thus, $T \in \tilde{\Gamma}^{\bar{1}}_{p,q,r}$, and the proof is completed. $\square$

# G. Equivariant Mappings

Let $G$ be a group and $X$ be a set. A (left) *group action* is a map: $\circ : G \times X \to X$, $(g, x) \mapsto g \circ x$ that satisfies associativity (i.e. $(gh) \circ x = g \circ (h \circ x)$ for any $g, h \in G$ and $x \in X$) and identity condition (i.e. $e \circ x = x$ for any $x \in X$).

Suppose $G$ is a group and $\circ_X$ and $\circ_Y$ are its actions on the sets $X$ and $Y$ respectively. A *function* (network) $L : X \to Y$ is called *G-equivariant* iff it commutes with these actions:

$$L(g \circ_X x) = g \circ_Y L(x), \qquad \forall g \in G, \qquad \forall x \in X. \tag{152}$$

**Example G.1.** Consider the generalized Lipschitz group $\tilde{\Gamma}^{\bar{1}}_{p,q,r}$ with the action $\tilde{\text{ad}}$ and a function $L : C\ell_{p,q,r} \to C\ell_{p,q,r}$. This function is $\tilde{\Gamma}^{\bar{1}}_{p,q,r}$-equivariant iff

$$L(\tilde{\text{ad}}_T(x)) = \tilde{\text{ad}}_T(L(x)), \qquad \forall T \in \tilde{\Gamma}^{\bar{1}}_{p,q,r}, \qquad \forall x \in C\ell_{p,q,r}, \tag{153}$$

i.e.

$$L(T\langle x\rangle_{(0)}T^{-1} + \widehat{T}\langle x\rangle_{(1)}T^{-1}) = T\langle L(x)\rangle_{(0)}T^{-1} + \widehat{T}\langle L(x)\rangle_{(1)}T^{-1}, \qquad \forall T \in \tilde{\Gamma}^{\bar{1}}_{p,q,r}, \qquad \forall x \in C\ell_{p,q,r}. \tag{154}$$

We prove several general statements about equivariance with respect to an arbitrary group. Suppose $G$ is a group, $\circ_X$ is its action defined on a set $X$.

**Lemma G.2.** *Consider $k$ G-equivariant mappings $f_i : X_{i-1} \to X_i$, $i = 1, \ldots, k$. Their composition is G-equivariant:*

$$f_k(\cdots(f_2(f_1(g \circ_{X_0} x)))) = g \circ_{X_k} f_k(\cdots(f_2(f_1(x)))) \tag{155}$$

*for any $g \in G$ and $x \in X_0$.*

*Proof.* We prove the statement by induction, using

$$f_2(f_1(g \circ_{X_0} x)) = f_2(g \circ_{X_1} (f_1(x)) = g \circ_{X_2} f_2(f_1(x)).$$

$\square$

**Lemma G.3.** *Suppose group $G$ action $\circ$ is linear, i.e. for any $g \in G$, $x, y \in X$, and $\alpha \in \mathbb{F}$*

$$g \circ_Y (\alpha x) = \alpha(g \circ_X x), \quad g \circ_Y (x + y) = g \circ_X x + g \circ_X y.$$

*Consider $k$ $G$-equivariant mappings $f_1, \ldots, f_k : X \to Y$. Their linear combinations are $G$-equivariant:*

$$\sum_{i=1}^{k} \alpha_i f_i(g \circ_X x) = g \circ_Y \left( \sum_{i=1}^{k} \alpha_i f_i(x) \right) \tag{156}$$

*for any $g \in G$, $x \in X$, $\alpha_i \in \mathbb{F}$.*

*Proof.* We have

$$\alpha_1 f_1(g \circ_X x) + \alpha_2 f_2(g \circ_X x) = \alpha_1(g \circ_Y f_1(x)) + \alpha_2(g \circ_Y f_2(x)) = g \circ_Y (\alpha_1 f_1(x) + \alpha_2 f_2(x)) \tag{157}$$

and obtain the result similarly for $k$ summands. $\qquad\square$

**Lemma G.4.** *Suppose group $G$ action $\circ$ is multiplicative, i.e.*

$$g \circ_Y (xy) = (g \circ_X x)(g \circ_X y), \qquad \forall g \in G, \quad x, y \in X.$$

*The product of two $G$-equivariant mappings $f_1, f_2 : X \to Y$ is $G$-equivariant:*

$$g \circ_Y (f_1(x)f_2(x)) = f_1(g \circ_X x)f_2(g \circ_X x) \tag{158}$$

*for any $g \in G$ and $x \in X$.*

*Proof.* We have

$$g \circ_Y (f_1(x)f_2(x)) = (g \circ_Y f_1(x))(g \circ_Y f_2(x)) = f_1(g \circ_X x)f_2(g \circ_X x). \tag{159}$$

$\qquad\square$

# H. Degenerate and Non-degenerate Lipschitz, Generalized Lipschitz, and Orthogonal Groups Equivariance

In this section, we consider the relation between the degenerate and non-degenerate pseudo-orthogonal (complex orthogonal) groups and the ordinary and generalized Lipschitz groups. We extend the work presented in Crumeyrolle, 1990; Brooke, 1980; Dereli et al., 2010; Ruhe et al., 2023, etc.

Consider the degenerate and non-degenerate *pseudo-orthogonal group* (in the real case $V = \mathbb{R}^{p,q,r}$) or *complex orthogonal group* (in the complex case $V = \mathbb{C}^{p+q,0,r}$) denoted by $\mathrm{O}(V, \mathfrak{q})$. It is defined as the Lie group of all linear transformations of an $n$-dimensional vector space $V$ that leave invariant a quadratic form $\mathfrak{q}$ of signature $(p, q, r)$ if $V$ is real and $(p + q, 0, r)$ if $V$ is complex:

$$\mathrm{O}(V, \mathfrak{q}) := \{\Phi : V \to V : \quad \Phi \text{ is linear, invertible}, \quad \mathfrak{q}(\Phi(v)) = \mathfrak{q}(v), \quad \forall v \in V\}. \tag{160}$$

Note that

$$\mathrm{O}(V, \mathfrak{q}) \cong \{A \in \mathrm{GL}(n, \mathbb{F}) : \quad A^{\mathrm{T}} \eta A = \eta\}.$$

When considering both the real and complex cases, we refer to the group $\mathrm{O}(V, \mathfrak{q})$ as the orthogonal group.

Let us use the following notation. The *radical subspace* is denoted by $\Lambda_r^1 := C\ell_{0,0,r}^1$. We have $V = C\ell_{p,q,r}^1 = C\ell_{p,q}^1 \oplus \Lambda_r^1$. The subalgebra generated by the basis elements of $\Lambda_r^1$ is denoted by $\Lambda_r$ and is a *Grassmann (exterior) algebra*.

Consider the following subgroup of the orthogonal group, which leaves invariant the radical subspace $\Lambda_r^1$:

$$\mathrm{O}_{\Lambda_r^1}(V, \mathfrak{q}) := \{\Phi \in \mathrm{O}(V, q) : \quad \Phi|_{\Lambda_r^1} = \mathrm{id}_{\Lambda_r^1}\}. \tag{161}$$

The following statement about the matrix forms of the orthogonal $\mathrm{O}(V, \mathfrak{q})$ and restricted orthogonal $\mathrm{O}_{\Lambda_r^1}(V, \mathfrak{q})$ groups is well-known and considered, for example, in Crumeyrolle, 1990; Ruhe et al., 2023. We use it in the proof of the main statements about the relation between the Lipschitz and orthogonal groups below.

**Lemma H.1.** *We have the following isomorphisms*

$$O(V, \mathfrak{q}) \cong \left\{ \begin{pmatrix} A & 0_{(p+q)\times r} \\ M & G \end{pmatrix} \right\}, \qquad A \in O(C\ell^1_{p,q}, \mathfrak{q}|_{C\ell^1_{p,q}}), \qquad M \in \mathrm{Mat}_{r\times(p+q)}(\mathbb{F}), \qquad G \in \mathrm{GL}(r, \mathbb{F}); \qquad (162)$$

$$O_{\Lambda^1_r}(V, \mathfrak{q}) \cong \left\{ \begin{pmatrix} A & 0_{(p+q)\times r} \\ M & \mathrm{I}_r \end{pmatrix} \right\}, \qquad A \in O(C\ell^1_{p,q}, \mathfrak{q}|_{C\ell^1_{p,q}}), \qquad M \in \mathrm{Mat}_{r\times(p+q)}(\mathbb{F}), \qquad (163)$$

*where $O(C\ell^1_{p,q}, \mathfrak{q}|_{C\ell^1_{p,q}})$ is the non-degenerate orthogonal group of transformations of $C\ell^1_{p,q}$ with a non-degenerate quadratic form $\mathfrak{q}|_{C\ell^1_{p,q}}$; $0_{(p+q)\times r}$ is the zero matrix of size $(p+q) \times r$; $\mathrm{Mat}_{r\times(p+q)}(\mathbb{F})$ is a set of arbitrary matrices of size $r \times (p+q)$ with coefficients in $\mathbb{F}$; $\mathrm{I}_r$ is the identity matrix of size $r \times r$; $\mathrm{GL}(r, \mathbb{F})$ is the general linear group acting on an $r$-dimensional vector space over $\mathbb{F}$.*

**Theorem H.2** (Kernel of $\tilde{\mathrm{ad}}^1$). *The kernel of the restricted twisted adjoint representation $\tilde{\mathrm{ad}}^1$ acting on the Lipschitz group $\tilde{\mathrm{ad}}^1 : \tilde{\Gamma}^1_{p,q,r} \to \mathrm{Aut}(C\ell_{p,q,r})$ has the following form:*

$$\ker(\tilde{\mathrm{ad}}^1 : \tilde{\Gamma}^1_{p,q,r} \to \mathrm{Aut}(C\ell_{p,q,r})) = \Lambda^\times_r. \qquad (164)$$

*Proof.* We have

$$\ker(\tilde{\mathrm{ad}}^1 : \tilde{\Gamma}^1_{p,q,r} \to \mathrm{Aut}(C\ell_{p,q,r})) = \{T \in \tilde{\Gamma}^1_{p,q,r} : \quad \widehat{T}vT^{-1} = v, \quad \forall v \in C\ell^1_{p,q,r}\} \qquad (165)$$

$$= \tilde{\Gamma}^1_{p,q,r} \cap \{T \in C\ell^\times_{p,q,r} : \quad \widehat{T}vT^{-1} = v, \quad \forall v \in C\ell^1_{p,q,r}\} \qquad (166)$$

$$= \tilde{\Gamma}^1_{p,q,r} \cap \Lambda^\times_r, \qquad (167)$$

where in (167), we apply the statement (109) of Lemma E.1. Note that $\Lambda^\times_r \subseteq \tilde{\Gamma}^1_{p,q,r}$, since for any $T \in \Lambda^\times_r$ and any $v \in C\ell^1_{p,q,r}$, we have $\widehat{T}vT^{-1} = vTT^{-1} = v \in C\ell^1_{p,q,r}$ again by Lemma E.1. Thus, we obtain (164).

$\square$

*Remark* H.3. Note that for any vectors $v, v_1, v_2 \in V$, we have

$$2\mathfrak{b}(v_1, v_2)e = v_1 v_2 + v_2 v_1, \qquad \mathfrak{q}(v)e = v^2. \qquad (168)$$

In Remark H.4, we consider how reflections can be represented in geometric algebras. We use this particular case of the relation between elements of the orthogonal groups $O(V, \mathfrak{q})$ and the Lipschitz groups $\tilde{\Gamma}^1_{p,q,r}$ in the proof of Theorem H.9 below.

*Remark* H.4. The mapping $\tilde{\mathrm{ad}}^1_v$, where $v \in C\ell^{1\times}_{p,q}$ is an invertible vector, acts on an arbitrary vector $x \in C\ell^1_{p,q,r}$ as a reflection of a vector $x$ across the hyperplane orthogonal to the vector $v$:

$$\tilde{\mathrm{ad}}^1_v(x) = \widehat{v}xv^{-1} = -vxv^{-1} = x - (xv + vx)v^{-1} = x - 2\mathfrak{b}(x, v)\frac{v^2}{\mathfrak{q}(v)}v^{-1} = x - 2\frac{\mathfrak{b}(x, v)}{\mathfrak{b}(v, v)}v, \qquad (169)$$

where in the right-hand side, we have the difference between the vector $x$ and twice the projection of the vector $x$ onto the vector $v$.

In Remarks H.5 and H.6, we consider several examples of elements that belong to the Lipschitz group $\tilde{\Gamma}^1_{p,q,r}$ (10).

*Remark* H.5. All invertible non-degenerate vectors belong to the Lipschitz group:

$$C\ell^{1\times}_{p,q} \subseteq \tilde{\Gamma}^1_{p,q,r}, \qquad (170)$$

since, by Remark H.4, we have $\tilde{\mathrm{ad}}^1_v(x) = x - 2\frac{\mathfrak{b}(x,v)}{\mathfrak{b}(v,v)}v \in C\ell^1_{p,q,r}$ for any $x \in C\ell^1_{p,q,r}$ and $v \in C\ell^{1\times}_{p,q}$.

*Remark* H.6. Any element $U \in C\ell_{p,q,r}$ of the form

$$U = e + me_{ab} \in C\ell^0 \oplus C\ell^2_{p,q,r}, \qquad \forall m \in \mathbb{F}, \quad \forall e_a \in C\ell^1_{p,q}, \quad \forall e_b \in \Lambda^1_r, \qquad (171)$$

is invertible and belongs to the Lipschitz group:

$$U \in \tilde{\Gamma}^1_{p,qr}. \tag{172}$$

Let us prove these statement. Firstly, $U$ is invertible, since $(e + me_{ab})(e - me_{ab}) = e$. Secondly $U$ preserves the grade-1 subspace under $\tilde{\text{ad}}$, since

$$\widehat{U}e_a U^{-1} = (e + me_{ab})e_a(e - me_{ab}) = (e_a - m\eta_{aa}e_b)(e - me_{ab}) = e_a - 2m\eta_{aa}e_b \in C\ell^1_{p,q,r}, \tag{173}$$

$$\widehat{U}e_b U^{-1} = (e + me_{ab})e_b(e - me_{ab}) = e_b \in C\ell^1_{p,q,r}, \tag{174}$$

$$\widehat{U}e_c U^{-1} = (e + me_{ab})e_c(e - me_{ab}) = (e_c + me_{ab}e_c)(e - me_{ab}) = e_c \in C\ell^1_{p,q,r}, \quad \forall c = 1, \ldots, n, \quad c \neq a, b. \tag{175}$$

Thus, by definition of the Lipschitz group $\tilde{\Gamma}^1_{p,q,r}$ (10), we get (172).

Let us prove auxiliary Lemmas H.7 and H.8, which we use in the proof of Theorem H.9.

**Lemma H.7.** *In the case $r \neq 0$ and $r \neq n$, consider any matrix of the form*

$$B_{i,j} = \begin{pmatrix} I_{p+q} & 0 \\ M_{i,j} & I_r \end{pmatrix} \in O_{\Lambda^1_r}(V, \mathfrak{q}), \tag{176}$$

*where $I_{p+q}$ and $I_r$ are the identity matrices of the sizes $(p+q) \times (p+q)$ and $r \times r$ respectively, and $M_{i,j} \in \text{Mat}_{r,p+q}(\mathbb{F})$ has at most one non-zero element $m_{ij}$, which is in the $i$-th row and $j$-th column, for fixed $i = 1, \ldots, r$, $j = 1, \ldots, p+q$. Then for the following multivector*

$$U_{i,j} := e + c_{ij}e_j e_{p+q+i} \in (C\ell^0 \oplus C\ell^2_{p,q,r})^\times, \qquad c_{ij} := -\frac{m_{ij}}{2\eta_{jj}} \in \mathbb{F}, \qquad e_j \in C\ell^1_{p,q}, \qquad e_{p+q+i} \in \Lambda^1_r,$$

*we have*

$$B_{i,j} = \tilde{\text{ad}}^1_{U_{i,j}}. \tag{177}$$

*In the cases $r = n$ or $r = 0$, for the identity matrices $I_r$ or $I_{p+q}$ respectively, we have $I_r = \tilde{\text{ad}}^1_e$ and $I_{p+q} = \tilde{\text{ad}}^1_e$ respectively.*

*Proof.* In the cases $r = n$ or $r = 0$, we have $\tilde{\text{ad}}^1_e(v) = eve^{-1} = v$ for any $v \in C\ell^1_{p,q,r}$ and get the statement.

Further in the proof, we consider the case $r \neq 0$ and $r \neq n$. Without loss of generality, let us prove the statement for $B_{1,1}$. Since $B_{1,1} \in O_{\Lambda^1_r}(V, \mathfrak{q})$ (163), it leaves invariant $\Lambda^1_r$. Thus, $B_{1,1}v = \text{id}v$ for any $v \in \Lambda^1_r$. On the other hand, for $U_{1,1} := e + c_{11}e_1 e_{p+q+1}$, we have $\tilde{\text{ad}}^1_{U_{1,1}}(v) = U_{1,1}vU_{1,1}^{-1} = vU_{1,1}U_{1,1}^{-1} = v$, for any $c_{ij} \in \mathbb{F}$, where we apply the statement (84) of Lemma D.3, since $U_{1,1} \in C\ell^{(0)}_{p,q,r}$ and $v \in \Lambda^1_r$.

Let us consider how $B_{1,1}$ acts on the vectors from a canonical basis of $C\ell^1_{p,q}$.

For the vector $v_1 = (1, 0, \ldots, 0) \in V$ corresponding to $e_1$, we have

$$B_{1,1}v_1 = (1, 0, \ldots, 0, m_{11}, 0, \ldots, 0), \tag{178}$$

where $m_{11}$ is on the $(p+q+1)$-th position. On the other hand, for $U_{1,1}$ and $e_1$, we get

$$\tilde{\text{ad}}^1_{U_{1,1}}(e_1) = (e_1 - c_{11}\eta_{11}e_{p+q+1})(e - c_{11}e_1 e_{p+q+1}) = e_1 - 2\eta_{11}c_{11}e_{p+q+1} = e_1 + m_{11}e_{p+q+1}, \tag{179}$$

which corresponds to (178).

For any other vector $v_k \in V$, $k = 2, \ldots, p+q$, corresponding to $e_k$, we have

$$B_{1,1}v_k = (0, \ldots, 0, 1, 0, \ldots, 0), \tag{180}$$

where 1 is on the $k$-th position. On the other hand, for $U_{1,1}$ and $e_k$, we get

$$\tilde{\text{ad}}^1_{U_{1,1}}(e_k) = (e + c_{11}e_1 e_{p+q+1})e_k(e - c_{11}e_1 e_{p+q+1}) = e_k, \tag{181}$$

which corresponds to (180). We have proved that $B_{1,1}v = \tilde{\text{ad}}^1_{U_{1,1}}(v)$ for any $v$ from a canonical basis of $V$. By linearity, we get the statement (177). This completes the proof. $\qquad\square$

**Lemma H.8.** *We have*

$$\widehat{X}V = VX, \qquad \forall X \in (Cl_{p,q,r}^{(0)\times} \cup Cl_{p,q,r}^{(1)\times})\Lambda_r^{\times}, \qquad \forall V \in \Lambda_r^{1\times}. \tag{182}$$

*Proof.* Suppose $V \in \Lambda_r^{1\times}$ and $X = WH$, where $W \in (Cl_{p,q,r}^{(0)\times} \cup Cl_{p,q,r}^{(1)\times})$ and $H \in \Lambda_r^{\times}$.

Note that $\widehat{H}V = VH$ by the statement (109) of Lemma E.1.

If $W \in Cl_{p,q,r}^{(0)\times}$, then we have $\widehat{W}V = WV = VW$ by the statement (84) of Lemma D.3.

Consider the case $W \in Cl_{p,q,r}^{(1)\times}$. If $r = n$, then $W \in \Lambda_r^{(1)\times}$, and we get $\widehat{W}V = VW$ again by the statement (109) of Lemma E.1. If $r \neq n$, then there exists such generator $e_i$ that $(e_i)^2 \neq 0$, and we can always represent $W$ as $W = e_i(\eta_{ii}e_iW)$. Since $\eta_{ii}e_iW \in Cl_{p,q,r}^{(0)}$, we again get $(\widehat{\eta_{ii}e_iW})V = V(\eta_{ii}e_iW)$ by the statement (84) of Lemma D.3. Also we have $\widehat{e_i}V = -e_iV = Ve_i$, since $V \in \Lambda_r^1$ does not contain $e_i$. So, in this case, we get $\widehat{W}V = \widehat{e_i}(\widehat{\eta_{ii}e_iW})V = \widehat{e_i}V\eta_{ii}e_iW = Ve_i\eta_{ii}e_iW = VW$ as well.

Using the notes above, we finally get

$$\widehat{X}V = \widehat{W}\widehat{H}V = \widehat{W}VH = VWH = VX, \tag{183}$$

and the statement is proved. $\qquad \square$

**Theorem H.9** (Image of $\tilde{\mathrm{ad}}^1$). *The image of the restricted twisted adjoint representation $\tilde{\mathrm{ad}}^1$ acting on the Lipschitz group $\tilde{\mathrm{ad}}^1 : \tilde{\Gamma}_{p,q,r}^1 \to \mathrm{Aut}(Cl_{p,q,r})$ has the form:*

$$\mathrm{im}(\tilde{\mathrm{ad}}^1 : \tilde{\Gamma}_{p,q,r}^1 \to \mathrm{Aut}(Cl_{p,q,r})) = \mathrm{O}_{\Lambda_r^1}(V, \mathfrak{q}). \tag{184}$$

*Proof.* Firstly, we prove that $\mathrm{im}(\tilde{\mathrm{ad}}^1 : \tilde{\Gamma}_{p,q,r}^1 \to \mathrm{Aut}(Cl_{p,q,r})) \subseteq \mathrm{O}_{\Lambda_r^1}(V, \mathfrak{q})$, i.e. that $\tilde{\mathrm{ad}}_T^1 \in \mathrm{O}_{\Lambda_r^1}(V, \mathfrak{q})$ for any $T \in \tilde{\Gamma}_{p,q,r}^1$. Suppose $T \in \tilde{\Gamma}_{p,q,r}^1$, then $\tilde{\mathrm{ad}}_T^1$ is linear by Lemma 2.1. Moreover, $\tilde{\mathrm{ad}}_T^1$ is invertible with the inverse $\tilde{\mathrm{ad}}_{T^{-1}}^1$, since

$$\tilde{\mathrm{ad}}_{T^{-1}}^1(\tilde{\mathrm{ad}}_T^1(v)) = \widehat{T^{-1}}(\widehat{T}vT^{-1})T = v, \qquad \forall v \in Cl_{p,q,r}^1. \tag{185}$$

Now, we need to show that

$$\mathfrak{q}(\tilde{\mathrm{ad}}_T^1(v)) = \mathfrak{q}(v), \qquad \forall v \in Cl_{p,q,r}^1. \tag{186}$$

We have

$$\mathfrak{q}(\tilde{\mathrm{ad}}_T^1(v)) = \mathfrak{q}(\tilde{\mathrm{ad}}_T(v)) = \tilde{\mathrm{ad}}_T(v)\tilde{\mathrm{ad}}_T(v) = \tilde{\mathrm{ad}}_T(v^2), \tag{187}$$

where we apply Remark H.3, since $\tilde{\mathrm{ad}}_T(v) \in V$, and multiplicativity of $\tilde{\mathrm{ad}}_T$ (Lemma 2.1), since $T \in \tilde{\Gamma}_{p,q,r}^1 \subseteq (Cl_{p,q,r}^{(0)\times} \cup Cl_{p,q,r}^{(1)\times})\Lambda_r^{\times}$ by Theorem 3.3. Since $v^2 = \mathfrak{q}(v)$ again by Remark H.3, we finally get

$$\mathfrak{q}(\tilde{\mathrm{ad}}_T^1(v)) = \tilde{\mathrm{ad}}_T(\mathfrak{q}(v)) = \mathfrak{q}(v), \tag{188}$$

where in the last equality we use the property that $\tilde{\mathrm{ad}}_T$ leaves invariant the scalars by the statement (8) of Lemma 2.1. The statement (186) is proved, i.e. we have shown that $\tilde{\mathrm{ad}}_T^1 \in \mathrm{O}(V, \mathfrak{q})$. To prove $\tilde{\mathrm{ad}}_T^1 \in \mathrm{O}_{\Lambda_r^1}(V, \mathfrak{q})$, we finally need to show that $\tilde{\mathrm{ad}}_T^1|_{\Lambda_r^1} = \mathrm{id}_{\Lambda_r^1}$. This statement is true because for any $v \in \Lambda_r^1$ and $T \in \tilde{\Gamma}_{p,q,r}^1 \subseteq (Cl_{p,q,r}^{(0)\times} \cup Cl_{p,q,r}^{(1)\times})\Lambda_r^{\times}$ (Theorem 3.3), we have $\tilde{\mathrm{ad}}_T^1(v) = \widehat{T}vT^{-1} = vTT^{-1} = v$ by Lemma H.8.

Now let us prove $\mathrm{O}_{\Lambda_r^1}(V, \mathfrak{q}) \subseteq \mathrm{im}(\tilde{\mathrm{ad}}^1 : \tilde{\Gamma}_{p,q,r}^1 \to \mathrm{Aut}(Cl_{p,q,r}))$, i.e. let is prove the surjectivity of the mapping $\tilde{\mathrm{ad}}^1$. We have by Lemma H.1:

$$\mathrm{O}_{\Lambda_r^1}(V, \mathfrak{q}) \cong \left\{ \begin{pmatrix} A & 0 \\ M & \mathrm{I}_r \end{pmatrix} \right\}, \qquad A \in \mathrm{O}(Cl_{p,q}^1, \mathfrak{q}|_{Cl_{p,q}^1}), \qquad M \in \mathrm{Mat}_{r \times (p+q)}(\mathbb{F}). \tag{189}$$

Suppose $\Phi \in \mathrm{O}_{\Lambda_r^1}(V, \mathfrak{q})$. Then, there exist $A \in \mathrm{O}(C\ell_{p,q}^1, \mathfrak{q}|_{C\ell_{p,q}^1})$ and $M \in \mathrm{Mat}_{r \times (p+q)}(\mathbb{F})$ such that

$$\Phi = \begin{pmatrix} A & 0 \\ M & I_r \end{pmatrix} = \begin{pmatrix} A & 0 \\ 0 & I_r \end{pmatrix} \begin{pmatrix} I_{p+q} & 0 \\ M & I_r \end{pmatrix} = \begin{pmatrix} A_1 & 0 \\ 0 & I_r \end{pmatrix} \cdots \begin{pmatrix} A_k & 0 \\ 0 & I_r \end{pmatrix} \begin{pmatrix} I_{p+q} & 0 \\ M_{1,1} & I_r \end{pmatrix} \cdots \begin{pmatrix} I_{p+q} & 0 \\ M_{r,p+q} & I_r \end{pmatrix}, \tag{190}$$

where $A = A_1 \cdots A_k$, $k \leq p + q$, and $A_1, \ldots, A_k \in \mathrm{O}(C\ell_{p,q}^1, \mathfrak{q}|_{C\ell_{p,q}^1})$ are matrices representing reflections, by the well-known Cartan–Dieudonné theorem (Dieudonné, 1971), which states that every orthogonal transformation in a $(p + q)$-dimensional space with a non-degenerate symmetric bilinear form can be represented as a composition of at most $p + q$ reflections; and $M_{i,j} \in \mathrm{Mat}_{r \times (p+q)}(\mathbb{F})$, $i = 1, \ldots, r$, $j = 1, \ldots, p + q$ is a matrix with at most one non-zero element, which is on the $i$-th row and $j$-th column, such that $M = \sum_{i=1,\ldots,r} \sum_{j=1,\ldots,p+q} M_{i,j}$.

Note that each reflection matrix $A_i$, $i = 1, \ldots, k$, can be associated with some vector $v_i \in C\ell_{p,q}^1$, so that $A_i = \tilde{\mathrm{ad}}_{v_i}^1$, by Remark H.4. Any matrix

$$B_{i,j} = \begin{pmatrix} I_{p+q} & 0 \\ M_{i,j} & I_r \end{pmatrix} \tag{191}$$

can be associated with some multivector $\gamma_{i,j} = e + m_{i,j} e_j e_{p+q+i} \in C\ell^0 \oplus C\ell_{p,q,r}^2$, where $m_{i,j} \in \mathbb{F}$, $e_j \in C\ell_{p,q}^1$, and $e_{p+q+i} \in \Lambda_r^1$, so that $B_{i,j} = \tilde{\mathrm{ad}}_{\gamma_{i,j}}^1$, by Lemma H.7. Consider the following multivector, which consists of $k + p + q$ factors:

$$T = v_1 \cdots v_k \gamma_{1,1} \cdots \gamma_{r,p+q}. \tag{192}$$

Note that $T \in \tilde{\Gamma}_{p,q,r}^1$, since $v_1, \ldots, v_k \in \tilde{\Gamma}_{p,q,r}^1$ by Remark H.5 and $\gamma_{1,1}, \ldots, \gamma_{r,p+q} \in \tilde{\Gamma}_{p,q,r}^1$ by Remark H.6. We obtain for any $v \in V = C\ell_{p,q,r}^1$:

$$\Phi(v) = \begin{pmatrix} A_1 & 0 \\ 0 & I_r \end{pmatrix} \cdots \begin{pmatrix} A_k & 0 \\ 0 & I_r \end{pmatrix} \begin{pmatrix} I_{p+q} & 0 \\ M_{1,1} & I_r \end{pmatrix} \cdots \begin{pmatrix} I_{p+q} & 0 \\ M_{r,p+q} & I_r \end{pmatrix} v \tag{193}$$

$$= \tilde{\mathrm{ad}}_{v_1}^1 \left( \tilde{\mathrm{ad}}_{v_k}^1 \left( \tilde{\mathrm{ad}}_{\gamma_{1,1}}^1 \left( \cdots (\tilde{\mathrm{ad}}_{\gamma_{r,p+q}}^1 (v)) \right) \right) \right) \tag{194}$$

$$= \widehat{v_1} \cdots (\widehat{v_k} (\widehat{\gamma_{1,1}} \cdots (\widehat{\gamma_{r,p+q}} v \gamma_{r,p+q}^{-1}) \cdots \gamma_{1,1}^{-1}) v_k^{-1}) \cdots v_1^{-1}, \tag{195}$$

$$= (v_1 \cdots v_k \widehat{\gamma_{1,1}} \cdots \gamma_{r,p+q}) v (v_1 \cdots v_k \gamma_{1,1} \cdots \gamma_{r,p+q})^{-1} \tag{196}$$

$$= \tilde{\mathrm{ad}}_{v_1 \cdots v_k \gamma_{1,1} \cdots \gamma_{r,p+q}}^1 (v) \tag{197}$$

$$= \tilde{\mathrm{ad}}_T^1 (v). \tag{198}$$

Thus, for any $\Phi \in \mathrm{O}_{\Lambda_r^1}(V, \mathfrak{q})$, there exists $T \in \tilde{\Gamma}_{p,q,r}^1$ such that $\Phi = \tilde{\mathrm{ad}}_T^1$, and this completes the proof. $\square$

**Theorem H.10.** *In the case of any degenerate or non-degenerate $C\ell_{p,q,r}$, the mapping $\tilde{\mathrm{ad}}^1$ defines the following isomorphism:*

$$\tilde{\mathrm{ad}}^1 : \quad \tilde{\Gamma}_{p,q,r}^1 \Big/ \Lambda_r^\times \cong \mathrm{O}_{\Lambda_r^1}(V, \mathfrak{q}), \tag{199}$$

*Proof.* We have the homomorphism $\tilde{\mathrm{ad}}^1$ of the groups $\tilde{\Gamma}_{p,q,r}^1$ and $\mathrm{O}_{\Lambda_r^1}(V, \mathfrak{q})$, which is surjective by the statement (184) of Theorem H.9. By the fundamental homomorphism theorem, the group $\mathrm{O}_{\Lambda_r^1}(V, \mathfrak{q})$ is isomorphic to the quotient group $\tilde{\Gamma}_{p,q,r}^1 \Big/ \ker(\tilde{\mathrm{ad}}^1)$. By applying Theorem H.2, we complete the proof. $\square$

Note that in the particular case of the non-degenerate geometric algebra $C\ell_{p,q}$, Theorem H.10 has the form

$$\tilde{\mathrm{ad}}^1 : \quad \tilde{\Gamma}_{p,q}^1 / C\ell^{0\times} \cong \mathrm{O}(V, \mathfrak{q}) \tag{200}$$

and is well-known (see, for example, Benn & Tucker, 1987).

**Theorem H.11** (Theorem 3.4). *If a mapping $f : C\ell_{p,q,r} \to C\ell_{p,q,r}$ is equivariant with respect to any group* H *that contains the Lipschitz group $\tilde{\Gamma}^1_{p,q,r}$ as a subgroup, then $f$ is equivariant with respect to the corresponding restricted orthogonal group. In other words, if*

$$\tilde{\mathrm{ad}}_T(f(x)) = f(\tilde{\mathrm{ad}}_T(x)), \quad \forall T \in \mathrm{H}, \quad \forall x \in C\ell_{p,q,r},$$

*then*

$$f(\Phi(x)) = \Phi(f(x)), \quad \forall \Phi \in \mathrm{O}_{\Lambda^1_r}(V, \mathfrak{q}), \quad \forall x \in C\ell_{p,q,r},$$

*where $\Phi$ acts on $x$ in a sense* (34) *by applying an orthogonal transformation to its vector components.*

*Proof.* Suppose H is a group, $\tilde{\Gamma}^1_{p,q,r} \subseteq \mathrm{H}$, and $\tilde{\mathrm{ad}}_T(f(x)) = f(\tilde{\mathrm{ad}}_T(x))$ for some mapping $f$, for any $T \in \mathrm{H}$, and $x \in C\ell_{p,q,r}$. Then it holds, in particular, for any $T \in \tilde{\Gamma}^1_{p,q,r}$, and $f$ is $\tilde{\Gamma}^1_{p,q,r}$-equivariant. By Theorem H.10, a mapping is equivariant w.r.t. $\tilde{\Gamma}^1_{p,q,r}$ iff it is equivariant w.r.t. $\mathrm{O}_{\Lambda^1_r}(V, \mathfrak{q})$. Therefore, $f$ is $\mathrm{O}(V, \mathfrak{q})_{\Lambda^1_r}$-equivariant, and the statement is proved. $\square$

# I. Experimental Details

The implementation of GLGENN and summary of each experiment are available at https://github.com/katyafilimoshina/glgenn.

For CGENN (Ruhe et al., 2023), we follow the training setups from the corresponding public code release. For other models, we use the loss values from the corresponding code repository (Finzi et al., 2021).

We construct our models to closely resemble the CGENN architecture, replacing the $C\ell^k_{p,q}$-linear, $C\ell^k_{p,q}$-geometric product, and $C\ell^k_{p,q}$-normalization layers from Ruhe et al., 2023 with the same number of GLGENN's $C\ell^{\overline{k}}_{p,q}$ counterparts (see Section 4). This design choice results in GLGENN having significantly fewer parameters than CGENN, as they operate in a unified manner across 4 fundamental subspaces of geometric algebras defined by the grade involution ($\widehat{\phantom{a}}$) and reversion ($\widetilde{\phantom{a}}$); they processes geometric objects in groups with a step size of 4.

However, this parameter efficiency difference between GLGENN and CGENN becomes apparent only for tasks with $n > 3$. In lower-dimensional cases ($n \leq 3$), subspaces of fixed grades coincide with the subspaces determined by grade involution and reversion (4), i.e., $C\ell^k_{p,q,r} = C\ell^{\overline{k}}_{p,q,r}$ for $k = 0, 1, 2, 3$. Nonetheless, as experiments show, even in 5-dimensional cases, the performance gap between CGENN and GLGENN is substantial. Moreover, GLGENN's parameter efficiency advantage increases with dimension $n$, as the subspaces $C\ell^{\overline{k}}_{p,q,r}$ diverge further from $C\ell^k_{p,q,r}$.

## O(5, 0)-Regression Task

In our first experiment, we consider an $\mathrm{O}(5,0)$-invariant regression task proposed in (Finzi et al., 2021). The task is to estimate the function $\sin(\|x_1\|) - \|x_2\|^3/2 + \frac{x_1^T x_2}{\|x_1\|\|x_2\|}$, where $x_1, x_2 \in \mathbb{R}^{5,0}$ are vectors sampled from a standard Gaussian distribution. The loss function used is Mean squared error (MSE).

We evaluate the performance using four different training dataset sizes and compare against the ordinary MLP, MLP with augmentations, the $\mathrm{O}(5,0)$- and $\mathrm{SO}(5,0)$-equivariant MLP architectures proposed in (Finzi et al., 2021), and with CGENN (Ruhe et al., 2023). CGENN, MLP, MLP with augmentations, $\mathrm{O}(5,0)$- and $\mathrm{SO}(5,0)$-equivariant MLP architectures have approximately the same number of parameters, we use the setup from the similar experiment provided in Ruhe et al., 2023 (they state that they compare their model CGENN with other models with the same number of trainable parameters).

The number of parameters in Table 5 ($\mathrm{O}(5,0)$-Regression Experiment) is as follows. All the models besides GLGENN and CGENN have $\approx 150.3$K parameters in total. In CGENN and GLGENN, the architecture contains sequentially applied geometric algebra-based layers (applied to the subspaces of all grades) and ordinary MLP layers (applied only to the subspace of 0-grade (scalars)). The most significant layers and at the same time the most consuming for training are geometric algebra-based ones, and CGENN possess $\approx 1.8$K of parameters associated with such layers, while GLGENN has $\approx 0.6$K of such parameters. For the MLP part, which is very fast and easy for training, both CGENN and GLGENN both have $\approx 148.5$K parameters.

*Table 5.* MSE (↓) on the $O(5,0)$-Regression Experiment.

| MODEL | # OF TRAINING SAMPLES | | | |
|---|---|---|---|---|
| | $3 \cdot 10^1$ | $3 \cdot 10^2$ | $3 \cdot 10^3$ | $3 \cdot 10^4$ |
| **GLGENN** | 0.1055 | 0.0020 | 0.0031 | 0.0011 |
| MLP | 28.1011 | 0.2482 | 0.0623 | 0.0622 |
| MLP+AUG | 0.4758 | 0.0936 | 0.0889 | 0.0672 |
| EMLP-O(5) | 0.152 | 0.0344 | 0.0310 | 0.0273 |
| EMLP-SO(5) | 0.1102 | 0.0384 | 0.032 | 0.0279 |
| CGENN | 0.0791 | 0.0089 | 0.0012 | 0.0003 |

*Table 6.* MSE (↓) on the $O(5,0)$-Regression Experiment in the case of large training set size.

| MODEL | # OF TRAINING SAMPLES | |
|---|---|---|
| | $6 \cdot 10^4$ | $1 \cdot 10^5$ |
| **GLGENN** | 0.0001 | 0.0001 |
| CGENN | 0.0002 | 0.0002 |

The results, presented in Tables 5, 6 and Figure 2 (left), demonstrate that GLGENN achieves performance on a par with CGENN, while significantly outperforming the other models. Moreover, GLGENN attains these results with fewer parameters and reduced training time compared to CGENN. In Table 5, MSE for CGENN and GLGENN are averaged over 5 runs. MSE for other models are averaged over 3 runs (from Finzi et al., 2021). Number of iterations is the same for all algorithms.

### $O(5,0)$-Convex Hull Volume Estimation

In this equivariant experiment, the task is to estimate the volume of a convex hull generated by $K$ points in $\mathbb{R}^{5,0}$. We consider three different values of $K$: 16 (as used in (Ruhe et al., 2023; Liu et al., 2024)) and two additional settings, $K = 256$ and $512$, which are more relevant for real-world applications.

In the case $K = 16$, we consider 4 different sizes of the training set: 256, 1024, 4096, and 16384 samples. We compare GLGENN with the state-of-the-art model CGENN (Ruhe et al., 2023). While both models share a similar architecture, GLGENN has 24.1K trainable parameters, whereas CGENN has 58.8K. The number of training steps is the same for both models. The results are presented in Table 3 and Figure 2 (middle). GLGENN either outperforms or matches CGENN across all training set sizes. The most noticeable difference occurs with smaller training sets. We attribute this to GLGENN's lower parameter count, which reduces its tendency to overfit—an issue commonly observed in small datasets.

To study the behavior of CGENN and GLGENN on different iterations of training, we provide Figure 4. This figure shows the training and test loss at different iterations. Notably, CGENN tends to achieve lower training loss compared to GLGENN. However, at a certain point, while CGENN's training loss continues to decrease, its test loss plateaus and remains almost constant. In contrast, GLGENN's training loss decreases at a slower rate, but its test loss is consistently lower than that of CGENN, indicating better generalization.

For $K = 256$ and $K = 512$, the results are also presented in Table 3. To ensure a fair comparison, we first select CGENN architectures that perform best for each setup. We then construct corresponding GLGENN models by replacing CGENN layers with their GLGENN counterparts, resulting in significantly fewer trainable parameters. Note that GLGENN consistently outperform CGENN in these real-world settings.

There is a contrast in GLGENN and CGENN generalization behavior in cases of large $K$. We illustrate it in Figure 5, which shows the training and test losses for $K = 256$ in the case of 1024 (left) and 16384 (right) training set sizes. GLGENN demonstrate stable convergence without signs of overfitting: the training and test losses decrease in a similar way throughout the optimization trajectory. CGENN show a clear overfitting pattern: while the training loss quickly drops to near-zero, the test loss plateaus early and remains significantly higher, especially in the small-data regime.

Following the recommendation of one of the anonymous reviewers, in Table 7, we report the average wall-clock time required to process one training batch for GLGENN and CGENN across different training set sizes and number of points $K$.

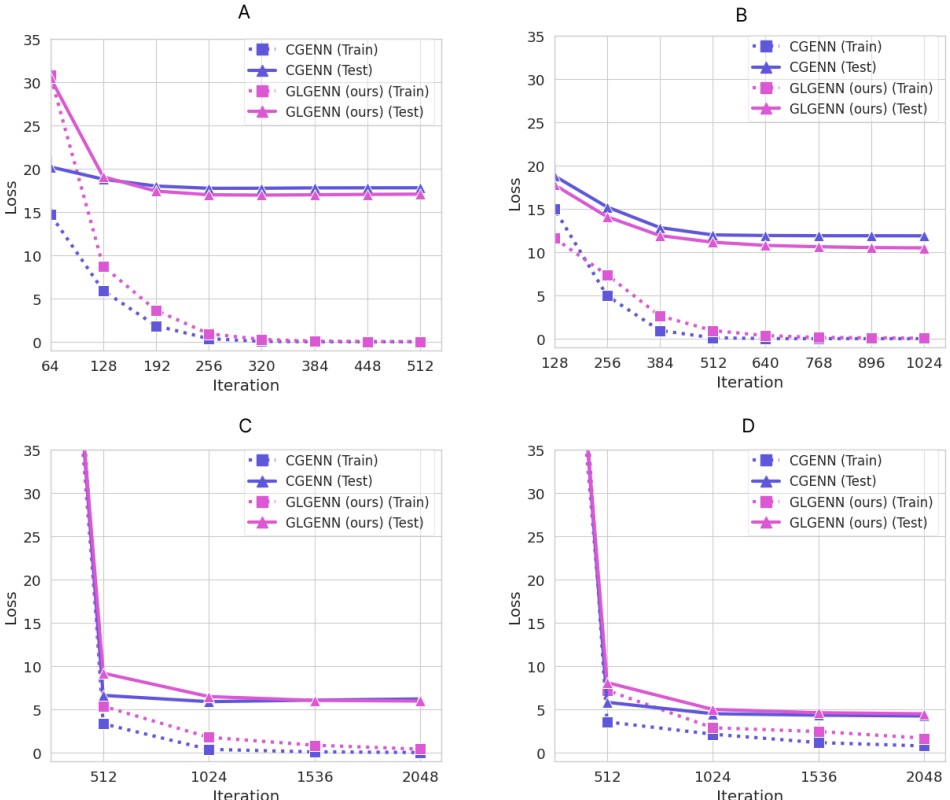

*Figure 4.* $\mathrm{O}(5,0)$-Convex Hull, $K = 16$. The plots illustrate the training and test loss curves for CGENN and GLGENN across different training iterations. Subfigures (A)–(D) correspond to different training set sizes: 256 (A), 1024 (B), 4096 (C), and 16384 (D), respectively.

The results demonstrate that GLGENN consistently achieve faster training times compared to CGENN. The performance gap remains notable across training dataset scales. The majority of memory usage is attributed to the Python environment and PyTorch's internal components rather than to the models themselves; as a result, the memory footprint is comparable between GLGENN and CGENN.

*Table 7.* Average time (in seconds) for processing one training batch in $\mathrm{O}(5,0)$-Convex Hull Experiment. In case of the number of points $K = 16$ and 256, for datasets with $2^8$, $2^{10}$, and $2^{12}$ training samples, the batch size is 128; for $2^{14}$ samples, the batch size is 256. In case of $K = 512$, for datasets with $2^{10}$ and $2^{12}$ samples, batch sizes are 256 and 512 respectively.

| **K** | | **16** | | | **256** | | **512** | |
|---|---|---|---|---|---|---|---|---|
| **MODEL** | **# TRAIN SAMPLES** | | | | **# TRAIN SAMPLES** | | **# TRAIN SAMPLES** | |
| | $2^8$ | $2^{10}$ | $2^{12}$ | $2^{14}$ | $2^{10}$ | $2^{14}$ | $2^{10}$ | $2^{14}$ |
| **GLGENN** | 0.22352 | 0.16708 | 0.1712 | 0.3408 | 3.9931 | 12.8758 | 13.3317 | 27.4383 |
| **CGENN** | 0.33692 | 0.2331 | 0.23418 | 0.43854 | 8.658 | 17.7424 | 17.6603 | 35.5605 |
| **GAP** | $-0.1134$ | $-0.06602$ | $-0.06298$ | $-0.09774$ | $-4.6649$ | $-4.8666$ | $-4.3286$ | $-8.1222$ |

## $\mathrm{O}(7,0)$-Convex Hull Volume Estimation

In this experiment, we extend the previous task with number of points $K = 16$ to a higher-dimensional setting and evaluate the performance of GLGENN and CGENN in estimating the volume of a convex hull formed by 16 points in $\mathbb{R}^{7,0}$. We consider three different training set sizes: 256, 512, and 1024 samples.

While both models share a similar architecture, GLGENN maintains 24.1K trainable parameters, as in the previous experiment, whereas CGENN's parameter count increases to 83.7K. The number of training steps remains the same for both models.

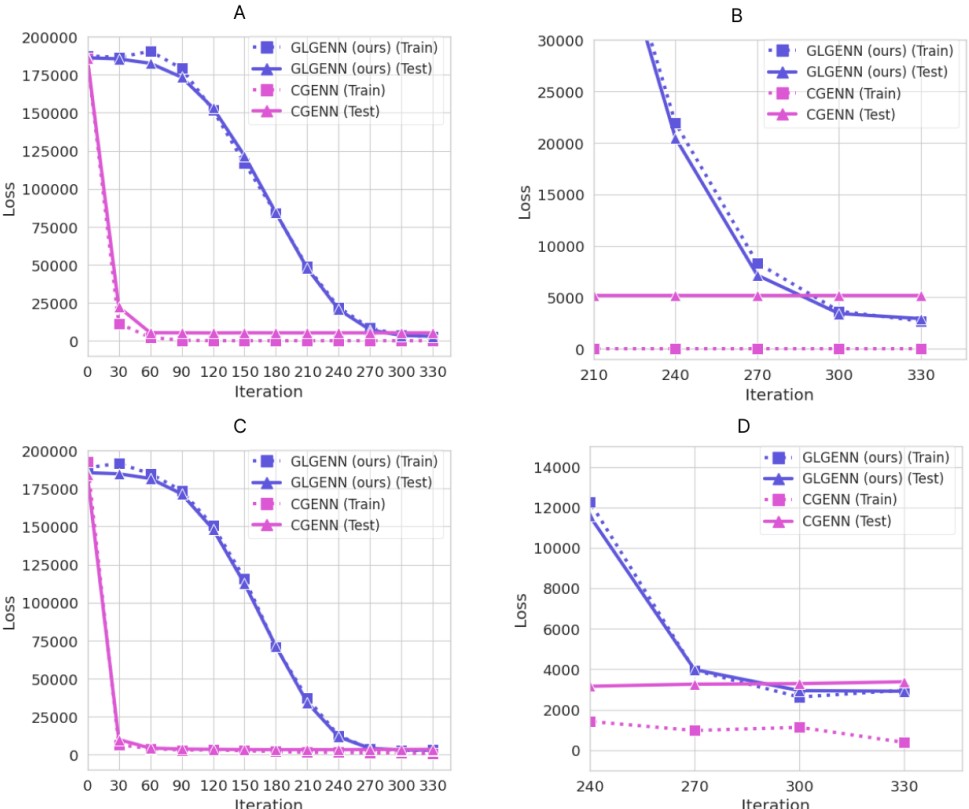

Figure 5. $O(5,0)$-Convex Hull, $K = 256$. The plots illustrate the training and test loss curves for CGENN and GLGENN across different training iterations. Subfigures (A)–(B) correspond to training with 1024 samples; subfigures (C)–(D) correspond to training with 16384 samples. The right-hand plots zoom in on the final iterations of training.

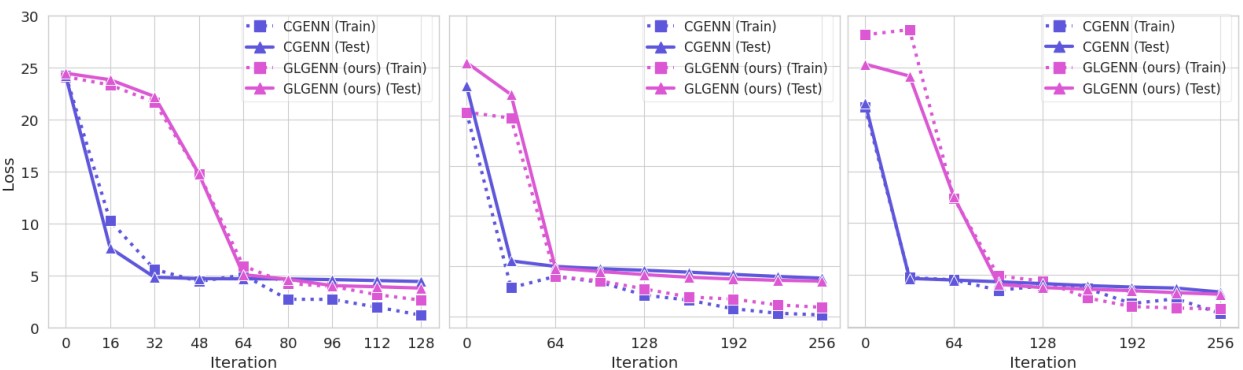

Figure 6. $O(7,0)$ Convex Hull. The plots illustrate the training and test loss curves for CGENN and GLGENN across different training iterations. Subfigures correspond to different training set sizes: 256 (left), 512 (middle), 1024 (right), respectively.

The results, presented in Table 8, show that GLGENN consistently outperforms or matches CGENN. The smaller is the training set size, the higher is the difference.

Figure 6 compares the training and test loss for CGENN and GLGENN across different iterations. The observed behavior is consistent with the results from the $O(5,0)$-Convex Hull Experiment: CGENN tends to achieve lower training loss, while GLGENN exhibits better generalization, particularly for smaller training sets.

*Table 8.* MSE ($\downarrow$) on the $O(7,0)$-Convex Hull Experiment on the test dataset.

| MODEL | # OF TRAINING SAMPLES | | | # OF PARA- |
|---|---|---|---|---|
| | $2^8$ | $2^9$ | $2^{10}$ | METERS |
| **GLGENN** | 4.0032 | 3.7378 | 3.5343 | 24.1K |
| CGENN | 4.4914 | 3.7756 | 3.5408 | 83.7K |

**CGENN Same Size as GLGENN**

In all our experiments, GLGENN architecture, by construction, has fewer trainable parameters than CGENN. This is because, for each task, we first construct the best-performing CGENN architecture and then obtain GLGENN by replacing CGENN layers with their corresponding GLGENN counterparts considered in Section 4, which have fewer parameters. Across all experiments, GLGENN either matches or outperforms CGENN, despite using fewer parameters.

In this subsection, we explore another setting: we compare GLGENN with CGENN architectures constrained to have the same number of parameters as the best-performing GLGENN. Our results show that, in almost all cases, CGENN performs worse under this constraint.

Table 9 presents results from the $O(5,0)$-Convex Hull Volume Estimation Experiment. In this setup, the CGENN model is downsized to have approximately 25K parameters (compared to 58.8K in its best-performing version), while GLGENN has 24.1K parameters. As shown, GLGENN outperforms the size-matched CGENN across all training set sizes.

*Table 9.* MSE ($\downarrow$) on the $O(5,0)$-Convex Hull Experiment ($K = 16$ Points) on the test dataset. Comparison of CGENN of different sizes with GLGENN.

| MODEL | # OF PARA- | # OF TRAINING SAMPLES | | | |
|---|---|---|---|---|---|
| | METERS | $2^8$ | $2^{10}$ | $2^{12}$ | $2^{14}$ |
| **GLGENN** | 24.1K | 16.94 | 10.40 | 6.2 | 4.46 |
| CGENN (SAME SIZE AS GLGENN) | 25K | 19.79 | 15.94 | 7.69 | 4.23 |
| CGENN (BEST PERFORMING) | 58.8K | 18.71 | 11.93 | 6.1 | 4.11 |

Table 10 reports results for the $O(5,0)$-Regression Task, where both GLGENN and CGENN are constrained to have approximately 149.1K total parameters, including about 0.6K parameters in geometric algebra-based layers. Again, GLGENN achieves superior or comparable performance.

*Table 10.* MSE ($\downarrow$) on the $O(5,0)$-Regression Task on the Test Dataset. Comparison of CGENN of different sizes with GLGENN. The second column shows the number of trainable parameters associated with geometric algebra-based layers.

| MODEL | # OF GA-PARA- | # OF TRAINING SAMPLES | | | |
|---|---|---|---|---|---|
| | METERS | $3 \cdot 10^1$ | $3 \cdot 10^2$ | $3 \cdot 10^3$ | $3 \cdot 10^4$ |
| **GLGENN** | 0.6K | 0.1055 | 0.0020 | 0.0031 | 0.0011 |
| CGENN (SAME SIZE AS GLGENN) | 0.6K | 0.2856 | 0.0076 | 0.0017 | 0.0005 |
| CGENN (BEST PERFORMING) | 1.8K | 0.0791 | 0.0089 | 0.0012 | 0.0003 |

**Application of Typical Activation Functions**

It is possible to combine geometric algebra-based layers with standard neural network layers, such as MLPs, while preserving equivariance — provided that the non-geometric algebra-based layers are applied only to scalars (i.e., elements of the subspace $C\ell^0$). The diversity of layers in this combination may lead to better results, although in equivariant tasks, non-geometric algebra-based layers on their own generally underperform compared to GLGENN layers or other geometric algebra-based layers. Applying non-geometric layers to $C\ell^0$ has inherent limitations compared to geometric algebra-based nonlinearities: they do not mix grades and instead isolate them, thereby hindering interactions across subspaces.

We provide an example in the case of the $O(5,0)$-Regression Task with 300 training samples. The results presented in Table 11 and Figure 3 show that the best performance is achieved by a combination of GLGENN (applied to all grades) and MLP (applied to scalars).

*Table 11.* MSE (↓) on combination of MLP with GLGENN and CGENN in O$(5, 0)$-Regression.

| MODEL | MSE ON TEST |
|---|---|
| MLP | 0.1706 |
| **GLGENN W/ MLP** | 0.0010 |
| CGENN W/ MLP | 0.0066 |
| **GLGENN W/O MLP** | 0.0265 |
| CGENN W/O MLP | 0.0723 |

## O$(5, 0)$-$N$-Body Experiment

We consider a system of $N = 5$ charged particles (bodies) in $\mathbb{R}^{5,0}$ with given masses, initial positions, and velocities. The task is to predict the final positions of the bodies after the system evolves under Newtonian gravity for 1000 Euler integration steps. For this purpose, we embed all the data in the geometric algebra $C\ell_{5,0}$.

We construct a graph neural network (GNN) based on the message-passing paradigm (Gilmer et al., 2017), where bodies are treated as nodes in a graph, and their pairwise interactions are modeled as edges. The message and update networks are equivariant GLGENN. We use $C\ell_{5,0}^{\overline{k}}$-linear, $C\ell_{5,0}^{\overline{k}}$-normalization, and $C\ell_{5,0}^{\overline{k}}$-geometric product GLGENN layers combined with MVSiLU layer from CGENN (Ruhe et al., 2023). We compare against CGENN, which itself outperforms several state-of-the-art methods, including steerable SE(3)-Transformers (Fuchs et al., 2020), Tensor Field Networks (Thomas et al., 2018), SEGNN (Brandstetter et al., 2022), Radial Field (Köhler et al., 2020), EGNN (Satorras et al., 2021), and NMP (Gilmer et al., 2017). To ensure a fair comparison, we use the best-performing CGENN architecture and then replace its layers with analogous GLGENN counterparts to obtain the GLGENN architecture, which automatically has two times fewer trainable parameters. The results are presented in Table 12 and Figure 2 (right).

*Table 12.* MSE (↓) on the O$(5, 0)$-$N$-Body Experiment.

| MODEL | # OF TRAINING SAMPLES | | | # OF PARA- |
|---|---|---|---|---|
| | $3 \cdot 10^1$ | $3 \cdot 10^2$ | $3 \cdot 10^3$ | METERS |
| **GLGENN** | 0.007 | 0.0011 | 0.0009 | 103K |
| CGENN | 0.0136 | 0.0015 | 0.0007 | 210K |

