# OpenReview forum: "GLGENN: A Novel Parameter-Light Equivariant Neural Networks Architecture Based on Clifford Geometric Algebras"
_ICML.cc/2025/Conference — ICML 2025 poster_

### Official Review · Reviewer_FkP7 · 2025-02-20

**Overall Recommendation:** 3

**Summary:**

The paper introduces Generalized Lipschitz Group Equivariant Neural Networks (GLGENN), a neural network architecture based on Clifford geometric algebras (GAs) that is equivariant to pseudo-orthogonal transformations of a vector space with a symmetric bilinear form. GLGENN uses a weight-sharing approach for layers calculating geometric products, resulting in a parameter-light architecture. The paper presents theoretical results on generalized Lipschitz groups, constructs the GLGENN architecture, and evaluates it empirically on toy tasks (regression of a function depending on two vectors and convex hull estimation of 16 points).

## update after rebuttal

Please see comments below.

**Claims And Evidence:**

The paper claims three main contributions:

1. Introduction of Generalized Lipschitz Groups

2. Design and Implementation of GLGENN

3. Superior Performance of GLGENN compared to other equivariant architectures

With regards to claim 1, the paper describes the theory of generalized Lipschitz groups in great detail and the concept seems sound to me. However, I cannot say whether this is truly novel: I believe it is possible (and even somewhat likely) that this is an already known result in mathematics that is just not widely known in the ML community, but I cannot say for sure (I don't know the mathematical literature well enough). I recommend reaching out to mathematicians that specialise in group/ring-theory to check this.

For claim 2, the design and implementation of GLGENN is well-described, and the authors also open-source their code. With regards to the claimed novelty of the work, it is unclear to me in what respects GLGENN goes beyond the already published CGENN (Ruhe et al., 2023). Even the parameter sharing technique seems to have been proposed previously. The differences with respect to published work, in particular, which improvements are made, need to be made more clear.

Finally, the claimed superior performance of GLGENN despite having significantly fewer trainable parameters seems not sufficiently well-supported by the results to me. While GLGENN outperforms MLPs, EMLP-O(5), and EMLP-SO(5) on the regression task, CGENN achieves a noticeably lower error (especially for larger training set sizes). For the convex hull estimation, performance of CGENN and GLGENN seem more similar, but the shown learning curve suggests that CGENN may start outperforming GLGENN for larger training set sizes. I think extended experiments on other (non-toy!) problems are required to support this claim.

**Essential References Not Discussed:**

The paper appears to cite all essential references. I'm not aware of any missing crucial publications. However, as mentioned above, I think it is possible/somewhat likely that the generalized Lipschitz groups are actually already known in the mathematical literature, and I recommend reaching out to mathematicians that specialise in group/ring-theory to check this.

**Experimental Designs Or Analyses:**

The experimental design of the toy problems investigated in this work seems sound to me. However, I would recommend adding additional experiments for real world datasets (see above), and possibly also extending the investigated toy problems (e.g., significantly more points for the convex hull estimation, larger training set sizes, etc.).

**Methods And Evaluation Criteria:**

The proposed methods (GLGENN layers) are well-motivated theoretically and make sense for the investigated problems. They are derived directly from the properties of geometric algebras and the generalized Lipschitz groups. The use of conjugation operations, adapted linear layers, geometric product layers, and normalization layers are all justified theoretically in terms of equivariance.

However, the toy problems investigated in this work are too simplistic to allow a meaningful assessment of the proposed architecture. I recommend that the authors also apply their architecture to real world problems where the use of equivariant models is common (e.g., regression of molecular properties such as energy or forces). This is necessary to demonstrate superiority over existing models, simple toy tasks are not sufficient.

**Other Comments Or Suggestions:**

While the visualization shown in Figure 1 is nice, the labels are too small and therefore hard to read.

**Other Strengths And Weaknesses:**

**Strengths**

+ The paper is mathematically rigorous, with detailed proofs and justifications provided in the appendices.

+ Despite the complex mathematical concepts, the paper is generally well-written and clearly explains the proposed methods and results.

+ The code is publicly available and the work is therefore easy to reproduce.

**Weaknesses**

- The experimental evaluation is limited to a toy regression and convex hull estimation problems. Evaluation on real world tasks (e.g., molecular property prediction, point cloud classification) and comparison to existing equivariant models on these tasks would strengthen the paper.

- It is not clear in what aspects the paper goes beyond existing work, in particular, CGENN (Ruhe et al., 2023).

**Questions For Authors:**

1. The case of Cl_(0,3,0) is arguably one of the most relevant for real world tasks in e.g., physics and chemistry. It seems that here, the scalars, vectors, bivectors (pseudovectors), and trivectors (pseudoscalars) of the Clifford algebra are isomorphic to irreducible representations of O(3) with degrees 0 and 1 of even and odd parity. Can the authors comment on this connection and do direct comparisons to equivariant models based on irreducible representations of O(3)?

2. The authors present Linear, Geometric Product, and Normalization layers for the Clifford algebra. I am aware that the Geometric Product and Normalization layers introduce nonlinearities, but I still wonder about the possibility of using more typical activation functions. It seems like any nonlinearity could be applied to the scalar feature components, and antisymmetric functions could be applied to the trivector/pseudoscalar feature components, without affecting the equivariance of the resulting model. Have the authors considered this possibility? How does the introduction of nonlinear activation functions affect the performance?

**Relation To Broader Scientific Literature:**

The paper clearly positions itself within the existing literature on equivariant neural networks. It primarily builds upon prior work on Clifford Group Equivariant Neural Networks (CGENN) (Ruhe et al., 2023). The paper also cites foundational work on equivariant networks (Cohen & Welling, 2016a; Weiler & Cesa, 2019; Thomas et al., 2018).

As I mentioned before, it is less clear to me in which aspects the work goes beyond existing work, and I think the authors should try to state this more clearly.

**Theoretical Claims:**

I did not notice any obvious mistakes in the proofs or theoretical claims while reading the paper, but I did not check carefully.

---

> ### Author Rebuttal · Authors · 2025-03-31
>
> We thank the reviewer for the valuable feedback. We are happy to answer the questions and address the concerns:
>
> **1. Novelty of theory**
>
> We claim that the proposed theory of generalized Lipschitz groups (GLG) in Clifford algebras (CA) $Cl_{p,q,r}$ is new (Sect. 3). These groups are introduced in arbitrary $Cl_{p,q,r}$ for the first time in this work. All theorems in the main part are original and cannot be found in the literature.
>
> **2. How GLGENN goes beyond CGENN**
>
> GLGENN goes beyond CGENN in the following: (1) Parameter-sharing approach for CA-based NN is presented for the first time (see Point 3). (2) New layers with parameter sharing are proposed, theoretical justification is provided (Sect.4). (3) General idea is proposed that if we need orthogonal groups equivariance, then we may search for broader groups equivariance, such as new GLG, and get reasonable results (Theor.3.4 and experiments). (4) New GLG, that are interesting by themselves, are introduced and studied (Sect.3). **We will add** these details.
>
> **3. Novelty of parameter-sharing technique**
>
> To the best of our knowledge, this work is the first to incorporate parameter-sharing techniques in CA-based NN that explicitly respect the inner structures of CA. By our approach, the weights are shared with the step size 4 in dimension of grades subspaces. This step size aligns with reversion and grade involution, the theoretical part of our work explains why this is important.
>
> **4. Limited experiments**
>
> We thank the reviewer for the ideas and would like to make clarifications. (1) While our experiments are toy problems, they are benchmarks in the field and used in highly regarded papers on equivariant NN (e.g., Finzi et al.,2021; Ruhe et al.,2023; Liu et al.,2024). This is why we chose them as a proof of concept. (2) The convex hull task is particularly challenging in high-dimensional settings, as evidenced by the sufficient loss obtained by previous SOTA models (Fig.3). (3) Adaptation of GLGENN (or other CA-based NN) to complex domains such as molecular/protein studies or images/point clouds is a very important but extensive task that probably deserves a distinct dedicated paper (e.g. see Pepe et al. (2024) on CGENN for protein structure prediction). (4) To strengthen our experimental section, **we will add** experiments, see Point 6 of our response to Reviewer xqBf.
>
> **5. Q1: Connection of CAs to irreps of O(3)? Comparison of GLGENN and models based on irreps of O(3)?**
>
> We do not see a direct connection between $Cl_{0,3,0}$ and the irreps of O(3). Irreps of O(3) follow the tensor approach, whereas the CA operates with multivectors. The dimensions of the subspaces of scalars, vectors, bivectors, and trivectors in CA are equal to 1,3,3,1 resp.; in contrast, for rank-2 tensors, the dimensions of the l-irreps of O(3), l=0,1,2 are 1,3,5 resp., so there is no isomorphism between them. E.g., bivectors are isomorphic to antisymmetric tensors of degree 2, which do not correspond to l=2-irrep of O(3) for rank-2 tensors.
>
> While equivariant models based on irreps of O(3) rely on tensor product reps, which can be complex and require additional basis reps, GLGENN bypass the need for them. This leads to 2 benefits: GLGENN directly transform data in a vector basis, avoiding the need for operating on alternative basis reps such as spherical harmonics; GLGENN involve geometrically meaningful product structure through the geometric product.
>
> **6. Q2: Typical activations to scalars?**
>
> Yes, it is possible to apply standard activations to scalars and be equivariant, we explored this in our work. Our results indicate that for simple tasks, the best approach is to combine GLGENN (applied to all grades subspaces) with standard networks (e.g. MLP) with typical activations applied to scalars (0-grade subspace). E.g. in O(5,0)-Regression: (1) MLP (3 linear layers with ReLU) alone performs poorly, (2) GLGENN alone performs reasonably well but converges slower than (3) GLGENN (to all grades)+MLP (to scalars). In case of 300 train samples, we get the results [visualized here](https://drive.google.com/file/d/1VqlvFTx-SqGJo3-OSES6MMVBkuJJi1K5/view?usp=sharing) and presented in the [table](https://drive.google.com/file/d/14nemp06cX7XtjftbKHplxcPCLIk-VHcU/view?usp=sharing)). But in more complex tasks, this combination becomes less critical. In all our convex hull experiments, the best results are achieved with the GLGENN nonlinearities, w/o additional activations. The key issue is that nonlinearities applied only to specific subspaces (e.g. scalars) do not allow interactions between different grades (e.g. vectors and bivectors), isolating them; while the nonlinearities in geometric product layers mix up all grades, creating strong interactions. **We will add** these details.
>
> **7. GLGENN on large train datasets**
>
> We are positive regarding GLGENN scalability to larger train datasets and have extended our experiments. Please see Point 6 of our response to Reviewer RrP5.

---

> > ### Comment · Reviewer_FkP7 · 2025-04-03
> >
> > I thank the authors for their replies and promised additions to the paper. I raise my score to 3 (I am assuming the authors hold their word and will do the promised changes in the final version of the manuscript).
> >
> > However, there is one aspect of my review I wish the authors would not so easily dismiss: I still believe there is a connection to O(3) irreps (I also consulted with a mathematician colleague of mine, who agreed with me).
> >
> > > We do not see a direct connection between and the irreps of O(3). Irreps of O(3) follow the tensor approach, whereas the CA operates with multivectors. The dimensions of the subspaces of scalars, vectors, bivectors, and trivectors in CA are equal to 1,3,3,1 resp.; in contrast, for rank-2 tensors, the dimensions of the l-irreps of O(3), l=0,1,2 are 1,3,5 resp., so there is no isomorphism between them. E.g., bivectors are isomorphic to antisymmetric tensors of degree 2, which do not correspond to l=2-irrep of O(3) for rank-2 tensors.
> >
> > Note that a rank-2 tensor (3x3 matrix) can be written as a direct sum of irreps with l=0 (with even parity), l=1 (with odd parity), and l=2 (with even parity). It is correct that these have dimensions 1, 3, and 5, which matches the 3x3=9 degrees of freedom of a rank-2 tensor. But this is not what I am talking about, it is not about equivalence of CA with rank-2 tensors. In fact, l=2 irreps have no equivalent in CA, but l=0 and l=1 irreps do:
> >
> > Scalars in CA are equivalent to l=0 irreps with even parity (one dimension, does not change sign under reflections of the coordinate system), vectors are equivalent to l=1 irreps with odd parity (three dimensions, behave rotationally like a vector and change direction under reflections of the coordinate system), bivectors are equivalent to l=1 irreps with even parity (three dimensions, behave rotationally like a vector and do *not* change direction under reflections of the coordinate system), and trivectors are equivalent to l=0 irreps with odd parity (one dimension, *does* change sign under reflections of the coordinate system). The dimensionality matches, the behaviour under rotations/reflections of the coordinate system matches, and even coupling operations seem to match (for example, coupling two l=1 irreps with odd parity to an l=1 irrep with even parity is essentially computed by performing a vector cross product, same as in CA).
> >
> > Given that there is a wealth of literature on equivariant models based on O(3) irreps, I think it is worth exploring this connection further and try to bridge the two communities. I would really appreciate if the authors added at least a small paragraph to their paper that discusses this.
> >
> > > While equivariant models based on irreps of O(3) rely on tensor product reps, which can be complex and require additional basis reps, GLGENN bypass the need for them. This leads to 2 benefits: GLGENN directly transform data in a vector basis, avoiding the need for operating on alternative basis reps such as spherical harmonics; GLGENN involve geometrically meaningful product structure through the geometric product.
> >
> > That statement is wrong. If one is limited to irreps with l=0 and 1 (same as in CA), then l=0 can be represented by (pseudo)scalars and l=1 by ordinary (pseudo)vectors. Nothing is more complex here. The coupling operations you need are not more complex than for CA either. In fact, they are the same (I use + for even and - for odd parity):
> > * 0+ and 0+ to 0+: multiplication
> > * 0+ and 0- to 0-: multiplication
> > * 0+ and 1+ to 1+: scalar multiplication
> > * 0+ and 1- to 1-: scalar multiplication
> > * 0- and 1+ to 1-: scalar multiplication
> > * 0- and 1- to 1+: scalar multiplication
> > * 1- and 1- to 0+: scalar (dot) product
> > * 1- and 1+ to 0-: scalar (dot) product
> > * 1- and 1- to 1+: cross product
> > * 1- and 1+ to 1-: cross product
> > * 1+ and 1+ to 1+: cross product
> > (I hope I didn't miss anything, but I'm sure the general idea comes across).

---

> > > ### Author Response · Authors · 2025-04-09
> > >
> > > We sincerely thank the reviewer for the detailed and insightful comments and for raising the score. We agree that exploring connections between CA and irreps of O(3) could be valuable for bridging communities. We will add a dedicated paragraph to the paper clarifying the correspondence between CA subspaces (k-vectors) and specific O(3) irreps. We are grateful for the reviewer’s rigorous engagement, which will undoubtedly strengthened the paper.

---

### Official Review · Reviewer_RrP5 · 2025-03-07

**Overall Recommendation:** 4

**Summary:**

This paper formally introduces neural networks that are equivariant with respect to generalised Lipschitz groups $\tilde{\Gamma}_{p,q,r}^{\bar{k}}$ for $k = 0, 1, 2, 3$ that are constructed from an arbitrary degenerate or non-degenerate Clifford geometric algebra (GA). The construction is based on two fundamental conjugation operations in the GA, called grade involution and reversion, that create so-called "subspaces of quaternion types" for $k = 0, 1, 2, 3$ which are preserved by the generalised Lipschitz groups under the twisted adjoint representation $\tilde{ad}$. The authors show, in particular, that for $k = 1$, all generalised Lipschitz group equivariant mappings are also equivariant to group that is restricted from a degenerate or non-degenerate orthogonal group to the radical subspace $\Lambda_r^1$. The resulting neural network, termed GLGENN, is compared against CGENN (Ruhe et al. 2023) and other neural networks on two series of tasks, an $O(5,0)$-regression task and a $O(5,0)$- and $O(7,0)$-Convex Hull Volume estimation task. The authors claim that GLGENN achieves a reduced tendency to overfit having significantly fewer parameters than CGENN.

**Claims And Evidence:**

Let me start by saying that this is an absolute monster of a paper in terms of its conceptual difficulty, and to be able to give this review I have taken 3 full days to read/work out as many of the details of the paper as I can properly, in order to try to understand everything that it is saying and to give a fair and informed review to the authors. It seems to me that this paper follows/builds upon a sequence of previous publications (Ruhe et al, 2023; Brehmer et al, 2023; Zhdanov et al, 2024; Liu et al., 2024; Pepe et al. 2024), which I believe makes the entry into this paper very difficult if you have not read these previous works (I have not). I am convinced that for 95%+ of people reading this paper they would give up as soon as they read the Theoretical Background section (Section 2), since it is incredibly slick and technical. Despite this, having read it a few times, I felt that it was comprehensive and with a bit of work from my side I could understand what it was saying. I appreciate that the paper is already long (32 pages including Appendix) but I think to aid the reader even more I would like the authors to provide a running example (e.g on the Minkowski space $\mathbb{R}^{1,3}$) of their constructions that appear in this section in the Appendix. From there we get into the meat of the paper in Section 3. Again this is very slick, which I understand based on the 8 page limit, but it is apparent to me that in order to even begin to understand what is going on you really need to read the Appendix first. I can't help feeling that this paper, given its technical depth, would have been better served by being written as a Journal paper instead, since in my view the logical flow would be more apparent in a journal paper, without having to make jumps between theorems (as I do in its present form). In spite of this, everything that I could understand in this section (about 80% of the section, let's say) was to me correct (and that is going line-by-line through it and the Appendix). So in my mind, on the basis of probability, I do not have doubts about the theoretical claims that were made by the authors. I would still strongly encourage them to think even more about how to present this work more cleanly - I appreciate that they have done this to a large extent already (e.g with take-away summaries after each subsection, a notation overview in the appendix etc) but there is a lot going on (many groups and algebras are introduced, some with tildas, some without, some with bars, some without) and I think at times some notation is used without it being defined anywhere. For example, I don't think, as far as I can tell, that $(Cl_{p,q,r}^{(0)\times} \cup Cl_{p,q,r}^{(1)\times})\Lambda_r^{\times}$ is defined anywhere in the text. I would also like the authors to be consistent with whether they use $U$ or $x$ to refer to an element in $Cl_{p,q,r}$: there is a switch from $U$ to $x$ that takes place between Sections 2 and 3 that could confuse people.

**Essential References Not Discussed:**

N/A

**Experimental Designs Or Analyses:**

I was quite disappointed by the experiments that the authors chose, and it led me to thinking whether the theory that had been provided before was "overkill". Having produced their exceptionally technical theoretical results, the authors chose to demonstrate their results on tasks that are solely based on Euclidean space, with the equivariance being to a standard orthogonal group on said Euclidean space. Hence I am left wondering how their network performs on tasks that are equivariant to groups where $q > 0$. I am also not entirely convinced by the conclusions of their experiments either. In both cases the authors claim that GLGENN performs on a par with the state-of-the-art CGENN and reduces overfitting with fewer parameters.  I am not convinced that GLGENN performs on a par with CGENN. Looking at Figure 2 with Table 2, and then Figure 3 and Table 3, I see that GLGENN starts performing worse than CGENN as the number of training samples increases, and, in particular, in a divergent manner in Table 3. I therefore ask what is the behaviour of the test MSE of these two models as the number of training samples increases?

**Methods And Evaluation Criteria:**

For me, the methodology section (Section 4) was relatively clear, where the authors described the architecture of GLGENN. I did think the commentary about the data objects in 4.1, however, was not entirely clear - where do these data objects ($x_1, \dots, x_l$) live? I think this could be stated directly in the text. Otherwise I was happy with what the authors wrote in this section.

**Other Comments Or Suggestions:**

Here are some comments/suggestions/typos:

1) I think you should define $(Cl_{p,q,r}^{(0)\times} \cup Cl_{p,q,r}^{(1)\times})\Lambda_r^{\times}$ somewhere before it is first used in Lemma 2.1.

2) As discussed, I think you should prime the reader as to why you introduce the $Q$ families of Lie groups in lines 185 etc, second column, as well as the $Z$ centralisers (as in tell us where you're going with this much earlier than the concluding sentence.)

List of (potential) typos:

1. Line 320, 1st column: equivariant is spelled incorrectly
2. Line 341, 1st column: you write $Cl$, do you not mean $Cl_{p,q,r}$?
3. Line 362, second column: I think this should be Theorem 3.9, not Lemma 3.9
4. Appendix F: in the title, equivariant is spelled incorrectly
5. Appendix F, Example F1: do you not mean $T$ instead of $g$ in both (137), (138)?

**Other Strengths And Weaknesses:**

Without repeating myself:

Strengths: This is clearly a high quality technical work in the field of equivariance that demands a significant amount of the reader to understand the details. I felt that I learned a lot and in many parts enjoyed "grokking" the details. I also liked the mini-summaries as the work progressed, which helped me to take away the key points when I got entirely lost in the technical details (this is something I will be taking and using in my own papers!) I thought the Appendix was pretty comprehensive, and the discussion on the methodology was good too. Despite my strong doubts about whether this should be a journal paper or not, I still think there will some people at ICML who will appreciate a technical work like this, so on balance (with my score) I  recommend it (only just) for publication. However I would like the authors to take on board my comments about improving the paper's accessibility - even if it is a work in a line of works there should still be sufficient runway given to make it into enough of a standalone paper.

Weaknesses: As stated above I was disappointed by the types of experiments that appeared, which to me did not show the full benefit of the theoretical construction. I also think more can be done to help facilitate the entry into this paper - whilst I appreciate the contribution is technical, the onus is still on the authors to help make things as clear as they can. I have suggested providing a running example of all key constructions (not just a summary of their meaning) in the Appendix. I would also like the message of what the authors are actually trying to show to be clearer too: for example, in the main text, lines 165-170 (2nd column) the authors say that they are looking to construct equivariant NNs with respect to $\tilde{\Gamma}_{p,q,r}^{\bar{k}}$ for $k = 1$, but in Appendix B, lines 641-645, they say the NNs are equivariant to general $k$ - which one is it? I appreciate this is tough given the technical nature of the paper but again I really want the story to shine through (this doesn't always stick as you read it despite the mini summaries) - perhaps they could look to "give up" more of the story earlier on so that we know where you're going? An example of this appeared in Section 3.1 where the authors just suddenly introduce large families of groups (lines 184-207, second column) without telling us why/where it's leading - as readers we're already under the strain as it is!

**Questions For Authors:**

1) Line 134, 1st column: do you really mean the word "superposition"? To me, superposition means add/linear combination, but here I think you mean the combination/composition of the two operations, one after the other.

2) I think (10) could be formatted better (similarly (16), (17)). Also for (10) I don't think that the $:=$ that comes after $\tilde{ad}_T(...)$ is correct, isn't it a deduction from (7)? Upon first reading I had wondered if you had meant $\breve{ad}_T(...)$ instead.

3) In line 235, second column, in the discussion, there is a switch to $V$ - just to check, is this is the same as $\Lambda_r^1$? If so, I think you should be telling us otherwise it looks like a typo.

4) (a copy from above) in the main text, lines 165-170 (2nd column) the authors say that they are looking to construct equivariant NNs with respect to $\tilde{\Gamma}_{p,q,r}^{\bar{k}}$ for $k = 1$, but in Appendix B, lines 641-645, they say the NNs are equivariant to general $k$ - which one is it? If it is only $k = 1$, then I think in lines 165-170 they should tell us as they introduce everything for general $k$ and to the reader at this point it looks like a mistake (hence could be confusing as it goes along).

5) In (36) do you not need brackets in there around $x$ on the RHS and around $\langle x \rangle_{\bar{m}}$ on the LHS?

6) (a copy from above) in both of your experiments, what is the behaviour of the test MSE of these two models as the number of training samples increases? I am not entirely convinced of the "on par" performance of the two models GLGENN and CGENN.

7) To improve readability, is there any way to reduce the general complexity that is presented if you are only targeting equivariance for $k = 1$?

8) Finally I felt that Figure 1 did nothing for me - I still don't understand what I am meant to be taking from it, even after reading the text in full multiple times. Could the authors try to explain what I should be taking from this figure (and then perhaps improve it/move it further down somewhere where it might fit better).

## update after rebuttal: please see my comments in my response to the authors' rebuttal.

**Relation To Broader Scientific Literature:**

The relation to the broader scientific literature was comprehensive, with a very nice introduction that set the work in the appropriate context.

**Theoretical Claims:**

I have already answered this above. I have checked 80% of this line by line, and where it is simply beyond my technical understanding, I have had to leave it unchecked. But I am happy with everything that I have checked.

---

> ### Author Rebuttal · Authors · 2025-03-31
>
> We thank the reviewer for the valuable feedback. We are happy to answer the questions and address the concerns as follows:
>
> **1. Q1: Superposition in Clifford conjugation?**
>
> Thank you. We will change the word to ‘composition’.
>
> **2. Q2: Formatting in (10),(16),(17)?**
>
> Thank you. We will format definitions (10),(16),(17) so that they fit on a single line each, remove := in them, and add := in (5)-(7).
>
> **3. Q3: Restriction of $\tilde ad$ to V?**
>
> No, this is not a typo, we intentionally refer to V, which is the real $\mathbb R^{p,q,r}$ or complex $\mathbb C^{p+q,0,r}$ vector space (line 98). V is identified with the subspace $Cl^1_{p,q,r}$ of vectors. In line 235, we mean that $\tilde{ad}^1$ is defined as the twisted adjoint representation $\tilde ad$ (7) restricted to the entire subspace $V=Cl^1_{p,q,r}$, not just to the radical subspace $\Lambda^1_r=Cl^1_{0,0,r}$. Namely, while $\tilde{ad_T}$ acts on $Cl_{p,q,r}$, the restricted representation $\tilde{ad}^1_T$ acts only on all vectors $Cl^1_{p,q,r}$ (line 237). We will add this explanation to avoid any confusion.
>
> **4. Q4: Equivariance to which of new groups is used in GLGENN?**
>
> Thank you. GLGENN is equivariant w.r.t. $\tilde\Gamma_{p,q,r}^{\bar1}$. We will remove k=0,2,3 in lines 641-645 and clarify in the beginning of Sect.3, that for GLGENN, we are interested in $\tilde\Gamma_{p,q,r}^{\bar1}$, however other groups $\tilde\Gamma_{p,q,r}^{\bar k}$, k=0,2,3, are still necessary to prove the main statements about $\tilde\Gamma_{p,q,r}^{\bar1}$ (see Q7).
>
> **5. Q5: Brackets in $\tilde{ad}_T(x)$?**
>
> In the literature, both $\tilde{ad}_T(x)$ or $\tilde{ad}_T x$ are commonly used. We will put brackets in the whole paper for the sake of accuracy.
>
> **6. Q6: Behavior of test MSE when training set size increases?**
>
> The test MSE as the training set size increases is as [follows](https://drive.google.com/file/d/1llsG1wPRNyQ4IIicZNt4Wpyxn4IJn7oZ/view?usp=sharing). We will add these results. In regression, GLGENN consistently outperforms CGENN across all training set sizes. In the convex hull task, the performance gap becomes stable, and GLGENN still is competitive. Again note that in this task, GLGENN and CGENN have 24.1K and 58.8K parameters respectively. We are positive regarding GLGENN scaling. But as we state, our goal is to reduce the risk of overfitting in case of small training data — a common scenario in natural sciences, where datasets are often manually derived from experiments.
>
> **7. Q7: Role of $\tilde\Gamma_{p,q,r}^{\bar k}$, k=0,2,3?**
>
> We share the reviewer's concern about reducing general complexity to improve readability. While equivariance w.r.t. $\tilde\Gamma_{p,q,r}^{\bar1}$ is directly required for GLGENN, the other 3 groups play a crucial role in proving a key fact about this group: its elements preserve all the 4 subspaces $Cl_{p,q,r}^{\bar k}$, k=0,1,2,3 under $\tilde ad$ (see (28)-(29)). This implies that projections onto these 4 subspaces are $\tilde\Gamma_{p,q,r}^{\bar1}$-equivariant (Theorem 3.6). We appreciate and will implement the idea of making it clearer at the beginning of Sect.3 that $\tilde\Gamma_{p,q,r}^{\bar1}$ is the primary group of interest, while the others are auxiliary tools.
>
> **8. Q8: Figure of GLGENN?**
>
> Figure 1 does not introduce new information but serves as a visual representation of concepts already described in the text. We intended the following key points to be intuitively grasped by readers when looking at it: (1) GLGENN is equivariant w.r.t. orthogonal groups (we can apply an orthogonal transformation to the input or output of GLGENN and are guaranteed to get the same answer); (2) GLGENN acts in a unified manner across the subspaces $Cl_{p,q,r}^{\bar k}$, k=0,1,2,3 defined by the grade involution $\hat{}$ and reversion $\tilde{}$. As a result, scalars, vectors, bivectors, etc. are formed into 4 distinct groups. Point (1) is visualized by the orange and purple arrows and the 4 main large rectangles forming a diamond shape. Point (2) is represented by the 4 smaller rectangles (yellow, green, purple, and orange) and +/- signs indicating the actions of grade involution and reversion on these groups.
>
> **9. Notation**
>
> The notation $(Cl_{p,q,r}^{(0)\times}\cup Cl_{p,q,r}^{(1)\times})\Lambda_r^{\times}=${$ab | a\in Cl_{p,q,r}^{(0)\times}\cup Cl_{p,q,r}^{(1)\times}, b\in\Lambda_r^{\times}$} represents the product of 2 groups: the group of even or odd invertible elements (formula (3)) and the group of all invertible elements of the Grassmann subalgebra (line 104). We will add these clarifications and ensure that all other notations are properly defined.
>
> **10. Typos**
>
> We agree with all the corrected typos, thank you.
>
> **11. Improvement of presentation**
>
> We are very grateful to the reviewer for the comments on how to improve the presentation. We like and will surely implement the ideas of providing a running example of the main concepts and revealing more of the key ideas earlier in the paper.

---

> > ### Comment · Reviewer_RrP5 · 2025-04-02
> >
> > I thank the authors for their detailed response. I am satisfied with their comments and am glad to see the results of the additional experiments that were run - although I maintain my score I am of the view that this paper should be recommended for publication.

---

> > > ### Author Response · Authors · 2025-04-09
> > >
> > > Thank you for your kind feedback and for taking the time to review our work. Your constructive feedback is very valuable for strengthening the paper, and we’re grateful for your recommendation for publication.

---

### Official Review · Reviewer_xqBf · 2025-03-13

**Overall Recommendation:** 3

**Summary:**

This paper introduces a new version of Clifford Group Equivariant Neural Networks (CGENN) originally introduced by Ruhe et al., 2023 called GLGENN.  The authors develop a theory for generalized Lipschitz groups which generalize and contain Clifford groups.  Generalized Lipschitz groups preserve a subspace decomposition corresponding to involution and reversion operations on the Clifford algebra.  If I understand correctly, enforce equivariance to this larger group using a weight sharing scheme with respect to this coarser subspace decomposition as opposed to the finer grading results in fewer trainable parameters relative to CGENN.  Since the group is larger, there is still equivariance to the clifford group and thus to the orthogonal group.  The experiments show that there is no loss in performance despite the fewer parameters and the overconstraint.

**Claims And Evidence:**

- The paper notes that CGENNs are overparameterized and tend to overfit and have slow training times motivating the need for a fewer parameter version.  I didn’t specifically see evidence of overfitting in the experiments.  The slow training time for CGENN is certainly a concern, but there is not evidence in the main the GLGENNs remedy this.
- GLGENNs are shown to have comparable performance to CGENNs even with fewer parameters.
- Whether the larger group constraint poses a problem for expressiveness is not conclusively shown.  In the 2 synthetic tasks considered it did not.
- The equivariance of GLGENNs is firmly established by an extensive theory section.

**Essential References Not Discussed:**

No

**Experimental Designs Or Analyses:**

- As noted above, the experiments do demonstrate a smaller GLGENN performs comparably to CGENN, but it would be good to also show that a CGENN the same size as the GLGENN underperforms the GLGENN.  If not, it undermines them main claims.
- The experiments are very small scale and very synthetic.  The practical usefulness of the method is not demonstrated.  My guess is that it will scale, however, given how similar it is to CGENN which has ben demonstrated in practical applications.
- While GLGENN has fewer parameters, memory footprint and wall clock compute should also be compared.

**Methods And Evaluation Criteria:**

- The method is a reasonable evolution of CGENN as described in the summary above.  The authors provide analogs of the layers provided by Ruhe et al, 2023.  Except for the Conjugation operation, the novelty appears to be limited to generalization to the new group and grading.
- The evaluations are similar to those in prior work, but are small scale and synthetic.  As noted above, the question of overfitting and computation speed do not appear to be directly addressed.

**Other Comments Or Suggestions:**

320L: Equivariant misspelled

**Other Strengths And Weaknesses:**

n/a

**Questions For Authors:**

- Can the model accommodate higher order features such as tensors?
- Since the parameters of the conjugation operations are discrete, how are they optimized? Are they used in experiments? It would be good to comment in the paper on this.
- The linear layers cannot mix information among \bar{k}, correct?
- Are geometric products and normalization the main source of non-linearity and mixing \bar{k}?

**Relation To Broader Scientific Literature:**

There is no related work section, but related work is adequately addressed in the introduction and throughout the paper in my opinion.

**Theoretical Claims:**

- A definitive strength of the paper. The mathematics of clifford geometric algebras, and generalized Lipschitz groups is very clearly laid out both in the background and new contributions.
- The authors prove several results which firmly underlie their method: (1) generalized lipschitz groups contain the clifford groups and thus equivariance wrt gen. lipschitz groups implies clifford group equivariance (2) the subspace decomposition is preserved by gen. lipschitz groups, and (3) gen lipschitz equivariance implies orthogonal group equivariance, (4) several different operations are equivariant wrt gen. lipschitz groups, providing the basis for the layers in CGENN.
- The theory section could use a bit more scaffolding/prefacing to help connect the results to their eventual use in constructing CGENNs.  I had to read forward and backward a bit to understand.

---

> ### Author Rebuttal · Authors · 2025-03-31
>
> We thank the reviewer for the valuable feedback. We are happy to answer the questions and address the concerns as follows:
>
> **1. Q1: Higher order features such as tensors?**
>
> Yes, the model can accommodate higher-order features, such as tensors. For example, it can be applied to the N-body problem, where the goal is to predict the positions of N charged particles in an n-dimensional space after a certain number of timesteps, given their initial positions and velocities. In this task, tensors represent the N initial positions and velocities (each with n coordinates). The model can process such tensors when graph neural networks are used to define the structure of connections between objects.
>
> **2. Q2: Conjugation operations layers?**
>
> Thank you, this is a good question. We suggest the following method to make the optimization of conjugation operations possible, though other potentially more effective approaches may be suggested. We first apply a linear transformation of the projections of the input multivector $x_{in}$ as in (42) but with any parameters $\phi_{c_{in}k}\in\mathbb{R}$. Then we can round them to -1 or 1 either by directly applying $\phi_{c_{in}k}\mapsto sgn(\phi_{c_{in}k})$ or by firstly applying the sigmoid function, then scaling the value to [-1,1], and applying sign function to the result. In the experiments presented, these layers are not used in order to maintain the architecture choices of GLGENN close to the corresponding of CGENN as much as possible to ensure fair competition. We mention this in lines 1558-1560 of Appendix F and will add a comment on it in the main part of the paper.
>
> **3. Q3: Linear layers cannot mix information among $Cl^{\overline{k}}_{p,q,r}$?**
>
> Yes, you are correct. The linear layers do not mix information among the subspaces $Cl^{\overline{k}}_{p,q,r}$, $k=0,1,2,3$. Instead, they perform linear transformations independently within each of these four subspaces. A visual representation of how a linear layer operates can be found at the following [link](https://drive.google.com/file/d/1ev594pDgVjKuMfXY5rCB4eZPK4W2jEju/view?usp=sharing).
>
> **4. Q4: Geometric products and normalization are the main source of non-linearity and mixing $Cl^{\overline{k}}_{p,q,r}$?**
>
> Yes, you are correct. GLGENN introduce non-linearity through the geometric product and normalization layers, where the subspaces $Cl^{\overline{k}}_{p,q,r}$, $k=0,1,2,3$, are mixed.
>
> **5. Overfitting of CGENN.**
>
> Our experimental results (see, for example, Figure 3) align with the results in CGENN paper Ruhe et al., 2023 (see Figure 3 (right) and discussion of the convex hull experiment on page 9 of their work), where they also note the tendency of CGENN to overfit in scenarios with small training datasets.
>
> **6. Other experimental designs.**
>
> We thank the reviewer for the idea on how to improve the work. We are positive regarding scaling of GLGENN as well. To strengthen our experimental section, **we will add** the experiments done in prior works: (1) real-world task in $Cl_{1,3}$ of categorizing high-energy jets produced in particle collisions by their trajectories using the data from CERN’s ATLAS detector (this task is considered in CGENN paper by Ruhe et al., 2023), (2) classic benchmark N-body experiment (the task is considered in Ruhe et al., 2023 and GATr paper by Brehmer et al., 2023), and (3) harder settings for the convex hull and regression experiments, in particular, comparing CGENN of the same size as GLGENN. We will also include a comparison of memory footprint and wall clock time between GLGENN and CGENN. We would like to note that while the current experiments are toy problems, they are considered benchmark tasks in the field and have been used in highly regarded papers on equivariant neural networks (e.g., Finzi et al., 2021; Ruhe et al., 2023; Liu et al., 2024). This is why we chose them as a proof of concept, and GLGENN successfully validated our hypothesis that structured weight sharing improves performance.

---

> > ### Comment · Reviewer_xqBf · 2025-04-05
> >
> > I appreciate the authors response to my questions.  Regarding Q1, I believe the positions and velocities would just be vectors i.e. 1-tensors.  My question was about things like matrices (i.e. 2-tensors) where the group acts by conjugation.  If the authors do add additional experiments and the model performs well, I believe it would strengthen the paper.  I hold, however, that the paper should be accepted even without these larger scale experiments.

---

> > > ### Author Response · Authors · 2025-04-09
> > >
> > > We are grateful for your high evaluation of the paper. Thank you for your thoughtful feedback, time, and recommendation for publication.

---

### Official Review · Reviewer_qfnz · 2025-03-14

**Overall Recommendation:** 2

**Summary:**

This paper introduces a novel group equivariant architecture based on Clifford geometric algebra that are equivariant to pseudo orthogonal transformation and are more parameter efficient and avoids overfitting compared to prior works in equivariant networks based on Clifford algebra. The design is based on first designing a generalized Lipschitz group and then designing an equivariant network for this generalized Lipschitz group that leads to more parameter efficient networks. Experimental results on two synthetic datasets show competitive performance with respect to existing Clifford algebra equivariant networks.

**Claims And Evidence:**

While the theoretical contribution seems straightforward, the practical motivation is not very clear to me: how is equivariance to the generalized group leading to more parameter efficiency? Is it straightforward, please provide better explanation. Also, in the experiments, shouldn’t using a more general Lipschitz group lead to better performance? Why is the performance slightly worse than CGENN, i.e., is there a reason why the expressivity of the proposed method is limited compared to prior work?

**Essential References Not Discussed:**

Looks good to me

**Experimental Designs Or Analyses:**

Soundness/validity of experiments look good.

**Methods And Evaluation Criteria:**

The amount of experiments seem very limited compared to both the motivation of the work as well as prior works such as Geometric Algebra Transformers (GATr). Is it possible to provide more experiments done in prior works?

**Other Comments Or Suggestions:**

None

**Other Strengths And Weaknesses:**

Strengths:
1. This work proposes a novel architecture based on Clifford geometric algebra for a new class of Lie groups
2. The proposed equivariant model construction technique is light weight in terms of number of parameters compared to prior Clifford algebra networks
3. Experimental results show that the proposed method is competitive with existing methods as well as light-weight in terms of parameters

Weaknesses:
1. While the theoretical contribution seems straightforward, the practical motivation is not very clear to me: how is equivariance to the generalized group leading to more parameter efficiency? Is it straightforward, please provide better explanation. Also, in the experiments, shouldn’t using a more general Lipschitz group lead to better performance? Why is the performance slightly worse than CGENN, i.e., is there a reason why the expressivity of the proposed method is limited compared to prior work?
2. The amount of experiments seem very limited compared to both the motivation of the work as well as prior works such as Geometric Algebra Transformers (GATr). Is it possible to provide more experiments done in prior works?
3. What are the number of parameters in Table 2?

**Questions For Authors:**

Please check the weaknesses

**Relation To Broader Scientific Literature:**

This paper makes progress in the area of Clifford Algebra equivariant networks, which is an established direction in the area of efficient machine learning.

**Theoretical Claims:**

This looks good to me

---

> ### Author Rebuttal · Authors · 2025-03-31
>
> We thank the reviewer for the valuable feedback. We are happy to answer the questions and address the concerns as follows:
>
> **1. Q1: Relation between generalized Lipschitz groups equivariance and parameter efficiency?**
>
> Application of the generalized Lipschitz groups (introduced in this work) instead of ordinary Lipschitz groups allows to achieve parameter efficiency, we explain it below. The generalized Lipschitz groups are important because they preserve the four fundamental subspaces of Clifford algebras under the significant operation of the twisted adjoint representation. We prove that equivariance of a mapping w.r.t. these groups, as well as the ordinary Lipschitz groups, implies its orthogonal groups equivariance. The key distinction is that the generalized Lipschitz groups contain ordinary Lipschitz groups as subgroups. As a result, the set of operations equivariant w.r.t. the generalized Lipschitz groups is a subset of the set of operations equivariant w.r.t. ordinary Lipschitz groups. This reduction in the number of ‘degrees of freedom’ encourages us to parametrize operations in layers in a more ‘economic’ way (there is a smaller number of parameters that we can place). Specifically, in all GLGENN layers, we employ such equivariant operations as projections of inputs-multivectors onto the 4 fundamental subspaces of Clifford algebras mentioned above, whereas CGENN relies on projections onto the subspaces of fixed grades. GLGENN and CGENN layers parameterize linear combinations, products, and normalizations of the corresponding projections. The number of the fixed-grades subspaces is equal to $n+1$, where $n$ is the dimension of the task’s vector space. As $n$ increases, the number of CGENN parameters grows significantly, while for GLGENN, it remains constant. To summarize, the key reason why GLGENN layers are parameter-efficient lies in how we parametrize the operations in them: using the projections onto the 4 fundamental subspaces, instead of $n+1$ subspaces as in CGENN.
>
> The growth in the number of parameters for similar layers in GLGENN vs. CGENN as $n$ increases can be estimated by the formulas in our paper. GLGENN geometric product layer contains $4k^2+4^3k$ parameters for $k$ input channels (Sect. 4.4), while CGENN geometric product layer has $(n+1)k^2+(n+1)^3k$ parameters. Similar formulas for other GLGENN layers are in Sect. 4.3 and 4.5. The formulas for CGENN can be obtained by replacing 4 by $n+1$.
>
> We thank the reviewer for this good question for presentation improvement, **we will add** these more detailed explanations.
>
>
> **2. Q1 (part 2): Better performance of GLGENN in experiments?**
>
> One of the goals of our work is to design a model that performs better than CGENN in case of small training datasets, which is a common scenario in the natural sciences, where datasets are often manually derived from experimental results. GLGENN is designed to mitigate overfitting in such cases, while its expressivity does not necessarily provide an advantage when data is abundant. Specifically, GLGENN significantly outperforms CGENN in both convex hull experiments, particularly when the training set is small (e.g., $2^{8}$ or $2^{10}$ samples in $n=5$ case, Fig. 3). When the training set size is large for the task ($\geq 2^{12}$ samples), GLGENN and CGENN are on a par (with the difference <0.4 MSE, which is, in our opinion, almost insignificant in comparison to the difference in parameters quantity – 24.1K for GLGENN vs. 58.8K for CGENN), the similar behaviour remains in the training set size $>2^{14}$ samples. In the regression task, GLGENN again achieves lower MSE than CGENN when trained on a small dataset (300 samples, 0.002 vs. 0.0089). With an extremely small training dataset (30 samples), GLGENN is on a par with all other models, except MLP and MLP+Aug, likely due to insufficient data for meaningful generalization. When the dataset is large ($3\cdot10^3$ or $3\cdot10^4$ samples), CGENN slightly outperforms GLGENN, but the difference diminishes as data grows.
>
> **3. Q2: More experiments done in prior works?**
>
> Please refer to Point 6 in our response to Reviewer xqBf, where we provide details on which experiments **we will add**.
>
> **4. Q3: Number of parameters?**
>
> The number of parameters in Table 2 (O(5,0) Regression Experiment) is as follows. All the models besides GLGENN and CGENN have ~150.3K parameters in total. In CGENN and GLGENN, the architecture contains sequentially applied Clifford algebra-based layers (applied to the subspaces of all grades) and ordinary MLP layers (applied only to the subspace of 0-grade (scalars)). The most significant layers and at the same time the most consuming for training are Clifford algebra-based ones, and CGENN possess ~1.8K of parameters associated with such layers, while GLGENN has ~0.6K of such parameters. For the MLP part, which is very fast and easy for training, both CGENN and GLGENN both have ~148.5K. **We will add** this information.

---

### Decision · Program_Chairs · 2025-05-01

**Decision:**

Accept (poster)

**Comment:**

This paper had 3/4 reviewers recommending acceptance. Reviewers appreciated the high difficulty of the problem tackled, strong and rigorous theoretical contributions, and high quality of presentation.

Reviewers were generally disappointed by the experiments. The method was only tested on two toy datasets and improvements with respect to CGENN were limited and seem to appear only in the small data regimen. The rebuttals provided extra experiments and promised more in an updated version.

Reviewer qfnz is the only who believes the weak experiments are reason enough for rejection, but they unfortunately did not respond to the rebuttal.

My opinion is that the strong theoretical contributions are worth being published, but I urge the authors to include the experiments listed in the rebuttal to xqBf, which should mitigate the current weak experimental section.